# Transformers Learn Faster with Semantic Focus

**Parikshit Ram**[1], **Kenneth Clarkson**[1], **Tim Klinger**[1], **Shashanka Ubaru**[1], **Alexander Gray**[2,3]

[1]IBM Research, [2]Centaur AI Institute, [3]Purdue University

parikshit.ram@ibm.com, klclarks@us.ibm.com, tklinger@us.ibm.com,
shashanka.ubaru@ibm.com, alexander.gray@centaurinstitute.org

## Abstract

Various forms of sparse attention have been explored to mitigate the quadratic computational and memory cost of the attention mechanism in transformers. We study sparse transformers not through a lens of efficiency but rather in terms of learnability and generalization. Empirically studying a range of attention mechanisms, we find that input-dependent sparse attention models appear to converge faster and generalize better than standard attention models, while input-agnostic sparse attention models show no such benefits – a phenomenon that is robust across architectural and optimization hyperparameter choices. This can be interpreted as demonstrating that concentrating a model's "semantic focus" with respect to the tokens currently being considered (in the form of input-dependent sparse attention) accelerates learning. We develop a theoretical characterization of the conditions that explain this behavior. We establish a connection between the stability of the standard softmax and the loss function's Lipschitz properties, then show how sparsity affects the stability of the softmax and the subsequent convergence and generalization guarantees resulting from the attention mechanism. This allows us to theoretically establish that input-agnostic sparse attention does not provide any benefits. We also characterize conditions when semantic focus (input-dependent sparse attention) can provide improved guarantees, and we validate that these conditions are in fact met in our empirical evaluations.

## 1 Introduction

Transformers [1] are expressive set encoders, which when paired with positional encodings, can serve as sequence encoders. The attention mechanism in a *transformer block* allows us to model the long and short term dependencies in a sequence in an input-dependent manner instead of relying on handcrafted dependency modeling as in recurrent (uni-directional and bi-directional) and convolutional models. The single hidden layer multi-layered perceptron (or MLP) in the transformer block introduces non-linearities enabling further expressivity. Transformers have been extremely successful in modeling natural language, and are the core blocks of various large language models or LLMs. They have also been successful in vision, tabular data, and time series among various other applications.

The expressivity of attention-based transformers [2] comes with a computational overhead where the attention mechanism requires time and memory quadratic in the sequence length. To address this, various efficient transformers have been developed [3], utilizing various techniques such as fixed sparse attention patterns, low rank approximations of the attention matrix, and input-dependent sparse attention patterns. In this work, we focus on sparse attention mechanisms, both input-dependent and input-agnostic. Existing literature has studied sparse attention as a way to speed up the forward pass (inference), which in turn can speed up each training step [4]. However, sparse attention has always been viewed as an approximation of the gold standard full attention.

**Contributions.** One can view sparse attention as a form of *sensory gating*, and this is considered an essential component of biological cognitive systems, allowing rapid learning [5, 6], and the absence of it is often considered a marker for schizophrenia [7]. The gating is often achieved via inhibitory signals. Related observations made by Bengio [8] suggest some motivations. He makes a connection between a form of input-dependent sparse attention and the global workspace theory of consciousness in cognitive science, as well as the properties of natural language sentences and

symbolic AI representations used in planning and reasoning [9], "*stipulating that elements of a conscious thought are selected through an attention mechanism (such as the content-based attention mechanism we introduced in Bahdanau et al. [10]) and then broadcast to the rest of the brain, strongly influencing downstream perception and action as well as the content of the next conscious thought*". As the "elements" or weight vectors being attended to are often discussed as semantic concepts, one can refer to the same phenomenon as "semantic focus" and explore its possible benefit to learning efficacy. Motivated by this, we consider the following question in this paper: "Can sparse attention in transformers be beneficial in terms of learning convergence and generalization, in comparison to full attention?". To this end, we share the following findings:

- (§3) Focusing on benchmarks of structured languages designed to evaluate capabilities of transformers [4, 11], and controlling for all involved hyperparameters, we make two empirical observations:
  - Sparse attention with input-agnostic sparsity patterns empirically struggles with expressivity (as implied by Yun et al. [2, 12]), and does not show benefits in terms of learning convergence and generalization even when equipped with enough expressivity (via *global tokens* [13, 14]).
  - Sparse attention with a specific form of input-dependent sparsity pattern that limits the attention to the top attention scores – the *heavy-hitters* (such as top-$k$ attention [15, 16]) – are empirically as expressive as the standard full attention, and can converge significantly faster during training, while generalizing as well as, and at times better than, the full attention model. These improvements hold across various hyperparameters, both related to the architecture (such as the number of heads per transformer block, the number of transformer blocks, the MLP activation function), and the optimizer (such as the initial learning rate, and the learning rate decay).
- (§4) We then try to theoretically understand why this might be happening, and characterize conditions under which sparse attention can provide better learning convergence and generalization guarantees. Our analysis is based on two critical insights:
  - For any $\lambda$-Lipschitz learning objective (with respect to the learnable parameters), the convergence rate and algorithmic stability [17] of (stochastic) gradient-descent based algorithms are dependent on Lipschitz constant $\lambda$, with smaller values implying better convergence and stability guarantees; better stability implies better generalization [18]. We show that the Lipschitz constant of a transformer-based model is tied to the input-stability of the softmax in the attention mechanism – better input-stability implies better Lipschitz constant. Thus, we establish how the input-stability of softmax directly affects the learning convergence and generalization.
  - The sparsity pattern of the sparse attention affects the overall learning convergence and generalization through its effect on the input-stability of the softmax. The input-stability of the (sparse) softmax is closely tied to the range or the *semantic dispersion* of the values (the query-key dot-products) over which the softmax is applied (formally discussed in definition 2) – larger dispersion implies worse input-stability. While input-agnostic sparsity patterns do not necessarily improve the dispersion over the full-attention model, input-dependent sparsity that only focuses on the *heavy-hitters* can significantly improve this dispersion, thus implying improved input-stability. This effectively translates to an improved Lipschitz constant, thus convergence and generalization guarantees. We also empirically validate that the dispersion and the *estimated* Lipschitz constant of input-dependent sparse attention show improvements over full attention.

**Related Work.** Transformers have been studied from various aspects since their conception. Various sparse attention transformers have been developed to improve their computational complexity [3, 19] with both input-agnostic sparsity [20–23, 14, 13] and input-dependent ones that focus the attention on the keys with the highest dot-product scores – the heavy hitters – while explicitly ignoring the rest [15, 24, 25, 16]. Various benchmarks empirically study the efficiency [4] and capabilities [11, 26, 27] of transformers. These capabilities are also theoretically characterized under various computation models [28, 29]. Please see a more detailed literature survey around transformers in appendix C.

## 2 Problem Setup

In this section, we detail the problem setup, introducing the notation, and presenting the details of the transformer-based model, the training data and the learning loss.

**Notation.** We denote the index set as $[n] \triangleq \{1, \ldots, n\}$ for any natural number $n \in \mathbb{N}$. We use $X$ for input sequences of token indices $v \in [D]$ in a vocabulary $\mathcal{V}$ of size $D$, and $y$ for labels or targets. We use $\mathbf{x} \in \mathbb{R}^d$ for a token embedding vector and $\mathbf{X} \in \mathbb{R}^{d \times L}$ for the sequence (matrix) of $L$ token embeddings. For any vector $\mathbf{v}$, we use $v_i$ to denote its $i$-th entry, and $\|\mathbf{v}\|$ to denote

its Euclidean norm. For a matrix $\mathbf{W}$, we denote its $(i,j)$-th entry as $W_{ij}$, $i$-th column as $\mathbf{W}_{:i}$ and $i$-th row as $\mathbf{W}_{i:}$. We use $\|\mathbf{W}\|$ and $\|\mathbf{W}\|_{2,1}$ to denote the spectral and $\ell_{2,1}$ norms of $\mathbf{W}$. For a tuple $\theta = (\mathbf{W}^{(1)}, \ldots, \mathbf{W}^{(n)})$ of $n$ matrices, we let $\|\theta\| = \max_{i \in [n]} \|\mathbf{W}^{(i)}\|$. We consider a learning problem with input sequences $X = [v_1, \ldots, v_L] \in [D]^L$ of length exactly $L$ with its $i$-th entry $v_i$ denoting the $v_i$-th token in a vocabulary $\mathcal{V}$, with outputs $y \in \mathcal{Y}$. [1] For a learnable function $f : \mathcal{X} \to \mathcal{Y}$ with learnable parameters $\theta$, we explicitly write the function as $f_\theta(X)$ with $X \in \mathcal{X}$.

**Transformer block.** For a $L$ length sequence of token embeddings $\mathbf{X} \in \mathbb{R}^{d \times L}$ with the $i$-th token embedding denoted as $\mathbf{X}_{:i} \in \mathbb{R}^d$, let $\mathsf{TF} : \mathbb{R}^{d \times L} \to \mathbb{R}^{d \times L}$ denote a transformer block with learnable parameters $\theta = (\mathbf{W}, \mathbf{V}, \mathbf{P}, \mathbf{R})$ with $\mathbf{W}, \mathbf{V} \in \mathbb{R}^{d \times d}, \mathbf{P}, \mathbf{R} \in \mathbb{R}^{d_{\mathsf{MLP}} \times d}$ defined as:

$$\mathsf{TF}_\theta(\mathbf{X}) = \mathsf{LN}(\widetilde{\mathbf{X}} + \underbrace{\mathbf{R}^\top \sigma(\mathbf{P}\widetilde{\mathbf{X}})}_{\mathsf{MLP}_{\mathbf{P},\mathbf{R}}(\widetilde{\mathbf{X}})}), \quad \text{and} \quad \widetilde{\mathbf{X}} = \mathsf{LN}(\mathbf{X} + \underbrace{\mathbf{V}\mathbf{X}\,\mathsf{softmax}(\mathbf{X}^\top \mathbf{W} \mathbf{X})}_{\mathsf{A}_{\mathbf{W},\mathbf{V}}(\mathbf{X})}), \quad (1)$$

where $\mathsf{LN} : \mathbb{R}^d \to \mathbb{R}^d$ is the token-wise (columnwise) layer normalization (or LayerNorm), and $\mathbf{R}^\top \sigma(\mathbf{P}\widetilde{\mathbf{X}})$ denotes the token-wise single hidden layer MLP $: \mathbb{R}^d \to \mathbb{R}^d$. The columnwise $\mathsf{softmax}(\cdot)$ of the dot-products $\mathbf{D} = \mathbf{X}^\top \mathbf{W} \mathbf{X}$ between the query and key matrices, [2] combined with the value matrix $\mathbf{V}\mathbf{X}$, denotes the dot-product self-attention $\mathsf{A} : \mathbb{R}^{d \times L} \to \mathbb{R}^{d \times L}$. We consider single head attention here for the ease of exposition, but our analysis can be easily extended to multi-headed attention (see appendix F.4). While Vaswani et al. [1] utilized ReLU as the activation $\sigma$ in the MLP, subsequent works [30] have used other activations such as GELU [31] and ELU [32]. Furthermore, many different variations of the transformer block has also been utilized in literature. [3]

**Masked softmax.** A common modification of this transformer block is the replacement of the softmax with a sparse *masked* softmax which has an associated masking function $m : \mathbb{R}^{L \times L} \to \{0, 1\}^{L \times L}$ with $\mathbf{M} = m(\mathbf{D})$ for a dot-product matrix $\mathbf{D}$. The $(j, i)$-th entry $A_{ji}$ of the post-activation attention matrix $\mathbf{A} = \mathsf{softmax}(\mathbf{D})$ for standard and masked attention is given as follows:

$$A_{ji} = \frac{\exp(D_{ji})}{\sum_{j'=1}^L \exp(D_{j'i})}, \qquad A_{ji} = \frac{\exp(D_{ji}) \cdot M_{ji}}{\sum_{j'=1}^L \exp(D_{j'i}) \cdot M_{j'i}}. \qquad (2)$$

**Complete model.** The model is defined as $f_\Theta : [D]^L \to \hat{\mathcal{Y}}$ with token and position embeddings $\mathbf{T} \in \mathbb{R}^{d \times D}$ and $\mathbf{E} \in \mathbb{R}^{d \times L}$ respectively, $\tau$ transformer blocks each with parameters $\theta^{(t)} = (\mathbf{W}^{(t)}, \mathbf{V}^{(t)}, \mathbf{P}^{(t)}, \mathbf{R}^{(t)}), t \in [\tau]$, and a readout linear layer with weights $\mathbf{\Phi} \in \mathbb{R}^{Y \times d}$ using token projection vector $\boldsymbol{\omega} \in \mathbb{R}^L$, where $Y$ is the dimensionality of the output $\hat{\mathcal{Y}}$ (for example, the number of classes in output domain $\mathcal{Y}$). The $i$-th token $v_i \in [D]$ in the input $X$ is initially embedded as $\mathbf{X}_{:i}^{(0)} = \mathbf{T}_{:v_i} + \mathbf{E}_{:i}$ using the token and position embeddings:

$$f_\Theta(X) = \mathbf{\Phi}(\mathbf{X}^{(\tau)}\boldsymbol{\omega}), \quad \mathbf{X}^{(t)} = \mathsf{TF}_{\theta^{(t)}}(\mathbf{X}^{(t-1)}), \forall t \in [\tau]. \qquad (3)$$

Here $\boldsymbol{\omega} \in \mathbb{R}^L$ is the (*fixed*) token projection vector – we can set the $\boldsymbol{\omega} = [0, 0, \ldots, 0, 1]^\top$ to select the last token to make the final prediction, and $\boldsymbol{\omega} = (1/L)\mathbf{1}_L$ uses the average of the $L$ tokens (along the sequence length dimension), where $\mathbf{1}_L$ is the all-one $L$ dimensional vector. The $\Theta$ in $f_\Theta(\cdot)$ denotes the tuple of all the (learnable) model parameters, that is $\Theta \triangleq (\mathbf{T}, \theta^{(1)}, \ldots, \theta^{(\tau)}, \mathbf{\Phi})$. Here we are assuming that the position encodings are not learned, but that can also be incorporated in our study.

**Training.** Given a set $S$ of $n$ sequence-output pairs $(X, y), X \in [D]^L, y \in \mathcal{Y}$ for training, and a per-sample loss function $\ell : \mathcal{Y} \times \hat{\mathcal{Y}} \to \mathbb{R}$, the learning involves solving the following empirical risk minimization or ERM problem:

$$\min_{\Theta \triangleq (\mathbf{T}, \theta^{(1)}, \ldots, \theta^{(\tau)}, \mathbf{\Phi})} \mathcal{L}(\Theta) \triangleq \frac{1}{n} \sum_{(X,y) \in S} \ell(y, f_\Theta(X)) \quad (f_\Theta(\cdot) \text{ defined in equation (3)}). \qquad (4)$$

In the sequel, we will study, first empirically and then theoretically, (i) the convergence rate of stochastic gradient descent for this learning problem, and (ii) the generalization of the learned model.

---

[1] In our experiments, we consider supervised learning with $\mathcal{Y}$ as a set of labels, but our analysis applies to any $\mathcal{Y}$ where we have a scalar loss $\ell : \mathcal{Y} \times \hat{\mathcal{Y}} \to \mathbb{R}$, where $\hat{\mathcal{Y}}$ is the model output space: $\hat{\mathcal{Y}} \subset \mathbb{R}^{|\mathcal{Y}|}$ for multi-class classification with cross-entropy loss, and $\mathcal{Y} = \hat{\mathcal{Y}} \subset \mathbb{R}^m$ for $m$-output regression with mean-squared loss.

[2] With queries $\mathbf{Q}\mathbf{X}$, keys $\mathbf{K}\mathbf{X}$, the scores $(\mathbf{Q}\mathbf{X})^\top(\mathbf{K}\mathbf{X}) = \mathbf{X}^\top(\mathbf{Q}^\top\mathbf{K})\mathbf{X}$; we denote $\mathbf{W} = \mathbf{Q}\mathbf{K}^\top$.

[3] For example, instead of the transformer block described in equation (1), there are versions that modify the location where the LayerNorm is applied: $\mathsf{TF}_\theta(\mathbf{X}) = \widetilde{\mathbf{X}} + \mathsf{MLP}_{\mathbf{P},\mathbf{R}}(\mathsf{LN}(\widetilde{\mathbf{X}}))$ and $\widetilde{\mathbf{X}} = \mathbf{X} + \mathsf{A}_{\mathbf{W},\mathbf{V}}(\mathsf{LN}(\mathbf{X}))$.

# 3 Empirical Observations

In this section, we focus on empirically ablating the effect of the different forms of sparse attention on the ERM convergence and generalization. For this purpose, we ensure that all hyperparameters (architectural and optimization) are the same between the standard full attention, and the various sparse attention mechanisms. We consider a total of 8 tasks from LRA [4] and the NNCH benchmark [11]; we present results from 3 of the tasks here (results for all the tasks can be found in appendix E); we also present preliminary results on a NLP next-token prediction task in appendix E.3. Details on the tasks and the sparse attention choices, along with the hyperparameter (architectural and optimization) selection procedure and our compute resources are discussed in appendix D. First, we will present the comparison between the standard full attention and various sparse attentions. Then, focusing on full attention and heavy-hitter style input-dependent sparse attention, we will study the effect of hyperparameters (or the lack thereof) on their relative behaviors.

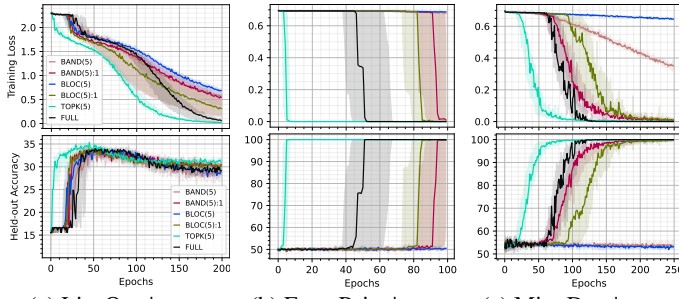

(a) List Ops | ReLU (b) Even Pairs | ReLU (c) Miss Dup | ReLU

Figure 1: Learning convergence and generalization curves for full attention and various sparse attention based models. Each column corresponds to a task. *The legend is the same across all datasets* – BAND(5) denotes banded attention with a band size of 5; BAND(5):1 denotes the same with a single global token. BLOC(5) denotes block local attention with a block size of 5; BLOC(5):1 denotes the same with single global token. TOPK(5) is top-$k$ attention with $k = 5$. **Top row**: Training cross-entropy loss trajectories – lower is better. **Bottom row**: Generalization performance on held-out set as training progresses – higher is better. Further results for 8 tasks with different mask sizes and global tokens is presented in appendix E.

We present our first set of experimental results in figure 1, comparing the overall learning convergence and generalization of full attention based models to those using sparse attention. Here we use ReLU for the MLP in the transformer block as in the original configuration [1]. These results are aggregated over 10 repetitions, and we present the median performance and its inter-quartile range. We present results for a single sparsity level (mask size) here; please see appendix E.1 for more variations.

**Observation I.** Input-dependent heavy-hitter sparse attention speeds up learning convergence while input-agnostic sparse attention do not show any improvement over full attention.

The results in figure 1 (top row) show that the input-agnostic sparse attention often converges slower than full attention. They also often struggle with expressivity in the absense of the global tokens, as seen for both block local and banded attention with Even Pairs and Missing Duplicates. This is expected as per the expressivity results of Yun et al. [12]. The inclusion of the global token addresses this issue. In contrast, the training loss of top-$k$ attention converges significantly faster than full attention. Top-$k$ attention shows improvements (in terms of achieving 95% training accuracy) over full attention ranging between $1.37\times$ (121 epochs vs 167 epochs) with ListOps to $8.83\times$ (6 epochs vs 53 epochs) with Even Pairs (see table 4 in appendix E.1). In all cases, top-$k$ attention is able to be as expressive as full attention without the need for any global tokens. This consistent faster training of top-$k$ attention in terms of the number of optimization steps needed to converge is not something discussed in existing literature to the best of our knowledge.

**Observation II.** Input-dependent heavy-hitter sparse attention generalizes faster during learning.

The results in figure 1 (bottom row) show that the input-dependent sparse attention achieves similar (Even Pairs and Missing Duplicates) or better (ListOps) holdout accuracy when compared to full attention. Furthermore, it attains this generalization level much earlier during the training process. Note that input-dependent heavy-hitter top-$k$ achieves better empirical generalization performance both in terms of the highest holdout accuracy during the training trajectory, and the final holdout accuracy. The latter highlights that the faster ERM convergence of input-dependent sparse attention does not lead to overfitting. In fact, with the ListOps task, the final holdout accuracy with full attention drops from around $35.1 \pm 0.6\%$ to $28.9 \pm 1.4\%$, while the drop with top-$k$ attention is only from $36.3 \pm 0.3\%$ to $31.3 \pm 0.9\%$ (see table 3 in appendix E for complete results). In general, the top-$k$ attention based transformers also have comparitively lower variations in their performance as evidenced by the fairly tight inter-quartile ranges of the trajectories of the loss and accuracy.

Next we study the effect of the different hyperparameter choices on the relative performances. First we change the activation in the MLP of the transformer block to evaluate whether the differences in empirical performance are due to the attention or the MLP in the block. Then, we vary the number of blocks and the number of heads in the model. Finally, we vary various optimizer hyperparameters such as the learning rate and its scheduling as well as the optimizer itself.

> **Observation III.** The gain from the input-dependent heavy-hitter sparse attention in terms of convergence and generalization is unaffected by the choice of the activation function $\sigma$ in the MLP of a transformer block.

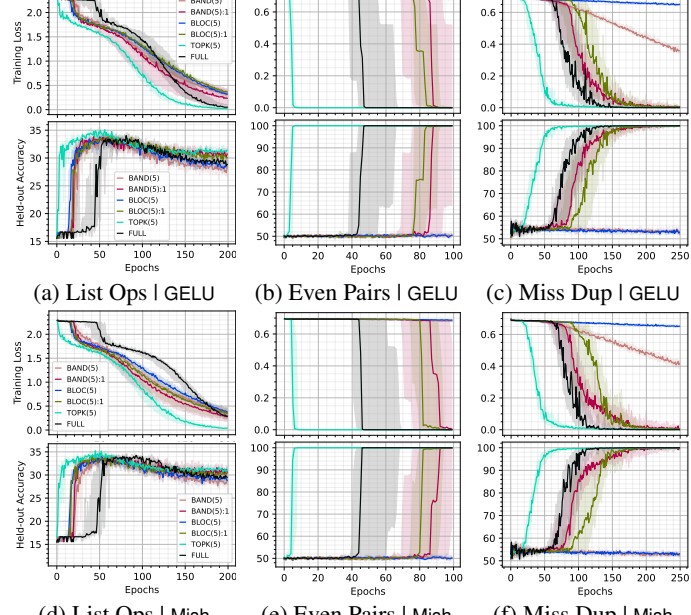

(a) List Ops | GELU  (b) Even Pairs | GELU  (c) Miss Dup | GELU

(d) List Ops | Mish  (e) Even Pairs | Mish  (f) Miss Dup | Mish

Figure 2: Same experimental setup as figure 1 with GELU activation (top 2 rows) and Mish activation (bottom 2 rows). See additional results in appendix E.2.

We present the performances of the different attention mechanisms with the GELU activation [31] in figure 2 (top 2 rows) and with the Mish activation [33] in figure 2 (bottom 2 rows), while keeping all other hyperparameters exactly the same as in figure 1 to ablate the effect of the change in the MLP activation. Comparing these results to figure 7, we see that there is not a lot of qualitative difference in the performances both in terms of learning convergence and generalization, indicating that the difference in performance is due to the difference in the attention mechanism of the transformer block.

> **Observation IV.** The improvement of input-dependent heavy-hitter sparse attention over full attention is not affected by the number of transformer blocks, and increases with the number of heads in each transformer block.

We study the effect of varying the number of transformer blocks (figure 3a) and the number of attention heads per transformer block (figure 3b). We have again fixed all other hyperparameters as in figure 1 to ablate the effect of the considered architectural changes. Here we focus on learning convergence of full and top-$k$ attention (with $k = 5$) for ListOps. The results indicate: (i) Top-$k$ attention continues to converge faster than full attention across all number of blocks $\tau$ tried ($\tau \in \{6, 10, 15, 22\}$). The relative performance is not affected by the number of blocks. (ii) Top-$k$ attention continues to converge faster than their full attention variants as the number of heads increase from 1 to 4 and 8. The convergence of the full attention model slows down while the convergence of top-$k$ attention stays almost the same, thus increasing the relative improvement with the number of heads.

> **Observation V.** The improvement of the input-dependent heavy-hitter sparse attention holds across varying optimizer hyperparameters, especially for hyperparameters that have the most promising convergence.

We present the effect of varying SGD parameters for full and top-$k$ attention on ListOps. In figure 3c, we fix the decay rate to 0.99 (as in figure 1) and vary the initial learning rate. Smaller initial learning rates (0.66 and 1) have the best convergence for both full and top-$k$ attention, with top-$k$ converging faster. For larger initial learning rates (1.5 and 2.25), convergence slows down for both, and the difference between full and top-$k$ attention is less pronounced, though top-$k$ appears to be slightly better, especially initially. In figure 3d, we fix the initial learning rate to 1.0, and vary the decay rate. For slower decay rates (0.9999 and 0.999), the overall convergence for both methods slow down though top-$k$ continues to converge faster. For faster decay rate of 0.9, top-$k$ initially appears to outperform full attention with a big margin. However, both methods prematurely stall as the learning rate becomes too small.

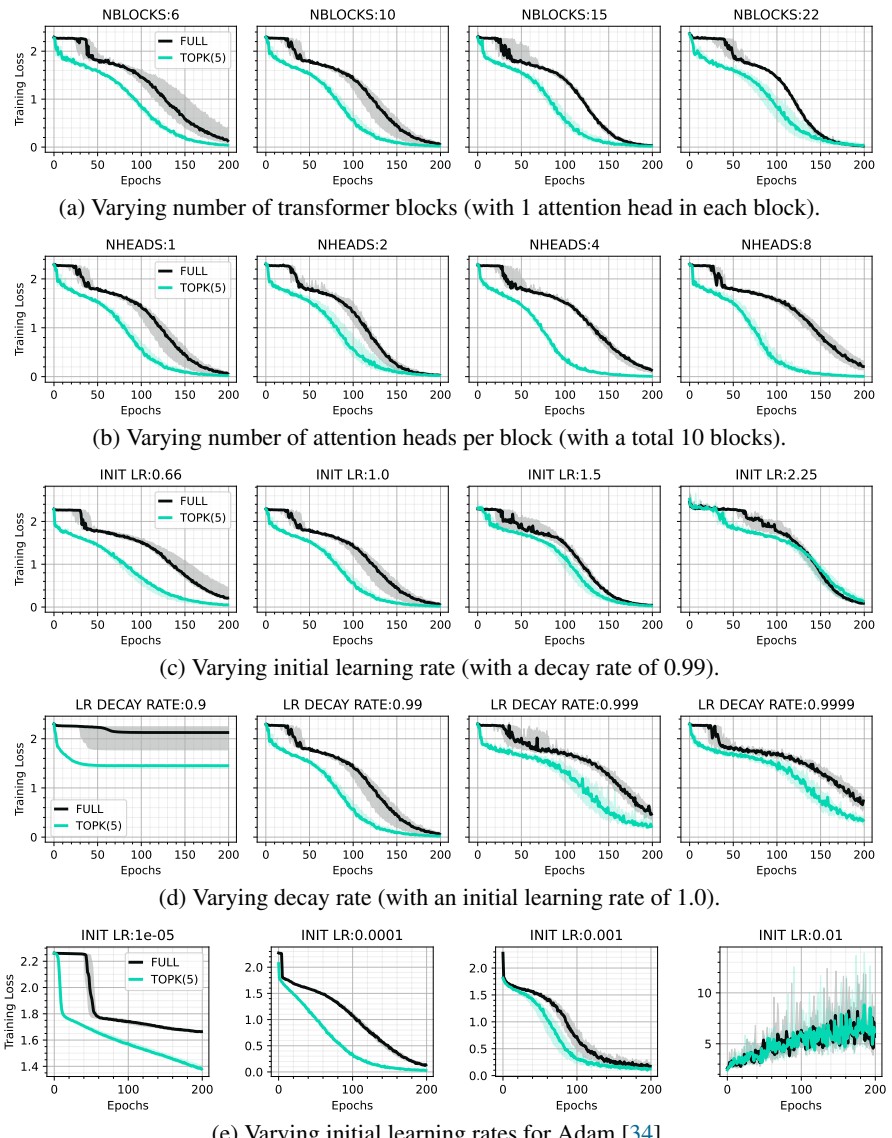

(a) Varying number of transformer blocks (with 1 attention head in each block).

(b) Varying number of attention heads per block (with a total 10 blocks).

(c) Varying initial learning rate (with a decay rate of 0.99).

(d) Varying decay rate (with an initial learning rate of 1.0).

(e) Varying initial learning rates for Adam [34].

Figure 3: Comparison of full and top-$k$ attention training loss trajectories for varying hyperparameters, both architectural (number of transformer blocks and attention heads), and optimization (initial learning rate, learning rate decay, and optimizer) with the List Ops task.

> **Observation VI.** The improvement of the input-dependent heavy-hitter sparse attention over full-attention also holds for the Adam optimizer with varying learning rates, especially for hyperparameters that have the most promising convergence.

While all our previous results use SGD, in figure 3e we evaluate whether the relative performances translate to the more widely used Adam optimizer [34] on ListOps. We evaluate various learning rates, and see that the learning rate that provided convergence for SGD (initial learning rate around 0.1-1.0) lead to divergence with Adam (see 4th column in figure 3e). With smaller learning rates, we see that the improved convergence of the input-dependent heavy-hitter sparse attention persists.

## 4    Theoretical Understanding

The empirical observations in the previous section demonstrate that input-agnostic sparse attention can struggle with expressivity, and does not show any consistent benefit over full attention. In contrast, input-dependent heavy-hitter top-$k$ attention shows significant advantages. In this section, we want to theoretically understand why this might be happening. We begin by considering the factors that affect the convergence and generalization of SGD based training.

Standard SGD analysis show that, for a $\alpha_1$-Lipschitz and $\alpha_2$-smooth finite-sum (non-convex) objective, with learning rates $\eta_i$ at the $i$-th step, learning converges to a $\epsilon$-stationary point in $T$ steps where $\epsilon \sim O(\alpha_2 \alpha_1^2 (\sum_{i=0}^{T-1} \eta_i^2)/(\sum_{i=0}^{T-1} \eta_i))$. Different choices of $\eta_i, i \in [T]$ (such as $\eta/i$ or $\eta/\sqrt{i}$ for some constant $\eta > 0$) provide different convergence rates (such as $O(1/\log(T))$ or $O(1/\sqrt{T})$). As we control for the learning rate and its scheduling in our experiments, and ensure that all models start learning from the same initial parameters, the main distinction between the different forms of attention could be the Lipschitz constant $\alpha_1$ and the smoothness constant $\alpha_2$. Note that, with non-smooth activation function like ReLU, we are effectively performing stochastic sub-gradient descent, where the guarantees are much weaker but still depend on the Lipschitz constant.

Generalization error is the difference between empirical risk (on the training samples) and the true risk (over the population). A low training error combined with a low generalization error implies strong performance on unseen data. Utilizing the seminal work [17] on algorithmic stability, Hardt et al. [18, Theorem 2.2] show that learning with a $\varepsilon$-stable randomized algorithm guarantees an expected generalization error of at most $\varepsilon$. Furthermore, for $\alpha_1$-Lipschitz and $\alpha_2$-smooth finite-sum nonconvex objective, the $T$ step SGD algorithm with per-step learning rate $\eta_i \leq \eta/i$ is $\varepsilon$-uniformly stable with $\varepsilon \sim O\left((\eta\alpha_1^2)^{1/1+\alpha_2\eta}(1 + 1/\alpha_2\eta)T^{\alpha_2\eta/1+\alpha_2\eta}\right)$ [18, Theorem 3.12]. As before, the distinguishing factors between our models pertinent to generalization are the Lipschitz and smoothness constants.

Based on this intuition, we will focus on the Lipschitz constant. First, we will try to characterize how the behavior of the softmax – specifically the input stability of softmax – in the attention mechanism of the transformer block affects the Lipschitz continuity of the overall learning objective.

**Definition 1.** *A masked* softmax *is $\xi$-input-stable if $\forall \mathbf{z}, \bar{\mathbf{z}} \in \mathbb{R}^d$, $\|\text{softmax}(\mathbf{z}) - \text{softmax}(\bar{\mathbf{z}})\|_1 \leq \xi\|\mathbf{z} - \bar{\mathbf{z}}\|_1$. The attention $\mathsf{A}$ with parameters $\mathbf{W}, \mathbf{V}$ is stable with respect to its input and parameters if $\forall \mathbf{X}, \bar{\mathbf{X}} \in \mathbb{R}^{d \times L}, \mathbf{W}, \bar{\mathbf{W}}, \mathbf{V}, \bar{\mathbf{V}} \in \mathbb{R}^{d \times d}$, with constants $\lambda_X(\xi), \lambda_W(\xi)$ depending on $\xi$:*

$$\|\mathsf{A}_{\mathbf{W},\mathbf{V}}(\mathbf{X}) - \mathsf{A}_{\mathbf{W},\mathbf{V}}(\bar{\mathbf{X}})\|_{2,1} \leq \lambda_X(\xi)\|\mathbf{X} - \bar{\mathbf{X}}\|_{2,1}, \tag{5}$$

$$\|\mathsf{A}_{\mathbf{W},\mathbf{V}}(\mathbf{X}) - \mathsf{A}_{\bar{\mathbf{W}},\mathbf{V}}(\mathbf{X})\|_{2,1} \leq \lambda_W(\xi)\|\mathbf{W} - \bar{\mathbf{W}}\|, \tag{6}$$

$$\|\mathsf{A}_{\mathbf{W},\mathbf{V}}(\mathbf{X}) - \mathsf{A}_{\mathbf{W},\bar{\mathbf{V}}}(\mathbf{X})\|_{2,1} \leq \lambda_V\|\mathbf{V} - \bar{\mathbf{V}}\|. \tag{7}$$

We will precisely characterize the values of the constants in the above definition ($\xi, \lambda_X(\xi), \lambda_W(\xi), \lambda_V$) for the different (masked) softmax and corresponding (masked) self-attention in the sequel. However, we define them here to highlight how the stability of softmax affects the stability of the self-attention $\mathsf{A}$, and how this affects the Lipschitz continuity of the learning objective in equation (4) with respect to the model parameters $\Theta = (\mathbf{T}, \theta^{(1)}, \dots, \theta^{(\tau)}, \mathbf{\Phi})$. For completeness, we first need to establish the stability properties of the MLP component of a TF block (see proof in appendix F.1):

**Lemma 1.** *Assuming that the* MLP *activation $\sigma$ is $\lambda_\sigma$-Lipschitz with $\sigma(0) = 0$, and the* MLP *parameters have norms bounded by $B > 0$, that is $\|\mathbf{P}\| \leq B$ and $\|\mathbf{R}\| \leq B$, the token-wise* MLP *and* LN *operations are stable with respect to their input and model parameters as follows $\forall \mathbf{x}, \mathbf{x}' \in \mathbb{R}^d, \|\mathbf{x}\|, \|\bar{\mathbf{x}}\| \leq \Xi, \mathbf{P}, \bar{\mathbf{P}} \in \mathbb{R}^{d_{\text{MLP}} \times d}, \mathbf{R}, \bar{\mathbf{R}} \in \mathbb{R}^{d_{\text{MLP}} \times d}$, with $\eta_X = B^2\lambda_\sigma, \eta_P = \eta_R = \lambda_\sigma B \Xi$:*

$$\|\text{MLP}_{\mathbf{P},\mathbf{R}}(\mathbf{x}) - \text{MLP}_{\mathbf{P},\mathbf{R}}(\bar{\mathbf{x}})\| \leq \eta_X\|\mathbf{x} - \bar{\mathbf{x}}\|, \quad \|\text{MLP}_{\mathbf{P},\mathbf{R}}(\mathbf{x}) - \text{MLP}_{\bar{\mathbf{P}},\mathbf{R}}(\mathbf{x})\| \leq \eta_P\|\mathbf{P} - \bar{\mathbf{P}}\|,$$

$$\|\text{MLP}_{\mathbf{P},\mathbf{R}}(\mathbf{x}) - \text{MLP}_{\mathbf{P},\bar{\mathbf{R}}}(\mathbf{x})\| \leq \eta_R\|\mathbf{R} - \bar{\mathbf{R}}\|, \quad \|\text{LN}(\mathbf{x}) - \text{LN}(\bar{\mathbf{x}})\| \leq \zeta_{\text{LN}}\|\mathbf{x} - \bar{\mathbf{x}}\|. \tag{8}$$

The Lipschitz continuity of the LayerNorm has been previously established in Kim et al. [35]. Given this, we can establish the following results for a transformer block (see proof in appendix F.2):

**Theorem 1.** *Given definition 1 and lemma 1, a transformer block* TF *with learnable parameters $\theta = (\mathbf{W}, \mathbf{V}, \mathbf{P}, \mathbf{R})$ is $\lambda_\theta(\xi)$-stable with respect to its learnable parameters $\theta$ with*

$$\lambda_\theta(\xi) = \zeta_{\text{LN}}\left(\zeta_{\text{LN}}(1 + \eta_X)(\lambda_W(\xi) + \lambda_V) + L(\eta_P + \eta_R)\right), \tag{9}$$

*and* TF *is $\lambda_{\mathbf{X}}(\xi)$-stable with respect to its input $\mathbf{X}$ with $\lambda_{\mathbf{X}}(\xi) = \zeta_{\text{LN}}^2(1 + \eta_X)(1 + \lambda_X(\xi))$, where we explicitly note the dependence on the stability $\xi$ of the (masked)* softmax *operation. Thus, for any parameter tuples $\theta, \bar{\theta}$ and input $\mathbf{X}, \bar{\mathbf{X}}$, we have*

$$\|\text{TF}_\theta(\mathbf{X}) - \text{TF}_{\bar{\theta}}(\mathbf{X})\|_{2,1} \leq \lambda_\theta(\xi)\|\theta - \bar{\theta}\|, \quad \|\text{TF}_\theta(\mathbf{X}) - \text{TF}_\theta(\bar{\mathbf{X}})\|_{2,1} \leq \lambda_{\mathbf{X}}(\xi)\|\mathbf{X} - \bar{\mathbf{X}}\|. \tag{10}$$

Thus, we establish the following for our model with $\tau$ transformer blocks (proof in appendix F.3):

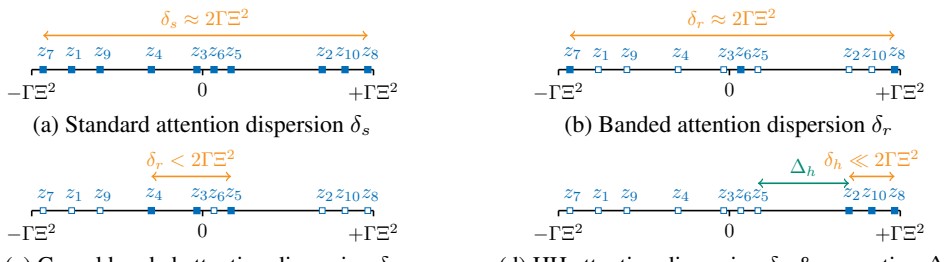

Figure 4: Semantic dispersion $\delta$ (definition 2) and heavy-hitter (HH) attention semantic separation $\Delta$ (definition 3): For a sequence of length $L = 10$, we demonstrate the concepts for query token $\mathbf{X}_{:6}$. Let $z_j = \mathbf{X}_{:6}^\top \mathbf{W} \mathbf{X}_{:j}$ denote the $j^{\text{th}}$ query-key dot-product. (a) Figure 4a shows that in full attention, the $z_j$s (■) can range between $-\Gamma\Xi^2$ and $+\Gamma\Xi^2$ (theorem 1), giving us a semantic dispersion $\delta_s \approx 2\Gamma\Xi^2$. In general, we cannot expect a tighter bound on $\delta_s$. (b) Figure 4b shows the same example for an input-agnostic banded masked attention, where the query token $\mathbf{X}_{:6}$ only attends to succeeding key tokens $\mathbf{X}_{:6}, \mathbf{X}_{:7}, \mathbf{X}_{:8}$ (■), while the rest are masked (□). Here, the dispersion $\delta_r \approx \delta_s \approx 2\Gamma\Xi^2$, no better than full-attention. (c) Figure 4c shows the example with an input-agnostic causal banded attention mask where token $\mathbf{X}_{:6}$ only attends to the preceding key tokens $\mathbf{X}_{:5}, \mathbf{X}_{:4}, \mathbf{X}_{:3}$. Here, this masked attention has a small dispersion $\delta_r < 2\Gamma\Xi^2$ better than that of full-attention $\delta_s \approx 2\Gamma\Xi^2$. However, there is usually no way to ensure that a condition where $\delta_r \ll \delta_s$ will exist. (d) Figure 4d shows the example with an input-dependent HH attention, where only the high values are unmasked, and there is a significant semantic separation $\Delta_h$ between the masked and unmasked dot-products. Here, we can expect significantly smaller semantic dispersion $\delta_h \ll 2\Gamma\Xi^2$ implying $\delta_h \ll \delta_s$.

**Theorem 2.** *Assuming that the per-sample loss function $\ell$ in (4) is $\alpha$-Lipschitz and $\|\mathbf{\Phi}\| \leq 1$ with $\omega = (1/L)\mathbf{1}_L$, under the conditions of definition 1 and theorem 1, the learning objective $\mathcal{L}$ in (4) is $\lambda_{\mathcal{L}}(\xi)$-Lipschitz with respect to the learnable parameters $\Theta = (\mathbf{T}, \theta^{(1)}, \dots, \theta^{(\tau)}, \mathbf{\Phi})$, where*

$$\lambda_{\mathcal{L}}(\xi) = \alpha \left( \Xi + \lambda_{\mathbf{X}}(\xi)^\tau \left( 1 + \frac{\lambda_\theta(\xi)}{L(\lambda_{\mathbf{X}}(\xi) - 1)} \right) \right), \tag{11}$$

*and $|\mathcal{L}(\Theta) - \mathcal{L}(\bar{\Theta})| \leq \lambda_{\mathcal{L}}(\xi)\|\Theta - \bar{\Theta}\|$ for any set of model parameters $\Theta, \bar{\Theta}$.*

This characterizes how the Lipschitz constant of the learning loss, and thus the convergence rate of the SGD based ERM, is tied to the input-stability constant $\xi$ of the (masked) softmax. Thus, based on theorem 1, the larger the values of $\lambda_W(\xi)$, $\lambda_X(\xi)$ and $\lambda_V$ in definition 1, the larger the Lipschitz constant of the training loss. We will characterize these quantities in the sequel. To understand the effect of sparsity on the stability of softmax, we begin with the stability of the standard full softmax and the standard full attention in the following (see lemma 4 in appendix G.1):

**Lemma 2** (adapted from Li et al. [36] Lemma B.1). *For any $\mathbf{z}, \bar{\mathbf{z}} \in \mathbb{R}^L$ with $\max_{i,j \in [L]}(z_i - z_j) \leq \delta$, and $\max_{i,j \in [L]}(\bar{z}_i - \bar{z}_j) \leq \delta$, for a positive constant $\delta > 0$, we have the following:*

$$\|\text{softmax}(\mathbf{z})\|_\infty \leq e^\delta / L, \quad \|\text{softmax}(\mathbf{z}) - \text{softmax}(\bar{\mathbf{z}})\|_1 \leq (e^\delta / L)\|\mathbf{z} - \bar{\mathbf{z}}\|_1. \tag{12}$$

A critical factor in the softmax stability is this quantity $\delta$ that is the upper bound on the difference between the largest and smallest values over which the softmax is applied. In the context of dot-product self-attention, it corresponds to the difference between the largest and smallest query-key dot-products for any query. We term this as *semantic dispersion*, and define it precisely as follows:

**Definition 2.** *For a (sparse) attention transformer block with $L$ length input sequences $\mathbf{X} \in \mathbb{R}^{d \times L}$, and a mask $\mathbf{M} \in \{0, 1\}^{L \times L}$ (input dependent or input agnostic), we define the per-query semantic dispersion as a scalar $\delta > 0$ such that, for any query token $\mathbf{X}_{:i}, i \in [L]$ the maximum difference between the largest and smallest unmasked query-key dot-products is bounded from above by $\delta$. That is, for any input sequence of token representations $\mathbf{X} \in \mathbb{R}^{d \times L}$, mask $\mathbf{M} \in \{0, 1\}^{L \times L}$ and attention parameters $\mathbf{W} \in \mathbb{R}^{d \times d}$, for all query tokens $i \in [L]$, we have*

$$\delta \geq \max_{j,j' \in [L]: M_{ji} = M_{j'i} = 1} \left( \mathbf{X}_{:i}^\top \mathbf{W} \mathbf{X}_{:j} - \mathbf{X}_{:i}^\top \mathbf{W} \mathbf{X}_{:j'} \right). \tag{13}$$

We discuss this definition with examples in figure 4. We now establish the stability of standard softmax and self-attention A by characterizing $\xi, \lambda_X(\xi), \lambda_W(\xi)$ in definition 1 in terms of the semantic dispersion as follows (see theorem 7, appendix G.1 for details):

**Theorem 3** (partially adapted from [36] Lemma B.2). *Assuming that the per-token Euclidean norms are bounded as $\|\mathbf{X}_{:i}\| \leq \Xi \, \forall i \in [L]$, and the parameter norms are bounded at $\|\mathbf{W}\| \leq \Gamma$ and $\|\mathbf{V}\| \leq \Upsilon$, and the per-query semantic dispersion is bounded by $\delta_s > 0$. Then the standard* softmax *is $\xi_s$-stable with $\xi_s = e^{\delta_s}/L$, and the standard attention is stable as in definition 1 with (i) $\lambda_X(\xi_s) = \xi_s \Upsilon L(2\Gamma\Xi^2 + 1) = e^{\delta_s} \Upsilon(2\Gamma\Xi^2 + 1)$, (ii) $\lambda_W(\xi_s) = \xi_s \Upsilon L^2\Xi^3 = e^{\delta_s}\Upsilon L\Xi^3$, (iii) $\lambda_V = L\Xi$.*

The semantic dispersion $\delta_s$ plays a significant role in $\lambda_X(\xi_s)$ and $\lambda_W(\xi_s)$, with larger values implies higher per-transformer-block stability constants $\lambda_\theta(\xi_s)$ and $\lambda_{\mathbf{X}}(\xi_s)$ in theorem 1. As discussed in figure 4a, we cannot expect this dispersion $\delta_s$ to be significantly smaller than $2\Gamma\Xi^2$.

Next we study the stability of input-agnostic *regular $k$-sparse* attention transformers, where *each query attends to exactly $k$ keys, and each key is attended to by exactly $k$ queries*. This form includes banded attention [20], block-local attention [21] and strided attention [22, 23]; random attention [14] satisfies this only in expectation. We establish its properties as follows (see theorem 8 in appendix G.2):

**Theorem 4.** *Consider self-attention with a $k$-regular input-agnostic mask $\mathbf{M}$. Assume that the per-token Euclidean norms are bounded as $\|\mathbf{X}_{:i}\| \leq \Xi \, \forall i \in [L]$, the parameter norms are bounded at $\|\mathbf{W}\| \leq \Gamma$, $\|\mathbf{V}\| \leq \Upsilon$, and the per-query semantic dispersion is bounded by $\delta_r$. Then the masked* softmax *is $\xi_r$-stable with $\xi_r = e^{\delta_r}/k$, and the attention is stable as in definition 1 with (i) $\lambda_X(\xi_r) = \xi_r \Upsilon k(2\Gamma\Xi^2 + 1) = e^{\delta_r}\Upsilon(2\Gamma\Xi^2 + 1)$, (ii) $\lambda_W(\xi_r) = \xi_r \Upsilon L k\Xi^3 = e^{\delta_r}\Upsilon L\Xi^3$, (iii) $\lambda_V = L\Xi$.*

This shows that this input-agnostic sparse attention provides guarantees very similar to full attention except for the $e^{\delta_r}$ term, implying significant improvement in stability **only if** the per-query semantic dispersion $\delta_r$ is sufficiently small relative to the full attention dispersion $\delta_s$; see one such situation in figure 4c. In general, the dispersion $\delta_r$ would be small only if the per-query dot-products somehow align with the sparsity patterns – with temporal locality based patterns (banded, block-local), the dot-products for nearby keys (in terms of sequence position) would require to have a small dispersion; with strided patterns, the dot-products for keys matching the stride regularity should span a small range. These conditions are too restrictive, and thus, in general $\delta_r \not\lesssim \delta_s \approx 2\Gamma\Xi^2$ (see figure 4b).

In input-dependent heavy-hitter sparse attention, for any query token $i \in [L]$, we mask all but the highest values of $\mathbf{X}_{:i}^\top \mathbf{W} \mathbf{X}$, and there is a significant gap between the unmasked dot-product $\mathbf{X}_{:i}^\top \mathbf{W} \mathbf{X}_{:j}$ for the unmasked keys $j$ with $M_{ji} = 1$, and the masked dot-product $\mathbf{X}_{:i}^\top \mathbf{W} \mathbf{X}_{:j'}$ for the masked keys $j'$, $M_{j'i} = 0$. Unlike the regular $k$-sparse attention, here each query attends to $k$ keys, but each key can be attended to by anything between 0 and $L$ queries, making the analysis of input-agnostic regular sparse attention (theorem 4) inapplicable. To study these heavy-hitter sparse attentions, we formalize a notion of *semantic separation* between the masked and unmasked keys:

**Definition 3.** *For a sparse attention transformer block with $L$ length input sequences $\mathbf{X} \in \mathbb{R}^{d \times L}$, and an input-dependent heavy-hitter mask $\mathbf{M} \in \{0,1\}^{L \times L}$, we define the per-query semantic separation as a scalar $\Delta > 0$ such that, for any query token $\mathbf{X}_{:i}, i \in [L]$ the minimum difference between the a pair of masked and unmasked query-key dot-products is bounded from below by $\Delta$. That is, for all query tokens $i \in [L]$, with unmasked key $j$ and masked key $j'$, we have*

$$\Delta \leq \min_{\forall j,j' \in [L]: M_{ji}=1, M_{j'i}=0} \left( \mathbf{X}_{:i}^\top \mathbf{W} \mathbf{X}_{:j} - \mathbf{X}_{:i}^\top \mathbf{W} \mathbf{X}_{:j'} \right). \tag{14}$$

The notion of semantic separation is visualized in figure 4d. We present the stability of the heavy-hitter attention in the following (see theorem 9 in appendix G.3):

**Theorem 5.** *Consider the self-attention with a $k$-heavy-hitter input-dependent masking function $m$, applied columnwise to the dot-product matrix to get a mask matrix $\mathbf{M} \in \{0,1\}^{L \times L}$. Assuming the following: (i) For any query-key pairs $\mathbf{X}, \bar{\mathbf{X}} \in \mathbb{R}^{d \times L}$ and parameter $\mathbf{W} \in \mathbb{R}^{d \times d}$, the $k$-heavy-hitter mask $\mathbf{M} = m(\bar{\mathbf{X}}^\top \mathbf{W} \mathbf{X})$ (applied columnwise) has a minimum per-query semantic separation of $\Delta_h$, (ii) A maximum of $\beta k, \beta > 1$ query tokens attend to a single key token, (iii) The per-token Euclidean norms are bounded as $\|\mathbf{X}_{:i}\| \leq \Xi \, \forall i \in [L]$, and the parameter norms are bounded at $\|\mathbf{W}\| \leq \Gamma$, $\|\mathbf{V}\| \leq \Upsilon$, and (iv) The per-query semantic dispersion is bounded by $\delta_h$. Then the masked* softmax *is $\xi_h$-stable with $\xi_h = \left( e^{\delta_h}/k \right) \left( 1 + 1/\Delta_h \right)$, and the sparse attention is stable as in definition 1 with*

$$\lambda_X(\xi_h) = \xi_h \Upsilon k \left( 2\Gamma\Xi^2(\beta+1) + \frac{\beta}{1 + 1/\Delta_h} \right) = e^{\delta_h} \Upsilon \left( \beta + 2\Gamma\Xi^2(\beta+1)(1 + 1/\Delta_h) \right),$$

$$\lambda_W(\xi_h) = 2\xi_h \Upsilon L k\Xi^3 = 2e^{\delta_h}\Upsilon L\Xi^3(1 + 1/\Delta_h), \quad \lambda_V = L\Xi. \tag{15}$$

With the heavy-hitter attention, we would expect the per-query dispersion $\delta_h$ to be significantly smaller than $\delta_s$ especially for small $k$. In appendix G.4, we explicitly characterize the conditions under which the stability constants for input-dependent sparse attention (theorem 5) show improvements over full attention (theorem 3). We see that moderate reduction in the dispersion ($\delta_h$ vs $\delta_s$) allow for significant improvements in $\lambda_W$ even for small separation $\Delta_h$, while improvements in $\lambda_X$ are more moderate.

To see how these stability constants affect the loss landscapes, we also visualize them in figure 5 (top and middle rows) utilizing the techniques proposed in Li et al. [37] (see appendix G.5). We see that the contours on the loss surfaces of full attention model are somewhat asymmetric – see for example, around the center in figure 5b, figure 5c, and moderately in figure 5a. In contrast, the loss surfaces of the heavy-hitter top-$k$ attention model

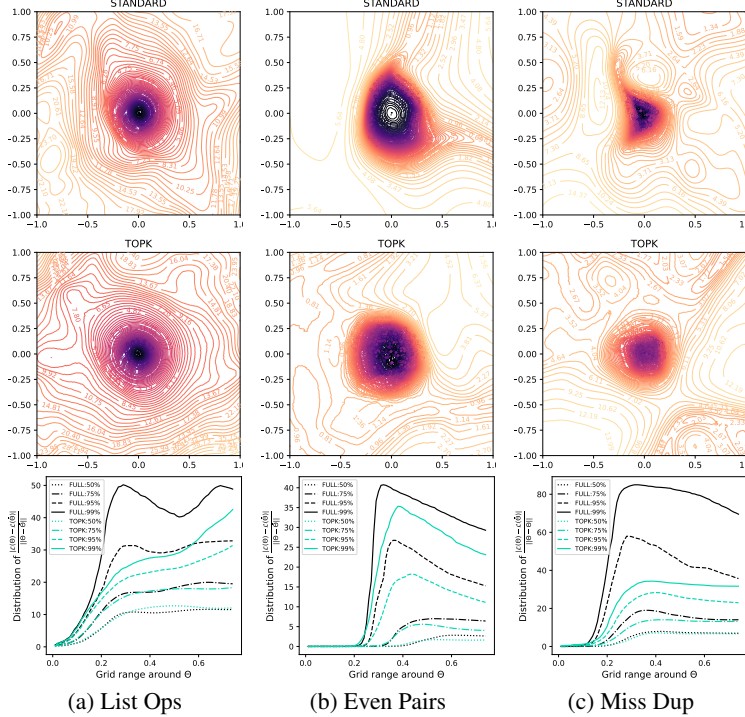

(a) List Ops     (b) Even Pairs     (c) Miss Dup

Figure 5: **Top and middle rows**: Loss surfaces of the models with full attention (top row) and top-$k$ attention (middle row) for the tasks considered in figure 7 with the corresponding hyperparameters utilizing the filter-normalized version of the loss landscape visualization. The (0,0) grid point corresponds to the final trained model – the optimum. **Bottom row**: Distribution of the estimated Lipschitz constants computed in the random directions used to generate the loss landscapes. We report the distributions on the vertical axis in terms of the 50-th (dotted), 75-th (dash-dotted), 95-th (dashed) and 99-th (solid) percentiles (lower is better). On the horizontal axis, we denote the distance of the parameters from the optimum on the grid, and visualize how the distributions vary with the distance.

are quite symmetric, especially around the origin, which corresponds to the final learned model.

We also utilize the loss surface to approximately estimate the Lipschitz constant across the loss landscape (see details in appendix G.5). We plot the distribution of these estimates in the bottom row of figure 5 for varying distance from the optimum. We see that near the optimum (the final trained model), the distributions of these estimates are close for both the models. However, as we move farther away from the trained model, the distributions change significantly, and top-$k$ attention provides a smaller Lipschitz constant estimate compared to full attention all percentiles of the distribution. This indicates that, empirically, the loss for top-$k$ attention has a more favorable Lipschitz continuity compared to full attention, which in turn implies both faster convergence and better generalization guarantees. Thus, our stability-based theoretical investigation in this section appears to align with our empirical observations in section 3.

## 5 Conclusion

In this paper, we study the potential advantages and drawbacks of sparse attention over standard attention beyond the currently studied computational perspective. Our empirical findings, characterized by our theory, show that (i) input-agnostic sparse attention can in general only provide computational benefits, but (ii) input-dependent heavy-hitter sparse attention can provide significant improvements over full attention in terms of learning convergence and generalization. We hope that this motivates further use of heavy-hitter sparse attention at scale with transformer based models. We discuss the limitations of our work in appendix A.

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

# Appendix

## Table of Contents

# A   Discussion of Limitations

Our results imply that there is value in pursuing input-dependent sparse attention [25, 24, 15] in real world LLMs given that they would be both computationally cheaper while having improved generalization guarantees. However, we would like to list some limitations of our work:

(1) Our empirical results are limited to benchmarks developed to study transformers under a controlled setup, and do not speak of their capabilities (in terms of improved training speed, equivalent expressivity and improved generalization) in the wild as we are unable to perform such experiments at scale. The potential advantages of this input-dependent sparse attention at scale remains an open question, though our theoretical results and accompanying preliminary experiments provide a strong motivation.

(2) We study transformers in a supervised learning setup with an encoder-only architecture, where the models are trained from scratch. We do not consider the effect of pretraining, which has been shown to be quite useful with transformers [38], and we do not cover how our results would transfer to a sequence-to-sequence learning setup with an encoder-decoder architecture (though the now common decoder-only architecture can be easily analyzed in our framework).

(3) In our empirical evaluations, we have considered a few representative input-dependent and input-agnostic sparse attention to validate our theoretical results. However, there are various other sparse attention mechanisms [3] that we have not considered in our empirical evaluations.

(4) Our analyses establish upper bounds for the worst case performance (convergence rate or generalization error) for various forms of full and sparse attention, and we compare these upper bounds in our discussion to understand relative behavior. We do support our discussion with empirical evaluations. Furthermore, our study is focused on in-distribution generalization, and does not consider the commonly studied problem of length generalization.

(5) As with any theoretical analysis involving neural networks, we acknowledge that there might a gap between the theoretical constants (such as Lipschitz constant or weight norm upper bounds) we utilize and the practical estimates of those constants empirically seen with these models. However, much of our analysis is *adaptive* in nature, where an improved value of such a constant can be directly incorporated for improved guarantees.

# B   Table of Symbols

Table 1: Problem and transformer model specific symbols discussed in section 2.

| Symbol | Meaning |
| --- | --- |
| $X$ | Input string of token indices |
| $f$ | Ground truth function |
| $y$ | Output $f(X)$ |
| $\mathcal{V}$ | Vocabulary |
| $D$ | Number of tokens in the vocab |
| $L$ | Sequence length |
| $\mathbf{X}$ | Sequence embeddings |
| $\mathbf{W} = \mathbf{Q}^{\top}\mathbf{K}$ | Query-key projection matrix |
| $\mathbf{V}$ | Value projection matrix |
| $\mathbf{P}$ | MLP first layer weights |
| $\mathbf{R}$ | MLP second layer weights |
| LN | Layer normalization |
| $\sigma$ | MLP activation |
| $\mathbf{M}$ | Mask matrix |
| $\mathbf{T}$ | Initial token embeddings |
| $\mathbf{E}$ | Positional embeddings |
| $\boldsymbol{\omega}$ | Token projection vector |
| $\boldsymbol{\Phi}$ | Readout layer weights |
| $\theta^{(t)}$ | $t$-th transformer block parameters |
| $\Theta$ | Full model parameters |
| $f_{\Theta}$ | Learned model with parameters $\Theta$ |
| $k$ | Number of unmasked keys per query in sparse attention |

Table 2: Analysis specific symbols discussed in the theoretical analysis in section 4.

| Symbol | Meaning |
| --- | --- |
| $\xi$ | (Sparse) softmax input stability |
| $\lambda_X(\xi)$ | Input stability constant for self-attention |
| $\lambda_W(\xi)$ | Stability constant w.r.t. to parameter $\mathbf{W}$ for self-attention |
| $\lambda_V$ | Stability constant w.r.t. parameter $\mathbf{V}$ for self-attention |
| $\lambda_\theta(\xi)$ | Per-transformer-block parameter stability |
| $\lambda_{\mathbf{X}}(\xi)$ | Per-transformer-block input stability |
| $\lambda_{\mathcal{L}}(\xi)$ | Learning objective Lipschitz constant |
| $\delta_s$ | Per-query maximum semantic dispersion for standard softmax attention |
| $\delta_r$ | Per-query maximum semantic dispersion for regular input-agnostic $k$-sparse attention |
| $\delta_h$ | Per-query maximum semantic dispersion for heavy-hitter $k$-sparse attention |
| $\Delta_h$ | Per-query minimum semantic separation for heavy-hitter $k$-sparse attention |
| $\beta k$ | The maximum number of queries that attend to a specific key |
| $\beta$ | The "sink ratio" |
| $\Xi$ | Per token embedding Euclidean norm upper bound |
| $\Gamma$ | Maximum spectral norm of the query-key projection matrix $\mathbf{W}$ |
| $\Upsilon$ | Maximum spectral norm of the value projection matrix $\mathbf{V}$ |

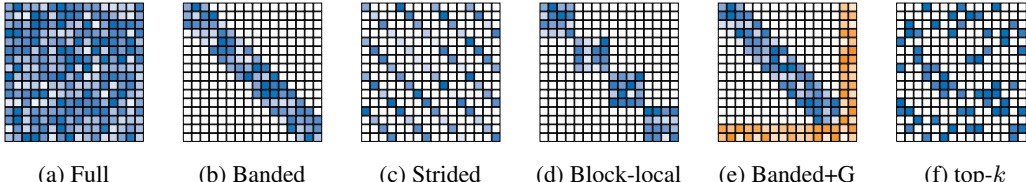

| (a) Full | (b) Banded | (c) Strided | (d) Block-local | (e) Banded+G | (f) top-$k$ |

Figure 6: Visualizations of dot-product based attention scores matrices, which along with the value matrix $\mathbf{VX}$, gives us the attention-based token updates $\mathsf{A}(\mathbf{X})$ (see equation (1) in section 2). The horizontal axis denotes keys and the vertical axis queries. The color intensities denote the value of the attention scores (higher intensities denote higher scores), and the white entries in the matrices corresponds to masked entries. Figure 6a depicts standard full attention score matrix; figure 6b, figure 6c and figure 6d depict various input-agnostic sparse attention score matrices. Figure 6e shows the use of global tokens (attention scores are shown in orange) in conjunction with banded attention (scores are shown in blue), with the last two tokens being the global tokens – all tokens attend to and are attended by these global tokens. Note that the per-query semantic dispersion (see definition 2, figure 4) of the unmasked attention scores in the input-agnostic masks would be similar *in general* to that of standard attention. Input-dependent masked attention such as top-$k$ attention (shown in figure 6f) can have a much smaller semantic dispersion compared to standard attention.

# C   Related Work

In this section, we cover literature on efficient transformers, and the theoretical and empirical investigations on the capabilities and limitations of transformers. Finally, we will also briefly discuss the existing research on optimization with transformers.

**Efficient transformers with sparse attention.** The transformer architecture [1] has had tremendous impact in various fields such as language modeling, vision and tabular data, and spurred new research into the development of architectural variants or X-formers [3, 39, 19]. Many of these have been developed to address the quadratic computational complexity of the attention mechanism in a transformer block with respect to the context length (the number of tokens in the context), with the goal of increasing the context length. One common technique is to sparsify the attention mechanism. Usually each (query) token in the context attends to all other (key) tokens as in figure 6a, leading to the quadratic cost. Instead, we can limit the set of key tokens attended to by any particular query token. Input-agnostic sparsification strategies include attending (i) within a window as in figure 6b [20] or a block as in figure 6d [21], (ii) in a strided manner as in figure 6c [22, 23], (iii) to random tokens [14], or (iv) to only a small number of *global tokens* and these global tokens attend to all other tokens [13, 14]; this is often used in conjunction with other forms of sparse attention as shown in figure 6e. Input-dependent sparsification strategies include (i) using a scoring mechanism and attending only to the highest scoring tokens as in figure 6f [40, 15], or (ii) clustering [24] or hashing [25] tokens into buckets and attending only to in-bucket tokens. Surveys such as Tay et al. [3] and Lin et al. [19] cover various other forms. These input-dependent sparse attention mechanisms focus the attention on the keys corresponding to the highest dot-product scores – the *heavy hitters* – while explicitly ignoring the remaining keys. Sparse attention is considered in all these cases as a way to speed up the attention mechanism in the transformer block during the forward pass without significantly deteriorating the downstream performance, with the standard full attention being the gold-standard. The *Long-range Arena* or LRA [4] serves as one such benchmark comparing different efficient transformers to the standard transformer.

In contrast to above, we theoretically study the effect of sparse attention based transformers on the *learning or empirical risk minimization (ERM) convergence* of the whole model (containing multiple transformer blocks), and the *in-distribution* generalization of the model obtained via ERM. We attempt to characterize conditions under which sparse attention might show improvements over full attention.

**Empirical evaluations of transformer capabilities.** While benchmarks such as the LRA [4] focus on the efficiency and in-distribution generalization, transformers have also been thoroughly evaluated on benchmarks studying specific forms of *out-of-distribution generalization* such as compositional generalization and length generalization. Compositional generalization benchmarks

such as COGS [27] and SCAN [26] consider sequence-to-sequence translation problems, and they have been used to highlight the inability of transformers to systematically generalize [41]. However, subsequent work such as Csordás et al. [42], Ontanon et al. [43] have demonstrated ways in which transformers can systematically generalize. The Neural Networks and Chomsky Hierarchy or NNCH benchmark [11] considers language transduction tasks from different formal language classes such as regular, deterministic context-free and context-sensitive languages. This benchmark studies the ability of various models (including transformers) to length generalize – that is, generalize to longer input sequences when being trained in a length limited manner. There has also been a lot of research on improving the performance of transformer based models on these out-of-distribution generalization benchmarks leveraging auxiliary tasks [44] and chain-of-thought prompting [45].

In our work, we focus on the theoretical analysis of the ERM convergence and the in-distribution generalization of models based on multiple transformer blocks, and empirically validate our theoretical insights utilizing these above benchmarks. We consider one multiclass classification task from the LRA benchmark [4] and a subset of the tasks from the NNCH benchmark [11] that can be posed as supervised classification problems.

**Theoretical treatment of transformer capabilities.** Given the widespread success of transformers, there have been various theoretical studies on the capabilities and limitations of transformers. One line of research focuses on the ability of transformers to express (and thus recognize) formal languages [28]. Some of these works study transformers with hard attention [46–49], while others consider the more commonly used softmax attention [50, 51]. Another line of research has focused on understanding the capabilities of transformers as algorithms [36], demonstrating how transformers can, under specific parameter settings, perform *in-context* gradient descent for linear regression [52] or in-context clustering [53], and how easily can such parameters can be found [36, 54, 55]. Yun et al. [12] focus on universal approximation of sparse attention transformer for sequence-to-sequence problems, and establish conditions on the sparsity pattern that ensure desired expressivity given enough number of transformer layers.

Viewing hard-attention as a form of input-dependent sparse attention, these existing expressivity results [28] are complementary to our focus on learning convergence and in-distribution generalization for models using multiple sparse attention based transformer blocks – existing hard-attention expressivity results discuss whether sparse attention transformers are expressive enough for the task at hand. Our study here focuses on how quickly and sample efficiently can such transformers learn the task, and how the attention sparsity pattern plays a role.

**Optimization with transformers.** There has been a lot of work on understanding the optimization of transformers in terms of the benefit of adaptive methods such as Adam over non-adaptive SGD [56–60]. However, the focus there is to understand why optimizers such as Adam converge significantly faster than SGD with transformer models; no such consistent difference has been established for previous architectures such as convolutional or residual. Li et al. [61] recently present an analysis of the training dynamics with SignGD for a single transformer block model for a specific noisy binary classification problem, working in the "feature learning framework", and empirically demonstrating that the dynamics of SignGD and Adam are quite similar, thus making SignGD a useful proxy for analyzing Adam.

Our study is complementary to this line of work where we study the effect of sparsity in attention to non-adaptive SGD convergence and generalization. We also consider a more general sequence learning problem with multiple transformer blocks.

# D    Details on Experimental Setup

The code and results for the paper are available in this GitHub repository.

**Tasks.** We consider the List Operations or ListOps task [62] from the LRA benchmark [4] with sequence lengths between 500 and 600 both for training and testing because we are evaluating in-distribution learning and generalization. This is a 10-class classification problem. We select this task over the other tasks in the LRA benchmark because (i) this is a task where transformers have better than random performance (around 30-40% compared to a random 10% performance), but there is still a significant room for improvement, and (ii) we can control the length of the input sequences and still have a meaningful problem, which is not as straightforward with the other document or image processing tasks in LRA. From the NNCH benchmark [11], we consider 3 tasks that can be solved as a binary classification problem – Parity, Even Pairs, and Missing Duplicates, and 4 tasks that can be solved as a multi-class classification problem – Cycle Navigation, Stack Manipulation, Modular Arithmetic with Brackets and Solve Equation. Parity, Even Pairs and Cycle Navigation are regular languages. Stack Manipulation, Modular Arithmetic and Solve Equation are deterministic context-free languages, while Missing Duplicates is a context-sensitive language. For the NNCH tasks, we consider input sequences of length 40 both for training and testing; Deletang et al. [11] train on the same length but test on longer to evaluate out-of-distribution length generalization. For all the tasks, we utilize a training / holdout sets of sizes 5000 / 2000.

**Sparse attention.** While there are various sparse attention mechanisms (as we discussed in appendix C), we will consider a representative subset for our empirical evaluations. For input-agnostic sparse attention, we choose banded attention (figure 6b [20]) and block-local attention (figure 6d [21]), with varying band and block sizes respectively. For input-dependent heavy-hitter sparse attention, we choose top-$k$ attention (figure 6f [15]). The main motivation for selecting top-$k$ over LSH based [25] or clustering based [24] input-dependent sparse attention is that we can then easily ensure that the input-dependent sparse attention attends to exactly the same number of tokens as in the input-agnostic ones – that is, the number of nonzeros in each column of the attention score matrix is exactly the same across all sparse attention patterns we consider. We also consider versions of these input-agnostic sparse attention with varying number of global tokens (figure 6e). Note that, as we have highlighted before, **the number of learnable parameters is exactly the same between the model using standard full attention and the one using sparse attention**. A *minor difference* is with global tokens where we also learn their initial global token embeddings. For this reason, we use *exactly the same hyperparameters* for the full and sparse attention versions of the same model to ablate the effect of the sparse attention.

**Compute resources and experimental setup.** All our empirical evaluations are performed on a Intel i7 Core CPU (16 threads, 64GB memory), and a Nvidia V100 GPU (8GB memory). Each experiment was executed with 10 random seeds and all results are aggregated across these 10 trials. Each trial took around 55 hours – ListOps: 21.5, Parity: 10, Missing Duplicates: 2.5, Even Pairs: 1, Stack Manipulation: 2, Modular Arithmetic: 6, Solve Equation: 6, Cycle Navigation: 7.5 – for a total of 550 hours for each of the 3 activation functions considered. Ablation of additional hyperparameters took another 160 hours. The implementation is in Pytorch 2.2 with CUDA 12.4. We implement our own attention block to handle different forms of sparse attention.

**Hyperparameters.** For the NNCH tasks, we considered the transformer architecture used in Deletang et al. [11] with (i) $T = 5$ transformer blocks, (ii) embedding dimension $d = 64$ and (iii) the MLP hidden layer $d_{\mathsf{MLP}} = 64$, but with a single head (instead of 8) and a dropout of 0.01. The final classification layer uses the average of all the token representations after the final transformer block. For the ListOps task, we utilize the same architecture but use $T = 10$ transformer blocks for the initial experiment. We also consider varying number of heads and blocks in our experiments studying the effect of hyperparameters. For all problems, we use the SGD optimizer and the StepLR learning rate scheduler with a decay rate of 0.99 for ListOps and 0.9995 for NNCH tasks and a decay period of 1 epoch. For the NNCH tasks, we use an initial learning rate of 0.1, while we use 1.0 for ListOps. The number of epochs is selected to ensure that standard full attention transformer is able to consistently achieve 100% training accuracy (and thus, the ERM has converged). Thus, we use 100 epochs for Even Pairs, 200 epochs for ListOps and Stack Manipulation, 250 epochs for Missing Duplicates, 600 epochs for Modular Arithmetic and Solve Equation, 750 epochs for Cycle Navigation, and 1000 for Parity.

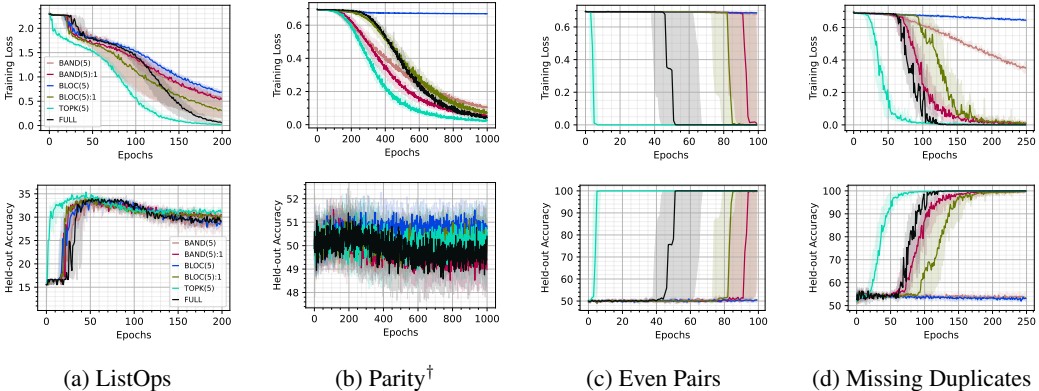

(a) ListOps        (b) Parity†        (c) Even Pairs        (d) Missing Duplicates

Figure 7: Learning convergence and generalization curves for full attention and various sparse attention based models. Each column corresponds to a task; we present 4 tasks here and 4 more in figure 8. *The legend is the same across all datasets* – BAND(5) denotes banded attention (figure 6b) with a band size of 5; BAND(5):1 denotes the same with a single global token (figure 6e). BLOC(5) denotes block local attention (figure 6d) with a block size of 5; BLOC(5):1 denotes the same with single global token. TOPK(5) is top-$k$ attention with $k = 5$. **Top row**: Training cross-entropy loss trajectories – lower is better. **Bottom row**: Generalization performance on held-out set as training progresses – higher is better. Further results with different mask sizes and different number of global tokens is presented in figure 9 (training cross-entropy), figure 10 (training accuracy), table 3 (generalization) and table 4 (convergence). † For the Parity task, all forms of attention have poor generalization, with a held-out accuracy as low as random guessing (50% for binary classification).

## E  Additional Empirical Results

### E.1  Detailed Evaluation

In this subsection, we present a detailed view of the results presented in figure 7 and figure 8, where we evaluate different mask sizes (number of nonzeros in each column of the attention matrix) and the number of global tokens included with the input-agnostic sparse attention patterns. We present the trajectories of the training cross-entropy loss in figure 9 and figure 11, and the trajectories of the training accuracies in figure 10 and figure 12. In table 3, we present the best accuracy on the held-out set for each of the sparse attention patterns and contrast it with that of the full attention model. Table 4 presents the number of epochs (aggregated over the 10 repetitions of each experiments) required by each attention pattern to (i) achieve at least 95% training accuracy for the first time (if at all), and (ii) achieve the best held-out accuracy.

The results in figure 9 and figure 11 (along with figure 10 and figure 12) show that the input-agnostic sparse attention has a slower ERM convergence than standard full attention, being unable to reach even 95% training accuracy with the ListOps and Even Pairs tasks. With the input-agnostic sparse attention, having the global tokens helps convergence in almost all cases, being critical for convergence in the NNCH binary classification tasks (Parity, Even Pairs and Missing Duplicates), especially with the block local attention. In contrast, the ERM convergence of the top-$k$ attention is significantly improved over the standard full attention in all 8 tasks, with improvements (in terms of achieving 95% training accuracy) over standard attention ranging between $1.37\times$ (121 epochs vs 167 epochs) with ListOps to $9.5\times$ (6 epochs vs 53 epochs) with Even Pairs (see table 4 for further results on this).

The results in table 3 show that, in almost all cases, the input-dependent sparse attention has similar (Even Pairs and Missing Duplicates) or better (ListOps) holdout accuracy than the standard full attention. This is true both in terms of the highest holdout accuracy during the training trajectory, and the final holdout accuracy. The latter highlights that the faster ERM convergence of input-dependent sparse attention does not lead to overfitting. In fact, with the ListOps task, the final holdout accuracy with standard attention drops from around $35.1 \pm 0.6\%$ to $28.9 \pm 1.4\%$, while the drop with top-$k$ attention is only from $36.3 \pm 0.3\%$ to $31.3 \pm 0.9\%$. In general, the top-$k$ attention based transformers also have comparitively similar or lower variations in their performance. This set of results align with our theoretical result that the improved stability of the input-dependent sparse attention translates

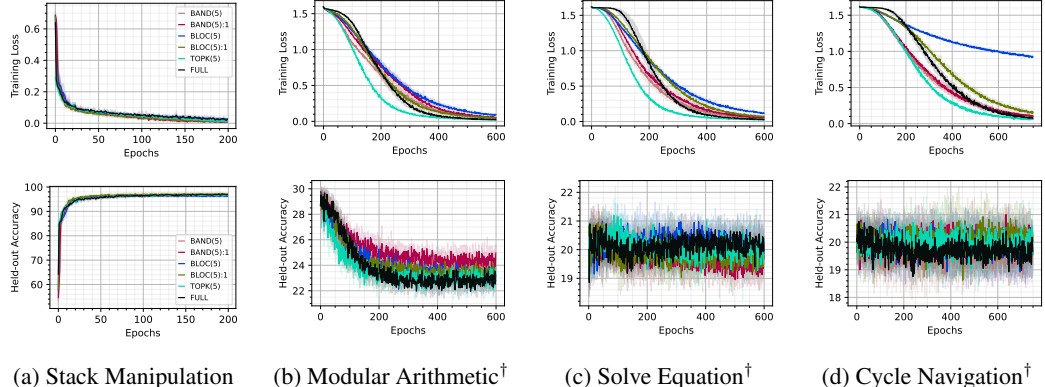

| (a) Stack Manipulation | (b) Modular Arithmetic$^\dagger$ | (c) Solve Equation$^\dagger$ | (d) Cycle Navigation$^\dagger$ |

Figure 8: Same as figure 7 with 4 more NNCH tasks. Further results with different mask sizes and different number of global tokens is presented in figure 11 (training cross-entropy) and figure 12 (training accuracy). $^\dagger$ For the Modular Arithmetic, Solve Equation and Cycle Navigation tasks, all forms of attention have poor generalization, with a held-out accuracy as low as random guessing (20% for each of these 5-class classification tasks).

Table 3: Generalization performance (higher is better) for standard full attention (highlighted in green) and sparse attention. We report the mean$_{\pm\text{std}}$ aggregated over the 10 trials (same as figure 9 and figure 10). The first set of columns show the best holdout accuracy obtained across the training trajectory, while the second set show the holdout accuracy at the end of training. We highlight methods that have not reached 95% training accuracy at the end of training in blue; among the remaining methods, the highest mean in each column is shown in **bold**. See figure 9 for the naming of the attention mechanisms.

| Attention | Best holdout accuracy | | | Final holdout accuracy | | |
|---|---|---|---|---|---|---|
| | ListOps | MissingDups | EvenPairs | ListOps | MissingDups | EvenPairs |
| Standard | $35.09_{\pm 0.60}$ | $\mathbf{100.00}_{\pm 0.00}$ | $\mathbf{100.00}_{\pm 0.00}$ | $28.92_{\pm 1.40}$ | $\mathbf{99.98}_{\pm 0.02}$ | $\mathbf{100.00}_{\pm 0.00}$ |
| Banded(5) | $34.47_{\pm 0.55}$ | $58.35_{\pm 1.34}$ | $52.57_{\pm 0.96}$ | $28.25_{\pm 1.70}$ | $54.04_{\pm 1.83}$ | $50.34_{\pm 1.42}$ |
| Banded(9) | $34.82_{\pm 0.51}$ | $96.53_{\pm 1.99}$ | $52.06_{\pm 1.07}$ | $28.40_{\pm 1.09}$ | $95.35_{\pm 2.19}$ | $50.42_{\pm 1.08}$ |
| Banded(5)+G1 | $34.73_{\pm 0.43}$ | $99.78_{\pm 0.34}$ | $81.86_{\pm 22.36}$ | $30.50_{\pm 0.58}$ | $99.62_{\pm 0.35}$ | $81.40_{\pm 22.88}$ |
| Banded(9)+G1 | $35.54_{\pm 0.53}$ | $99.81_{\pm 0.13}$ | $64.47_{\pm 19.70}$ | $31.05_{\pm 2.16}$ | $99.40_{\pm 0.38}$ | $63.07_{\pm 20.59}$ |
| Banded(5)+G3 | $35.23_{\pm 0.52}$ | $99.92_{\pm 0.12}$ | $75.94_{\pm 24.06}$ | $30.99_{\pm 1.15}$ | $99.80_{\pm 0.30}$ | $75.48_{\pm 24.52}$ |
| Banded(9)+G3 | $35.29_{\pm 0.60}$ | $99.90_{\pm 0.10}$ | $66.20_{\pm 22.13}$ | $31.80_{\pm 1.41}$ | $99.41_{\pm 0.48}$ | $65.61_{\pm 22.52}$ |
| Blklocal(5) | $34.83_{\pm 0.42}$ | $57.87_{\pm 1.16}$ | $51.98_{\pm 0.98}$ | $29.06_{\pm 1.33}$ | $52.90_{\pm 1.12}$ | $50.46_{\pm 0.81}$ |
| Blklocal(9) | $34.73_{\pm 0.25}$ | $58.12_{\pm 1.21}$ | $51.79_{\pm 0.84}$ | $28.59_{\pm 1.21}$ | $52.50_{\pm 0.71}$ | $50.28_{\pm 1.00}$ |
| Blklocal(5)+G1 | $35.20_{\pm 0.55}$ | $98.78_{\pm 3.28}$ | $85.62_{\pm 21.89}$ | $29.73_{\pm 1.63}$ | $98.47_{\pm 3.75}$ | $85.10_{\pm 22.69}$ |
| Blklocal(9)+G1 | $34.63_{\pm 0.43}$ | $99.02_{\pm 0.74}$ | $71.13_{\pm 23.56}$ | $30.57_{\pm 1.07}$ | $98.58_{\pm 0.81}$ | $70.33_{\pm 24.22}$ |
| Blklocal(5)+G3 | $35.53_{\pm 0.66}$ | $99.96_{\pm 0.09}$ | $66.55_{\pm 21.93}$ | $29.97_{\pm 1.35}$ | $99.86_{\pm 0.13}$ | $65.97_{\pm 22.33}$ |
| Blklocal(9)+G3 | $35.53_{\pm 0.61}$ | $99.73_{\pm 0.22}$ | $66.34_{\pm 22.05}$ | $31.82_{\pm 1.35}$ | $99.12_{\pm 0.73}$ | $65.45_{\pm 22.63}$ |
| Topk(5) | $36.02_{\pm 0.59}$ | $\mathbf{100.00}_{\pm 0.00}$ | $\mathbf{100.00}_{\pm 0.00}$ | $31.06_{\pm 0.73}$ | $99.94_{\pm 0.07}$ | $\mathbf{100.00}_{\pm 0.00}$ |
| Topk(9) | $\mathbf{36.25}_{\pm 0.29}$ | $\mathbf{100.00}_{\pm 0.00}$ | $\mathbf{100.00}_{\pm 0.00}$ | $\mathbf{31.33}_{\pm 0.85}$ | $99.95_{\pm 0.06}$ | $\mathbf{100.00}_{\pm 0.00}$ |

to matching or better generalization error. This does not hold with the Parity, Modular Arithmetic, Solve Equation and Cycle Navigation tasks. However, note that these are tasks for which al forms of attention have very close to random performance (which is 50% for a balanced binary classification problem and 20% for a 5-class classification problem), and thus none of the attention mechanisms are generalizing well. The inability of the input-agnostic sparse attention to obtain high training accuracy within the training budget translates to low holdout error, especially with the Even Pairs task.

## E.2 Effect of MLP Activation Function

Here, we present a detailed view of the results presented in figure 2, where we evaluate different mask sizes (number of nonzeros in each column of the attention matrix) and the number of global tokens included with the input-agnostic sparse attention patterns. We present the trajectories of the training cross-entropy loss with the GELU activation [31] in figure 13 and figure 17 for all 8 tasks, and their corresponding training accuracy trajectories in figure 14 and figure 18. Similar results for 4/8 tasks –

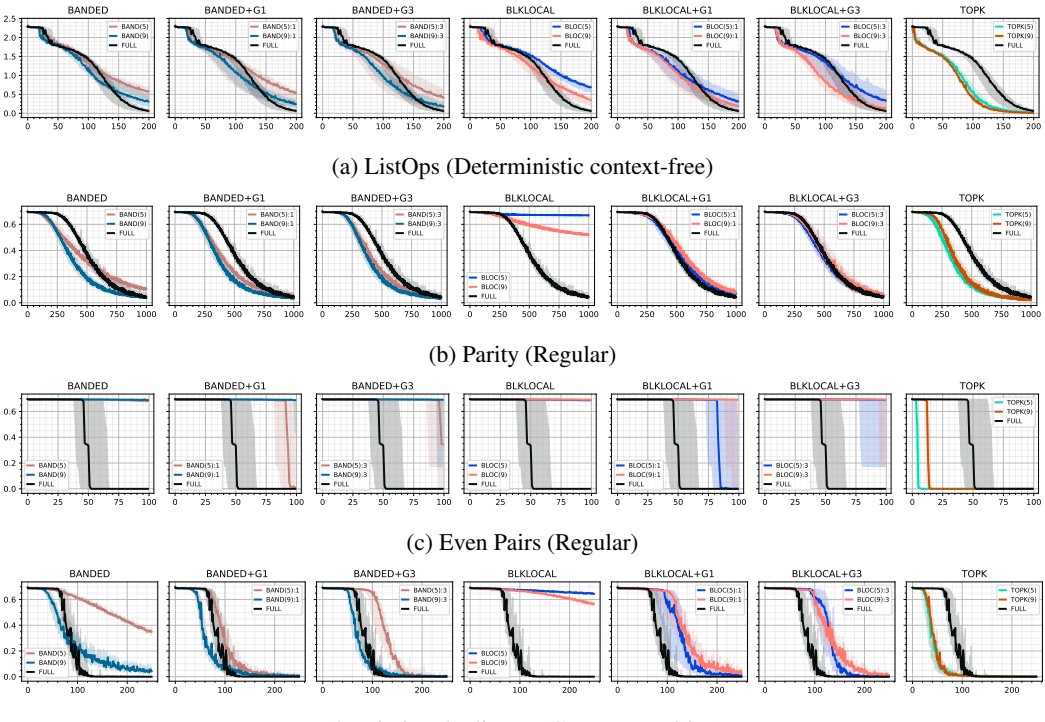

(a) ListOps (Deterministic context-free)

(b) Parity (Regular)

(c) Even Pairs (Regular)

(d) Missing duplicates (Context-sensitive)

Figure 9: Training cross-entropy (vertical axis, lower is better) vs number of epochs (horizontal axis) across different tasks and sparse attention forms aggregated across 10 repetitions. Each plot contains the training curve for the standard transformers (in black). Sparse attention: Banded (column 1), banded with 1 global token (column 2), banded with 3 global tokens (column 3), block-local (column 4), block-local with 1 global token (column 5), block-local with 3 global tokens (column 6), top-$k$ attention (column 7).

namely ListOps, Parity, Even Pairs and Missing Duplicates – with the Mish activation [33] in the MLP block are presented in figure 15 (training cross-entropy) and figure 16 (training accuracy).

In all cases, the qualitative results do not seem the change much from the previous results with the ReLU activation in the MLP block presented in figure 9 and figure 11 (and corresponding figure 10 and figure 12). The overall trend continues to be that (i) input-agnostic sparse attention models continue to train and generalize comparitively to the full attention model, and (ii) input-dependent heavy-hitter sparse attention models continue to converge faster (and generalize similarly or better) than the full attention model.

The input-agnostic sparse attention models continue to converge comparably to full attention with ListOps and Missing Duplicates while falling behind in Even Pairs. One marked difference here is that, with ListOps, the full attention model initially converges slower than the other sparse attention models (compare figure 7a with figure 2a and figure 2d). This is more marked with the Mish activation. However, finally the full attention model convergence catches up to the input-agnostic sparse attention models. This initial slowdown in the convergence is also reflected in the initial lower generalization accuracy. In contrast, the input-dependent heavy-hitter top-$k$ attention continues to consistently converge faster than full attention in terms of the training loss for both these MLP activations, with very little differences from the results with ReLU activation. This form of sparse attention also achieves better generalization performance earlier in the training process. This indicates that the difference is performance is probably due to the differences in the attention mechanism and not an artifact of the MLP block configuration.

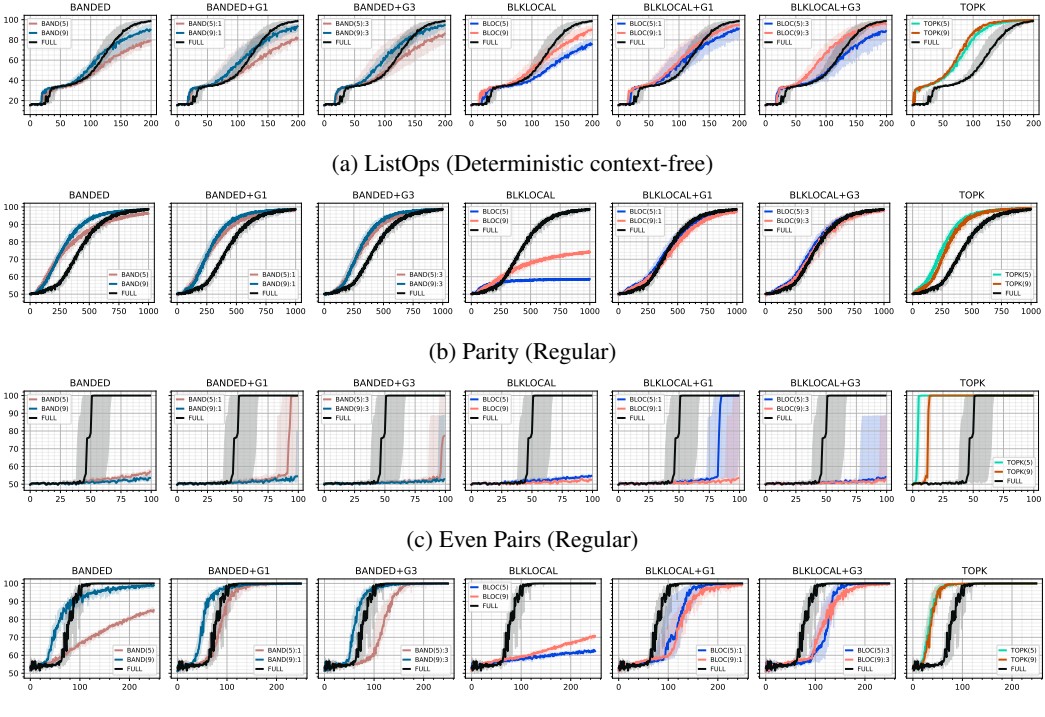

(a) ListOps (Deterministic context-free)

(b) Parity (Regular)

(c) Even Pairs (Regular)

(d) Missing duplicates (Context-sensitive)

Figure 10: Training accuracy (vertical axis – higher is better) vs number of epochs (horizontal axis) across different tasks and sparse attention aggregated across 10 repetitions (median (line) and inter-quartile range (shaded region)). Each plot contains the training curve for the standard full attention transformer (in black). Sparse attention are as follows with each with $k = 5$ and $k = 9$ nonzeros in each row of the attention score matrix – column 1: banded, column 2: banded with 1 global token, column 3: banded with 3 global tokens, column 4: block-local, column 5: block-local with 1 global token, column 6: block-local with 3 global tokens, column 7: top-$k$ attention.

### E.3    Natural Language Processing Evaluations

We consider a preliminary experiment with the Penn Tree Bank [63] natural language dataset where we use the context of tokens to predict the next token. The text is tokenized using the SentencePiece tokenizer [64] with BPE (byte-pair encoding) [65] and a vocabulary size of 4096. We consider a transformer with embedding size of 32 and MLP hidden dimensionality of 128, varying the number of transformer blocks with a single attention head per block. We train the model for 50 epochs with SGD. We consider full attention and top-$k$ attention with $k = 5$ and report the token misclassification cross-entropy on the training set at increasing number of epochs in figure 19.

As the results indicate, the top-$k$ attention mechanism continues to converge faster that full attention even in this NLP task for varying number of transformer blocks. We also consider a more challenging version of Penn Tree Bank with a larger vocabulary of 10000 tokens. Here we also consider a larger transformer model with an embedding dimension of 128 and MLP hidden dimensionality of 512, and vary the number of transformer blocks with 4 attention heads in each block. The training loss curves are presented in figure 20, and demonstrate that the input-dependent sparse attention continues to converge faster than the full attention transformer model.

Table 4: Additional generalization/convergence results: We note the number of iterations required during the training (i) to reach 95% training accuracy, with a '-' denoting that we do not reach that training accuracy, and (ii) to reach the highest holdout accuracy. For training that reach 95% training accuracy, the smallest in each column is highlighted in **bold**.

| Attention | Iterations to 95% training accuracy | | | Iterations to best holdout accuracy | | |
|---|---|---|---|---|---|---|
| | ListOps | MissingDups | EvenPairs | ListOps | MissingDups | EvenPairs |
| Standard | $167_{\pm 18}$ | $96_{\pm 21}$ | $53_{\pm 15}$ | $62_{\pm 18}$ | $\mathbf{174}_{\pm 55}$ | $64_{\pm 21}$ |
| Band(5) | - | - | - | $63_{\pm 19}$ | $38_{\pm 69}$ | $66_{\pm 28}$ |
| Band(9) | - | $114_{\pm 30}$ | - | $68_{\pm 17}$ | $236_{\pm 6}$ | $45_{\pm 35}$ |
| Band(5)+G1 | - | $114_{\pm 13}$ | - | $59_{\pm 23}$ | $236_{\pm 6}$ | $83_{\pm 28}$ |
| Band(9)+G1 | - | $72_{\pm 11}$ | - | $74_{\pm 21}$ | $231_{\pm 12}$ | $51_{\pm 37}$ |
| Band(5)+G3 | - | $137_{\pm 16}$ | - | $62_{\pm 20}$ | $227_{\pm 24}$ | $81_{\pm 27}$ |
| Band(9)+G3 | - | $80_{\pm 9}$ | - | $69_{\pm 24}$ | $224_{\pm 16}$ | $70_{\pm 32}$ |
| Blklocal(5) | - | - | - | $71_{\pm 11}$ | $5_{\pm 4}$ | $67_{\pm 26}$ |
| Blklocal(9) | - | - | - | $57_{\pm 11}$ | $3_{\pm 2}$ | $57_{\pm 26}$ |
| Blklocal(5)+G1 | - | - | - | $53_{\pm 9}$ | $238_{\pm 7}$ | $85_{\pm 18}$ |
| Blklocal(9)+G1 | - | $154_{\pm 26}$ | - | $54_{\pm 15}$ | $235_{\pm 12}$ | $67_{\pm 36}$ |
| Blklocal(5)+G3 | - | $134_{\pm 20}$ | - | $69_{\pm 21}$ | $230_{\pm 19}$ | $86_{\pm 12}$ |
| Blklocal(9)+G3 | - | $146_{\pm 18}$ | - | $60_{\pm 17}$ | $234_{\pm 9}$ | $62_{\pm 36}$ |
| Topk(5) | $131_{\pm 9}$ | $\mathbf{48}_{\pm 9}$ | $\mathbf{6}_{\pm 3}$ | $\mathbf{40}_{\pm 8}$ | $226_{\pm 14}$ | $\mathbf{18}_{\pm 0}$ |
| Topk(9) | $\mathbf{122}_{\pm 8}$ | $\mathbf{48}_{\pm 10}$ | $13_{\pm 3}$ | $41_{\pm 7}$ | $193_{\pm 59}$ | $19_{\pm 0}$ |

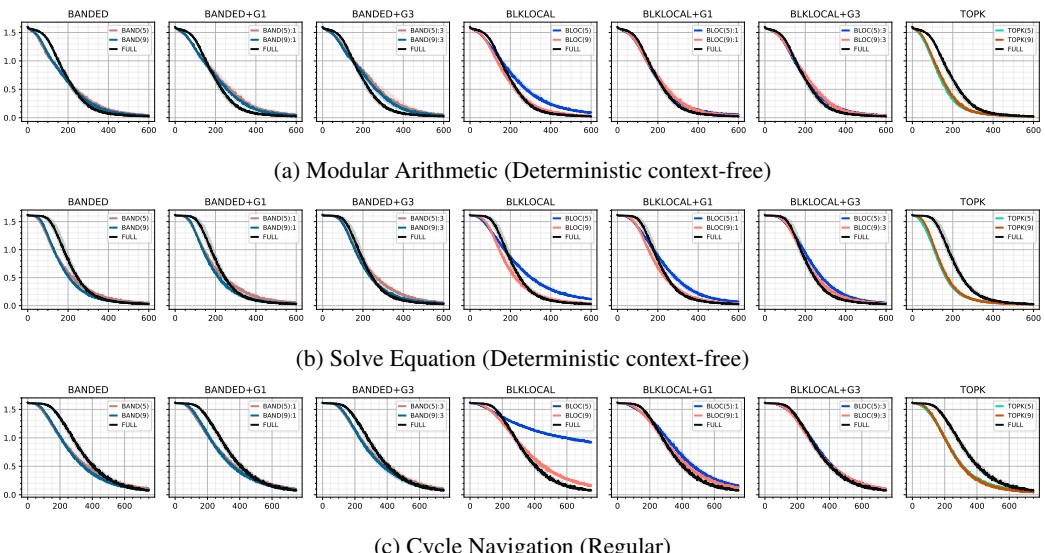

(a) Modular Arithmetic (Deterministic context-free)

(b) Solve Equation (Deterministic context-free)

(c) Cycle Navigation (Regular)

Figure 11: Training cross-entropy (vertical axis, lower is better) vs number of epochs (horizontal axis) across different tasks and sparse attention forms aggregated across 10 repetitions. Each plot contains the training curve for the standard transformers (in black). Sparse attention: Banded (column 1), banded with 1 global token (column 2), banded with 3 global tokens (column 3), block-local (column 4), block-local with 1 global token (column 5), block-local with 3 global tokens (column 6), top-$k$ attention (column 7).

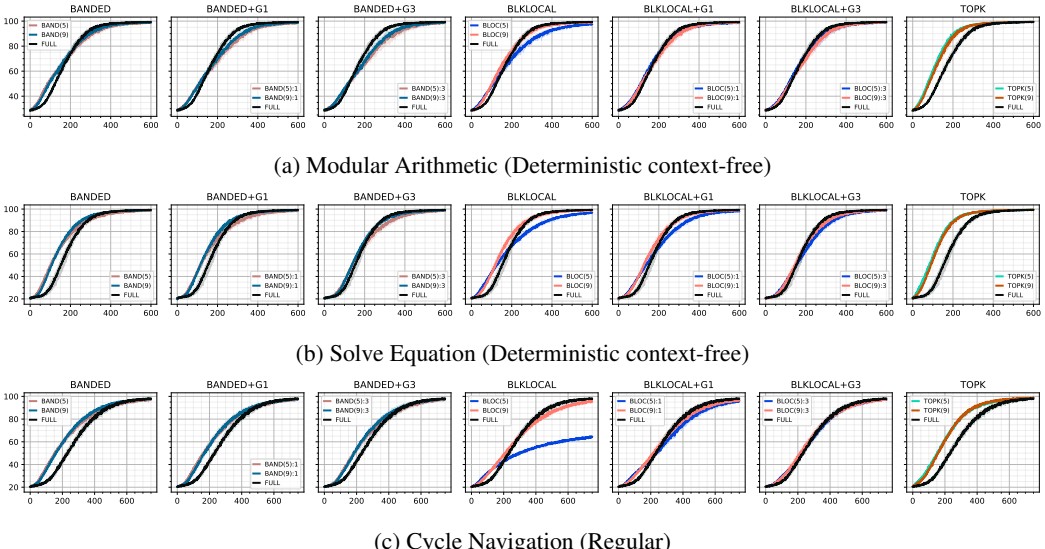

Figure 12: Training accuracy (vertical axis – higher is better) vs number of epochs (horizontal axis) across different tasks and sparse attention aggregated across 10 repetitions (median (line) and inter-quartile range (shaded region)). Each plot contains the training curve for the standard full attention transformer (in black). Sparse attention are as follows with each with $k = 5$ and $k = 9$ nonzeros in each row of the attention score matrix – column 1: banded, column 2: banded with 1 global token, column 3: banded with 3 global tokens, column 4: block-local, column 5: block-local with 1 global token, column 6: block-local with 3 global tokens, column 7: top-$k$ attention.

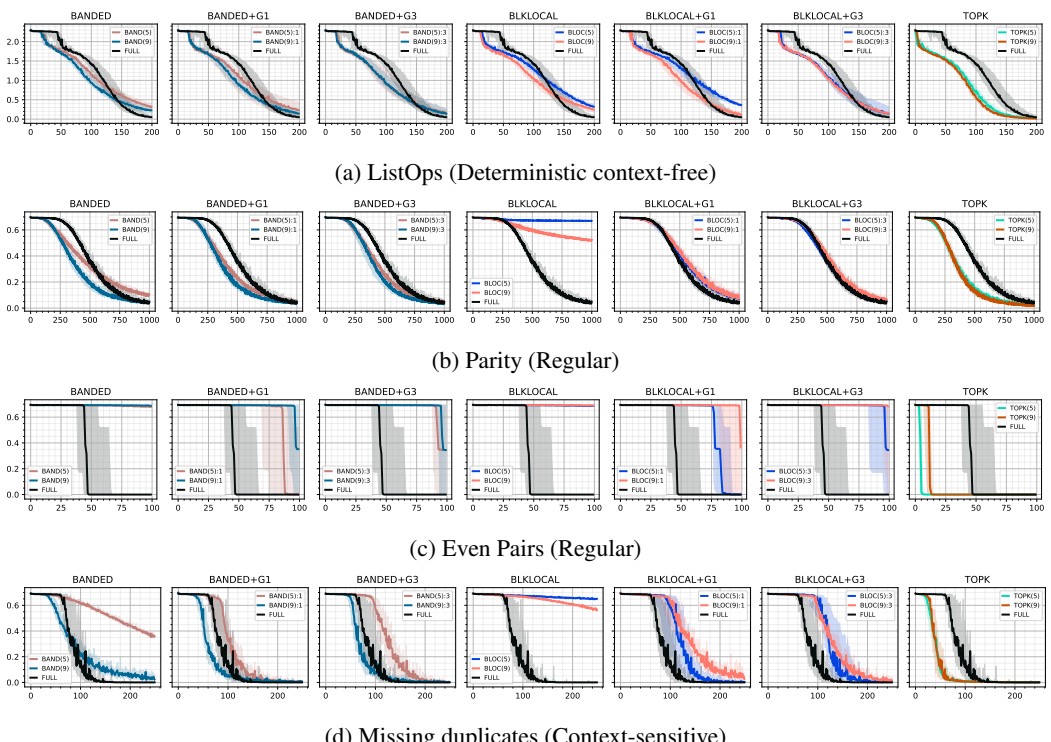

Figure 13: Same as figure 9 with GELU activation in the MLP component of the transformer block.

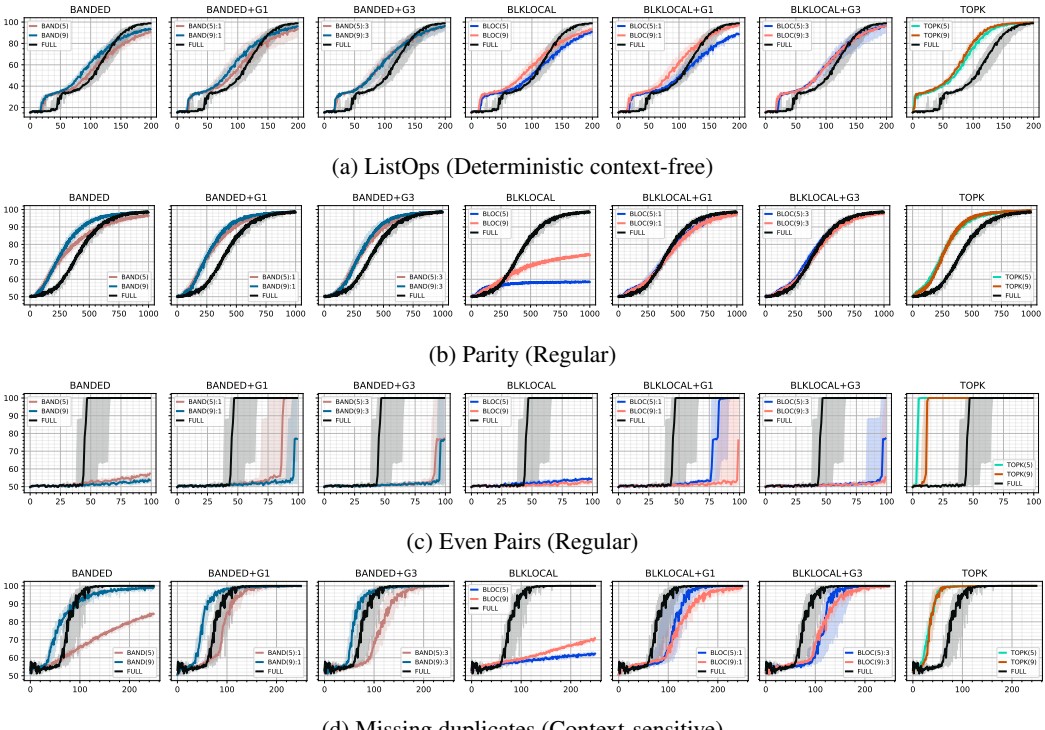

Figure 14: Same as figure 10 with GELU activation in the MLP component of the transformer block.

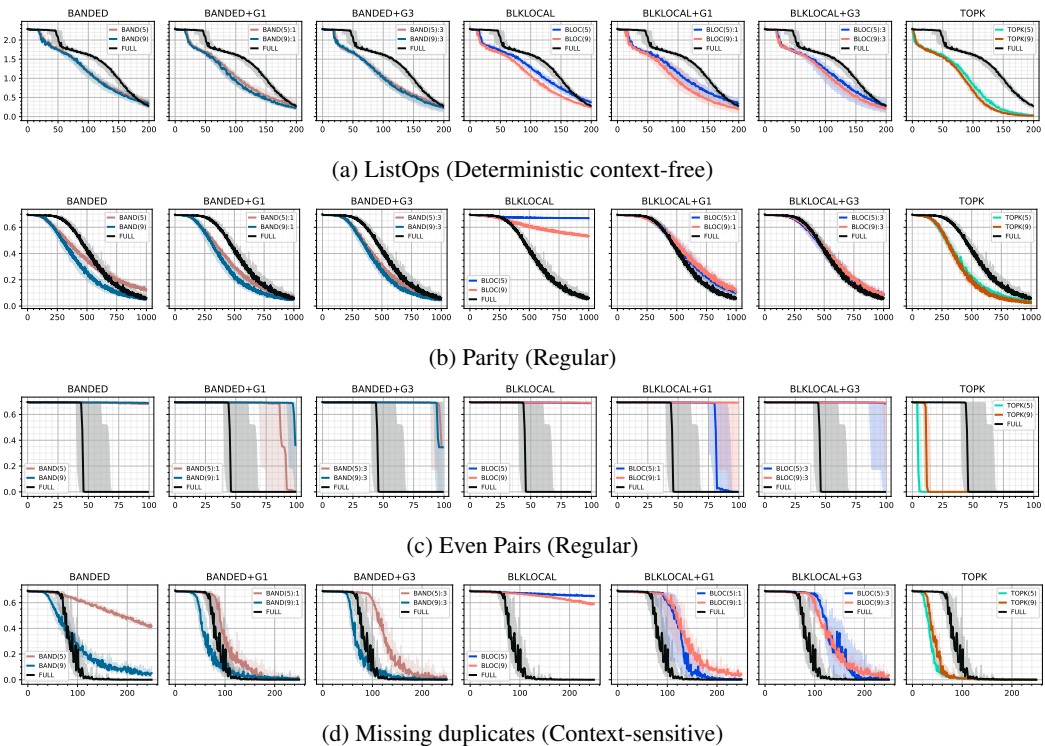

Figure 15: Same as figure 9 with Mish activation in the MLP component of the transformer block.

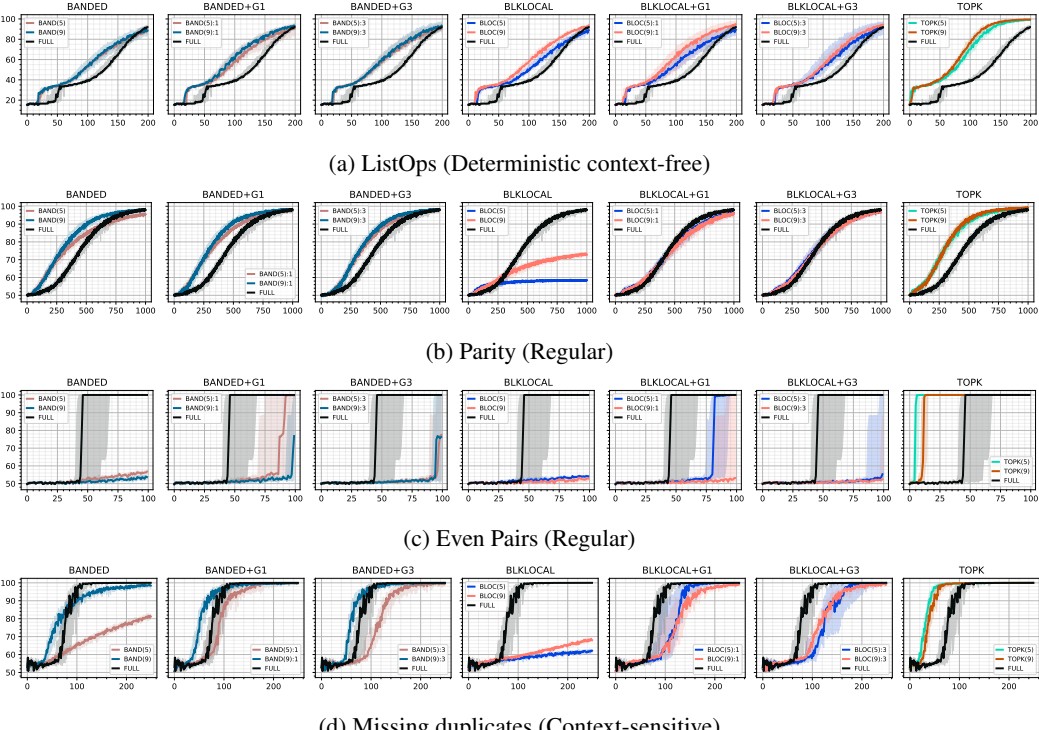

(a) ListOps (Deterministic context-free)

(b) Parity (Regular)

(c) Even Pairs (Regular)

(d) Missing duplicates (Context-sensitive)

Figure 16: Same as figure 10 with Mish activation in the MLP component of the transformer block.

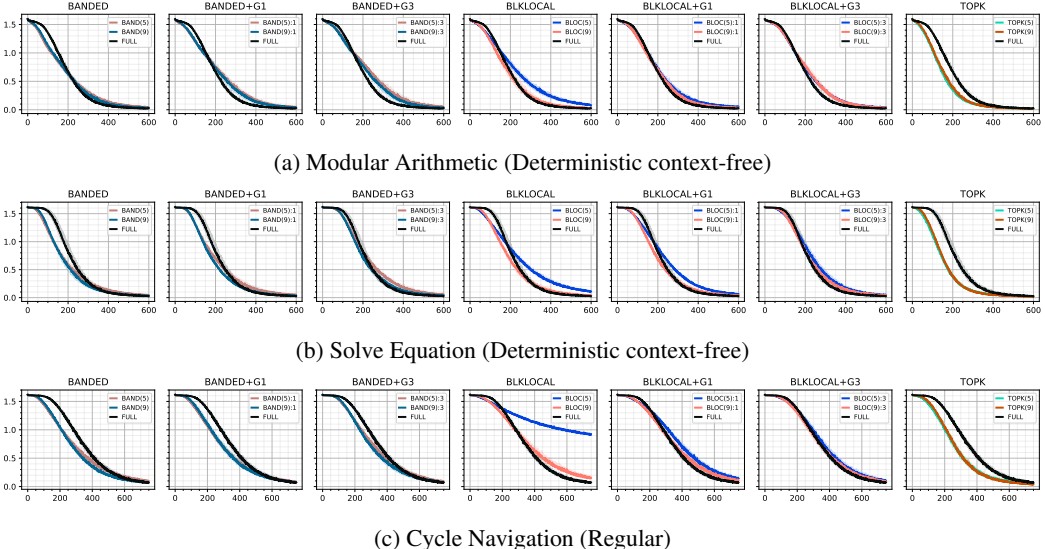

(a) Modular Arithmetic (Deterministic context-free)

(b) Solve Equation (Deterministic context-free)

(c) Cycle Navigation (Regular)

Figure 17: Same as figure 11 with GELU activation in the MLP component of the transformer block.

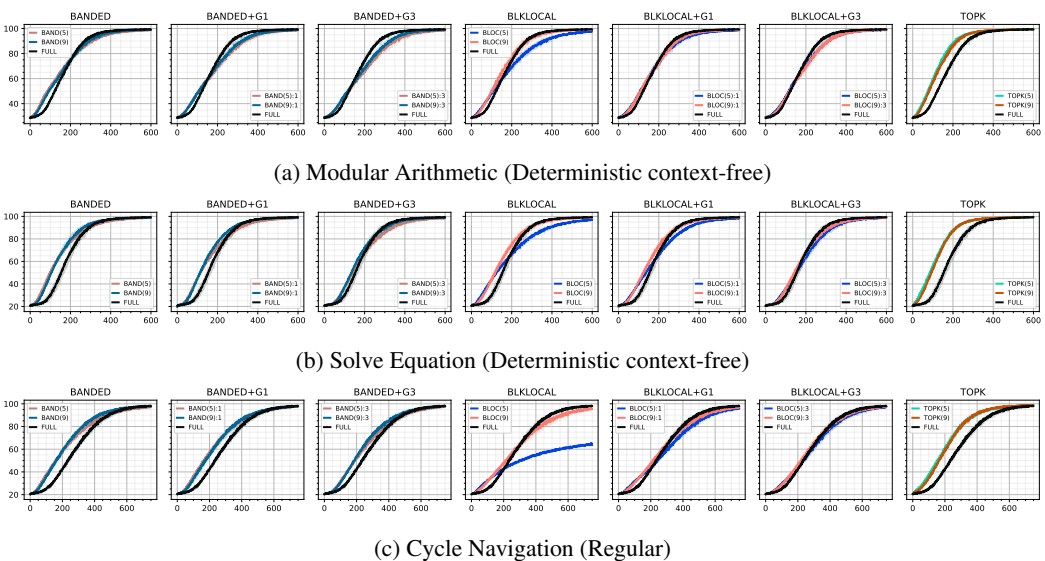

(a) Modular Arithmetic (Deterministic context-free)

(b) Solve Equation (Deterministic context-free)

(c) Cycle Navigation (Regular)

Figure 18: Same as figure 12 with GELU activation in the MLP component of the transformer block.

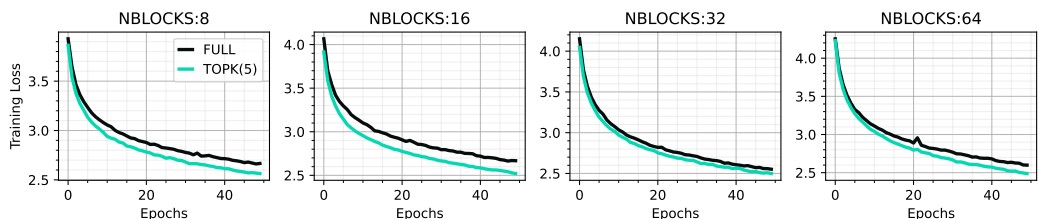

Figure 19: Training loss convergence for full attention and top-$k$ attention with a small transformer for 50 epochs with Penn Tree Bank on a vocabulary of size 4096.

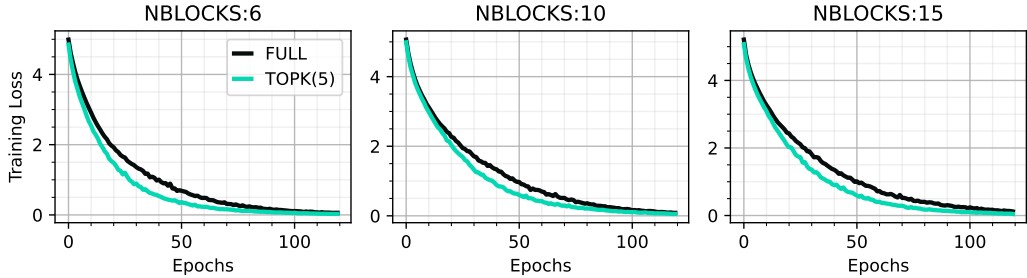

Figure 20: Training loss convergence for full attention and top-$k$ attention with a larger transformer for 120 epochs with Penn Tree Bank on a vocabulary of size 10000.

# F Softmax to Lipschitz Continuity: Technical Details

## F.1 Proof of Lemma 1

**Lemma 3.** *Consider the following assumptions:*

- *(M1) The MLP activation $\sigma$ is $\lambda_\sigma$ Lipschitz with $\sigma(0) = 0$.*
- *(M2) The MLP parameters have norms bounded by $B > 0$, that is $\|\mathbf{P}\| \leq B$ and $\|\mathbf{R}\| \leq B$.*
- *(M3) The input $\mathbf{x}$ to the MLP is bounded by $\Xi > 0$, that is $\|\mathbf{x}\| \leq \Xi$.*

*Then the token-wise MLP and LN operations are Lipschitz with respect to their input and model parameters as follows $\forall \mathbf{x}, \mathbf{x}' \in \mathbb{R}^d, \|\mathbf{x}\|, \|\bar{\mathbf{x}}\| \leq \Xi, \mathbf{P}, \bar{\mathbf{P}} \in \mathbb{R}^{d_{\mathsf{MLP}} \times d}, \mathbf{R}, \bar{\mathbf{R}} \in \mathbb{R}^{d_{\mathsf{MLP}} \times d}$:*

$$\|\mathsf{MLP}_{\mathbf{P},\mathbf{R}}(\mathbf{x}) - \mathsf{MLP}_{\mathbf{P},\mathbf{R}}(\bar{\mathbf{x}})\| \leq \eta_X \|\mathbf{x} - \bar{\mathbf{x}}\|, \tag{16}$$

$$\|\mathsf{MLP}_{\mathbf{P},\mathbf{R}}(\mathbf{x}) - \mathsf{MLP}_{\bar{\mathbf{P}},\mathbf{R}}(\mathbf{x})\| \leq \eta_P \|\mathbf{P} - \bar{\mathbf{P}}\|, \tag{17}$$

$$\|\mathsf{MLP}_{\mathbf{P},\mathbf{R}}(\mathbf{x}) - \mathsf{MLP}_{\mathbf{P},\bar{\mathbf{R}}}(\mathbf{x})\| \leq \eta_R \|\mathbf{R} - \bar{\mathbf{R}}\|, \tag{18}$$

$$\|\mathsf{LN}(\mathbf{x}) - \mathsf{LN}(\bar{\mathbf{x}})\| \leq \zeta_{\mathsf{LN}} \|\mathbf{x} - \bar{\mathbf{x}}\|, \tag{19}$$

*where $\eta_X = B^2 \lambda_\sigma$, $\eta_P = \eta_R = \lambda_\sigma B \Xi$.*

*Proof.* First, the Lipschitz property of the LayerNorm (and the corresponding value of $\zeta_{\mathsf{LN}}$) has been previously established in Kim et al. [35, Appendix N]. With LayerNorm $\mathsf{LN} : \mathbb{R}^d \to \mathbb{R}^d$ defined as

$$\mathsf{LN}(\mathbf{x}) = \frac{\mathbf{x} - \frac{1}{d}(\sum_{i \in [d]} x_i)}{\sqrt{\epsilon + \frac{1}{d}\left(x_i - \frac{1}{d}(\sum_{i \in [d]} x_i)\right)^2}} \odot \mathbf{a} + \mathbf{b}, \tag{20}$$

where $\mathbf{a}$ and $\mathbf{b}$ are the scale and shift hyperparameter. Then LayerNorm is Lipschitz with $\zeta_{\mathsf{LN}} = \epsilon^{-\frac{1}{2}} \|\mathbf{a}\|_\infty{}^{(d^2-2)/d}$ in equation (19).

For equation (16), we have the following:

$$\|\mathsf{MLP}_{\mathbf{P},\mathbf{R}}(\mathbf{x}) - \mathsf{MLP}_{\mathbf{P},\mathbf{R}}(\bar{\mathbf{x}})\| = \|\mathbf{R}^\top \sigma(\mathbf{P}\mathbf{x}) - \mathbf{R}^\top \sigma(\mathbf{P}\bar{\mathbf{x}})\| \leq \|\mathbf{R}\|\|\sigma(\mathbf{P}\mathbf{x}) - \sigma(\mathbf{P}\bar{\mathbf{x}})\| \tag{21}$$

$$\leq B\lambda_\sigma \|\mathbf{P}(\mathbf{x} - \bar{\mathbf{x}})\| \leq B\lambda_\sigma \|\mathbf{P}\|\|\mathbf{x} - \bar{\mathbf{x}}\| \leq B^2 \lambda_\sigma \|\mathbf{x} - \bar{\mathbf{x}}\|, \tag{22}$$

where we use the assumption (M1) that $\|\sigma(\mathbf{z}) - \sigma(\bar{\mathbf{z}})\| \leq \lambda_\sigma \|\mathbf{z} - \bar{\mathbf{z}}\|$, and assumption (M2) that $\|\mathbf{P}\| \leq B$ and $\|\mathbf{R}\| \leq B$.

For equation (17), we have the following:

$$\|\mathsf{MLP}_{\mathbf{P},\mathbf{R}}(\mathbf{x}) - \mathsf{MLP}_{\bar{\mathbf{P}},\mathbf{R}}(\mathbf{x})\| = \|\mathbf{R}^\top \sigma(\mathbf{P}\mathbf{x}) - \mathbf{R}^\top \sigma(\bar{\mathbf{P}}\mathbf{x})\| \leq \|\mathbf{R}\|\|\sigma(\mathbf{P}\mathbf{x}) - \sigma(\bar{\mathbf{P}}\mathbf{x})\| \tag{23}$$

$$\leq B\lambda_\sigma \|(\mathbf{P} - \bar{\mathbf{P}})\mathbf{x}\| \leq B\lambda_\sigma \Xi \|\mathbf{P} - \bar{\mathbf{P}}\|, \tag{24}$$

since $\|\mathbf{x}\| \leq \Xi$ for all tokens as per the assumption (M3).

For equation (18), we have the following:

$$\|\mathsf{MLP}_{\mathbf{P},\mathbf{R}}(\mathbf{x}) - \mathsf{MLP}_{\mathbf{P},\bar{\mathbf{R}}}(\mathbf{x})\| = \|\mathbf{R}^\top \sigma(\mathbf{P}\mathbf{x}) - \bar{\mathbf{R}}^\top \sigma(\mathbf{P}\mathbf{x})\| \tag{25}$$

$$\leq \|(\mathbf{R} - \bar{\mathbf{R}})\sigma(\mathbf{P}\mathbf{x})\| \leq \|\mathbf{R} - \bar{\mathbf{R}}\|\|\sigma(\mathbf{P}\mathbf{x})\| \tag{26}$$

$$= \|\mathbf{R} - \bar{\mathbf{R}}\|\|\sigma(\mathbf{P}\mathbf{x}) - \sigma(0)\| \tag{27}$$

$$= \|\mathbf{R} - \bar{\mathbf{R}}\|\lambda_\sigma \|\mathbf{P}\mathbf{x}\|, \leq \lambda_\sigma B \Xi \|\mathbf{R} - \bar{\mathbf{R}}\|, \tag{28}$$

since $\|\mathbf{x}\| \leq \Xi$ for all tokens as per assumption (M3) and $\sigma(0) = 0$ as per assumption (M1). $\square$

Note that the $\sigma(0) = 0$ holds for standard activations such as $\mathsf{ReLU}(x) = \max(x, 0)$ and $\mathsf{GELU}(x) = x\Phi(x)$ where $\Phi : \mathbb{R} \to [0, 1]$ is the cumulative density function of the standard Gaussian distribution.

With activations such as ReLU, the $\mathsf{MLP}(\mathbf{x}) = \mathbf{R}^\top \sigma(\mathbf{P}\mathbf{x})$ are often positive homogeneous such that, for any $\alpha \neq 0$, we have $\mathbf{R}^\top \sigma(\mathbf{P}\mathbf{x}) = \alpha \mathbf{R}^\top \sigma(\alpha^{-1}\mathbf{P}\mathbf{x})$, leading to symmetries in the paramter space,

and making analysis of optimization algorithms challenging [66–69], and some results focus on the convergence under specific conditions. However, most convergence rates depend on the Lipschitz-ness of the ReLU network, and the Lipschitz constant is not affected by this positive homogeneity as long as we assume that the matrices $\mathbf{R}, \mathbf{P}$ have bounded norms (which we do). As an example, note the following for the Lipschitz-ness with respect to $\mathbf{P}$:

$$\|\mathsf{MLP}_{\mathbf{P},\mathbf{R}}(\mathbf{x}) - \mathsf{MLP}_{\bar{\mathbf{P}},\mathbf{R}}(\mathbf{x})\| \leq \max_{\alpha} \|\mathsf{MLP}_{\alpha^{-1}\mathbf{P},\alpha\mathbf{R}}(\mathbf{x}) - \mathsf{MLP}_{\alpha^{-1}\bar{\mathbf{P}},\alpha\mathbf{R}}(\mathbf{x})\| \tag{29}$$

$$= \max_{\alpha} \|\alpha\mathbf{R}^{\top}\sigma(\alpha^{-1}\mathbf{P}\mathbf{x}) - \alpha\mathbf{R}^{\top}\sigma(\alpha^{-1}\bar{\mathbf{P}}\mathbf{x})\| \tag{30}$$

$$\leq \max_{\alpha} \|\alpha\mathbf{R}\|\|\sigma(\alpha^{-1}\mathbf{P}\mathbf{x}) - \sigma(\alpha^{-1}\bar{\mathbf{P}}\mathbf{x})\| \tag{31}$$

$$\leq \max_{\alpha} \alpha\|\mathbf{R}\|\lambda_{\sigma}\|\alpha^{-1}(\mathbf{P} - \bar{\mathbf{P}})\mathbf{x}\| \tag{32}$$

$$\leq \max_{\alpha} \alpha B\lambda_{\sigma}\alpha^{-1}\Xi\|\mathbf{P} - \bar{\mathbf{P}}\| \tag{33}$$

$$= B\lambda_{\sigma}\Xi\|\mathbf{P} - \bar{\mathbf{P}}\|, \tag{34}$$

where we see that the effect of the $\alpha$ is cancelled out and we get the result in lemma 3. Similar result can be shown for Lipschitz-ness with respect to $\mathbf{R}$.

### F.2 Proof of Theorem 1

**Theorem 6.** *Given definition 1 and lemma 1, a transformer block* $\mathsf{TF}$ *with learnable parameters* $\theta = (\mathbf{W}, \mathbf{V}, \mathbf{P}, \mathbf{R})$ *is* $\lambda_{\theta}(\xi)$*-Lipschitz with respect to its learnable parameters* $\theta$ *with*

$$\lambda_{\theta}(\xi) = \zeta_{\mathsf{LN}}\left(\zeta_{\mathsf{LN}}(1 + \eta_X)(\lambda_W(\xi) + \lambda_V) + L(\eta_P + \eta_R)\right), \tag{35}$$

*and* $\mathsf{TF}$ *is* $\lambda_{\mathbf{X}}(\xi)$*-Lipschitz with respect to its input* $\mathbf{X}$ *with*

$$\lambda_{\mathbf{X}}(\xi) = \zeta_{\mathsf{LN}}^2(1 + \eta_X)(1 + \lambda_X(\xi)), \tag{36}$$

*where we explicitly note the dependence of the Lipschitz constant with respect to learnable parameters* $\lambda_{\theta}(\xi)$*, and input* $\lambda_{\mathbf{X}}(\xi)$ *to the Lipschitz constant* $\xi$ *of the (masked)* $\mathsf{softmax}$ *operation.*

*Proof.* Let $\theta = (\mathbf{W}, \mathbf{V}, \mathbf{P}, \mathbf{R})$ and $\bar{\theta} = (\bar{\mathbf{W}}, \bar{\mathbf{V}}, \bar{\mathbf{P}}, \bar{\mathbf{R}})$. Then, we have the following:

$$\|\mathsf{TF}_{\theta}(\mathbf{X}) - \mathsf{TF}_{\bar{\theta}}(\mathbf{X})\|_{2,1} = \|\mathsf{TF}_{\mathbf{W},\mathbf{V},\mathbf{P},\mathbf{R}}(\mathbf{X}) - \mathsf{TF}_{\bar{\mathbf{W}},\bar{\mathbf{V}},\bar{\mathbf{P}},\bar{\mathbf{R}}}(\mathbf{X})\|_{2,1} \tag{37}$$

$$\leq \|\mathsf{TF}_{\mathbf{W},\mathbf{V},\mathbf{P},\mathbf{R}}(\mathbf{X}) - \mathsf{TF}_{\mathbf{W},\mathbf{V},\mathbf{P},\bar{\mathbf{R}}}(\mathbf{X})\|_{2,1} \tag{$T_1$}$$

$$+ \|\mathsf{TF}_{\mathbf{W},\mathbf{V},\mathbf{P},\bar{\mathbf{R}}}(\mathbf{X}) - \mathsf{TF}_{\mathbf{W},\mathbf{V},\bar{\mathbf{P}},\bar{\mathbf{R}}}(\mathbf{X})\|_{2,1} \tag{$T_2$}$$

$$+ \|\mathsf{TF}_{\mathbf{W},\mathbf{V},\bar{\mathbf{P}},\bar{\mathbf{R}}}(\mathbf{X}) - \mathsf{TF}_{\mathbf{W},\bar{\mathbf{V}},\bar{\mathbf{P}},\bar{\mathbf{R}}}(\mathbf{X})\|_{2,1} \tag{$T_3$}$$

$$+ \|\mathsf{TF}_{\mathbf{W},\bar{\mathbf{V}},\bar{\mathbf{P}},\bar{\mathbf{R}}}(\mathbf{X}) - \mathsf{TF}_{\bar{\mathbf{W}},\bar{\mathbf{V}},\bar{\mathbf{P}},\bar{\mathbf{R}}}(\mathbf{X})\|_{2,1}. \tag{$T_4$}$$

First, processing equation ($T_1$), let us denote with $\widetilde{\mathbf{X}} = \mathsf{LN}(\mathbf{X} + \mathsf{A}_{\mathbf{W},\mathbf{V}}(\mathbf{X}))$, then

$$(T_1) = \|\mathsf{TF}_{\mathbf{W},\mathbf{V},\mathbf{P},\mathbf{R}}(\mathbf{X}) - \mathsf{TF}_{\mathbf{W},\mathbf{V},\mathbf{P},\bar{\mathbf{R}}}(\mathbf{X})\|_{2,1} \tag{38}$$

$$= \|\mathsf{LN}(\widetilde{\mathbf{X}} + \mathsf{MLP}_{\mathbf{P},\mathbf{R}}(\widetilde{\mathbf{X}})) - \mathsf{LN}(\widetilde{\mathbf{X}} + \mathsf{MLP}_{\mathbf{P},\bar{\mathbf{R}}}(\widetilde{\mathbf{X}}))\|_{2,1} \tag{39}$$

$$= \sum_{i \in [L]} \|\mathsf{LN}(\widetilde{\mathbf{X}}_{:i} + \mathsf{MLP}_{\mathbf{P},\mathbf{R}}(\widetilde{\mathbf{X}}_{:i})) - \mathsf{LN}(\widetilde{\mathbf{X}}_{:i} + \mathsf{MLP}_{\mathbf{P},\bar{\mathbf{R}}}(\widetilde{\mathbf{X}}_{:i}))\| \tag{40}$$

$$\leq \sum_{i \in [L]} \zeta_{\mathsf{LN}}\|\mathsf{MLP}_{\mathbf{P},\mathbf{R}}(\widetilde{\mathbf{X}}_{:i}) - \mathsf{MLP}_{\mathbf{P},\bar{\mathbf{R}}}(\widetilde{\mathbf{X}}_{:i}))\| \quad \text{(using equation (19))} \tag{41}$$

$$\leq \sum_{i \in [L]} \zeta_{\mathsf{LN}}\eta_R\|\mathbf{R} - \bar{\mathbf{R}}\| = L\zeta_{\mathsf{LN}}\eta_R\|\mathbf{R} - \bar{\mathbf{R}}\| \quad \text{(using equation (18))}. \tag{42}$$

Handling equation ($T_2$) in a similar fashion, we have

$$(T_2) = \|\mathsf{TF}_{\mathbf{W},\mathbf{V},\mathbf{P},\bar{\mathbf{R}}}(\mathbf{X}) - \mathsf{TF}_{\mathbf{W},\mathbf{V},\bar{\mathbf{P}},\bar{\mathbf{R}}}(\mathbf{X})\|_{2,1} \tag{43}$$

$$= \|\mathsf{LN}(\widetilde{\mathbf{X}} + \mathsf{MLP}_{\mathbf{P},\bar{\mathbf{R}}}(\widetilde{\mathbf{X}})) - \mathsf{LN}(\widetilde{\mathbf{X}} + \mathsf{MLP}_{\bar{\mathbf{P}},\bar{\mathbf{R}}}(\widetilde{\mathbf{X}}))\|_{2,1} \tag{44}$$

$$= \sum_{i \in [L]} \|\mathsf{LN}(\widetilde{\mathbf{X}}_{:i} + \mathsf{MLP}_{\mathbf{P},\bar{\mathbf{R}}}(\widetilde{\mathbf{X}}_{:i})) - \mathsf{LN}(\widetilde{\mathbf{X}}_{:i} + \mathsf{MLP}_{\bar{\mathbf{P}},\bar{\mathbf{R}}}(\widetilde{\mathbf{X}}_{:i}))\| \tag{45}$$

$$\le \sum_{i \in [L]} \zeta_{\mathsf{LN}}\|\mathsf{MLP}_{\mathbf{P},\bar{\mathbf{R}}}(\widetilde{\mathbf{X}}_{:i}) - \mathsf{MLP}_{\bar{\mathbf{P}},\bar{\mathbf{R}}}(\widetilde{\mathbf{X}}_{:i}))\| \quad \text{(using equation (19))} \tag{46}$$

$$\le \sum_{i \in [L]} \zeta_{\mathsf{LN}}\eta_P\|\mathbf{P} - \bar{\mathbf{P}}\| = L\zeta_{\mathsf{LN}}\eta_P\|\mathbf{P} - \bar{\mathbf{P}}\| \quad \text{(using equation (17)).} \tag{47}$$

For equation $(T_3)$, let us denote with $\widetilde{\mathbf{X}}' = \mathsf{LN}(\mathbf{X} + \mathsf{A}_{\mathbf{W},\bar{\mathbf{V}}}(\mathbf{X}))$. Then we have

$$(T_3) = \|\mathsf{TF}_{\mathbf{W},\mathbf{V},\bar{\mathbf{P}},\bar{\mathbf{R}}}(\mathbf{X}) - \mathsf{TF}_{\mathbf{W},\bar{\mathbf{V}},\bar{\mathbf{P}},\bar{\mathbf{R}}}(\mathbf{X})\|_{2,1} \tag{48}$$

$$= \|\mathsf{LN}(\widetilde{\mathbf{X}} + \mathsf{MLP}_{\bar{\mathbf{P}},\bar{\mathbf{R}}}(\widetilde{\mathbf{X}})) - \mathsf{LN}(\widetilde{\mathbf{X}}' + \mathsf{MLP}_{\bar{\mathbf{P}},\bar{\mathbf{R}}}(\widetilde{\mathbf{X}}'))\|_{2,1} \tag{49}$$

$$= \sum_{i \in [L]} \|\mathsf{LN}(\widetilde{\mathbf{X}}_{:i} + \mathsf{MLP}_{\bar{\mathbf{P}},\bar{\mathbf{R}}}(\widetilde{\mathbf{X}}_{:i})) - \mathsf{LN}(\widetilde{\mathbf{X}}'_{:i} + \mathsf{MLP}_{\bar{\mathbf{P}},\bar{\mathbf{R}}}(\widetilde{\mathbf{X}}'_{:i}))\| \tag{50}$$

$$\le \sum_{i \in [L]} \zeta_{\mathsf{LN}}\|(\widetilde{\mathbf{X}}_{:i} - \widetilde{\mathbf{X}}'_{:i}) + (\mathsf{MLP}_{\bar{\mathbf{P}},\bar{\mathbf{R}}}(\widetilde{\mathbf{X}}_{:i}) - \mathsf{MLP}_{\bar{\mathbf{P}},\bar{\mathbf{R}}}(\widetilde{\mathbf{X}}'_{:i}))\| \quad \text{(using equation (19))} \tag{51}$$

$$\le \sum_{i \in [L]} \zeta_{\mathsf{LN}}(1 + \eta_X)\|(\widetilde{\mathbf{X}}_{:i} - \widetilde{\mathbf{X}}'_{:i})\| \quad \text{(using equation (16))} \tag{52}$$

$$= \sum_{i \in [L]} \zeta_{\mathsf{LN}}(1 + \eta_X)\|\mathsf{LN}(\mathbf{X}_{:i} + \mathsf{A}_{\mathbf{W},\mathbf{V}}(\mathbf{X})_{:i}) - \mathsf{LN}(\mathbf{X}_{:i} + \mathsf{A}_{\mathbf{W},\bar{\mathbf{V}}}(\mathbf{X})_{:i})\| \tag{53}$$

$$\le \sum_{i \in [L]} \zeta_{\mathsf{LN}}(1 + \eta_X)\zeta_{\mathsf{LN}}\|\mathsf{A}_{\mathbf{W},\mathbf{V}}(\mathbf{X})_{:i} - \mathsf{A}_{\mathbf{W},\bar{\mathbf{V}}}(\mathbf{X})_{:i}\| \quad \text{(using equation (19))} \tag{54}$$

$$= \zeta_{\mathsf{LN}}(1 + \eta_X)\zeta_{\mathsf{LN}}\|\mathsf{A}_{\mathbf{W},\mathbf{V}}(\mathbf{X}) - \mathsf{A}_{\mathbf{W},\bar{\mathbf{V}}}(\mathbf{X})\|_{2,1} \tag{55}$$

$$\le \zeta_{\mathsf{LN}}(1 + \eta_X)\zeta_{\mathsf{LN}}\lambda_V\|\mathbf{V} - \bar{\mathbf{V}}\| \quad \text{(using equation (7) in definition 1).} \tag{56}$$

For equation $(T_4)$, let us denote with $\widetilde{\mathbf{X}}'' = \mathsf{LN}(\mathbf{X} + \mathsf{A}_{\bar{\mathbf{W}},\bar{\mathbf{V}}}(\mathbf{X}))$. Then we can follow the same procedure as for equation $(T_3)$ and get the following:

$$(T_4) = \|\mathsf{TF}_{\mathbf{W},\bar{\mathbf{V}},\bar{\mathbf{P}},\bar{\mathbf{R}}}(\mathbf{X}) - \mathsf{TF}_{\bar{\mathbf{W}},\bar{\mathbf{V}},\bar{\mathbf{P}},\bar{\mathbf{R}}}(\mathbf{X})\|_{2,1} \tag{57}$$

$$= \|\mathsf{LN}(\widetilde{\mathbf{X}}' + \mathsf{MLP}_{\bar{\mathbf{P}},\bar{\mathbf{R}}}(\widetilde{\mathbf{X}}')) - \mathsf{LN}(\widetilde{\mathbf{X}}'' + \mathsf{MLP}_{\bar{\mathbf{P}},\bar{\mathbf{R}}}(\widetilde{\mathbf{X}}''))\|_{2,1} \tag{58}$$

$$= \sum_{i \in [L]} \|\mathsf{LN}(\widetilde{\mathbf{X}}'_{:i} + \mathsf{MLP}_{\bar{\mathbf{P}},\bar{\mathbf{R}}}(\widetilde{\mathbf{X}}'_{:i})) - \mathsf{LN}(\widetilde{\mathbf{X}}''_{:i} + \mathsf{MLP}_{\bar{\mathbf{P}},\bar{\mathbf{R}}}(\widetilde{\mathbf{X}}''_{:i}))\| \tag{59}$$

$$\le \sum_{i \in [L]} \zeta_{\mathsf{LN}}\|(\widetilde{\mathbf{X}}'_{:i} - \widetilde{\mathbf{X}}''_{:i}) + (\mathsf{MLP}_{\bar{\mathbf{P}},\bar{\mathbf{R}}}(\widetilde{\mathbf{X}}'_{:i}) - \mathsf{MLP}_{\bar{\mathbf{P}},\bar{\mathbf{R}}}(\widetilde{\mathbf{X}}''_{:i}))\| \quad \text{(using equation (19))} \tag{60}$$

$$\le \sum_{i \in [L]} \zeta_{\mathsf{LN}}(1 + \eta_X)\|(\widetilde{\mathbf{X}}'_{:i} - \widetilde{\mathbf{X}}''_{:i})\| \quad \text{(using equation (16))} \tag{61}$$

$$= \sum_{i \in [L]} \zeta_{\mathsf{LN}}(1 + \eta_X)\|\mathsf{LN}(\mathbf{X}_{:i} + \mathsf{A}_{\mathbf{W},\bar{\mathbf{V}}}(\mathbf{X})_{:i}) - \mathsf{LN}(\mathbf{X}_{:i} + \mathsf{A}_{\bar{\mathbf{W}},\bar{\mathbf{V}}}(\mathbf{X})_{:i})\| \tag{62}$$

$$\le \sum_{i \in [L]} \zeta_{\mathsf{LN}}(1 + \eta_X)\zeta_{\mathsf{LN}}\|\mathsf{A}_{\mathbf{W},\bar{\mathbf{V}}}(\mathbf{X})_{:i} - \mathsf{A}_{\bar{\mathbf{W}},\bar{\mathbf{V}}}(\mathbf{X})_{:i}\| \quad \text{(using equation (19))} \tag{63}$$

$$= \zeta_{\mathsf{LN}}(1 + \eta_X)\zeta_{\mathsf{LN}}\|\mathsf{A}_{\mathbf{W},\bar{\mathbf{V}}}(\mathbf{X}) - \mathsf{A}_{\bar{\mathbf{W}},\bar{\mathbf{V}}}(\mathbf{X})\|_{2,1} \tag{64}$$

$$\leq \zeta_{\mathsf{LN}}(1 + \eta_X)\zeta_{\mathsf{LN}}\lambda_W(\xi)\|\mathbf{W} - \bar{\mathbf{W}}\| \quad \text{(using equation (6) in definition 1).} \tag{65}$$

Putting these all together, we have

$$\|\mathsf{TF}_\theta(\mathbf{X}) - \mathsf{TF}_{\bar{\theta}}(\mathbf{X})\|_{2,1} \tag{66}$$
$$\leq \zeta_{\mathsf{LN}} L \left( \eta_R \|\mathbf{R} - \bar{\mathbf{R}}\| + \eta_P \|\mathbf{P} - \bar{\mathbf{P}}\| \right)$$
$$+ \zeta_{\mathsf{LN}}^2 (1 + \eta_X) \left( \lambda_V \|\mathbf{V} - \bar{\mathbf{V}}\| + \lambda_W(\xi)\|\mathbf{W} - \bar{\mathbf{W}}\| \right) \tag{67}$$
$$\leq \zeta_{\mathsf{LN}} L \left( \eta_R \|\theta - \bar{\theta}\| + \eta_P \|\theta - \bar{\theta}\| \right) + \zeta_{\mathsf{LN}}^2 (1 + \eta_X) \left( \lambda_V \|\theta - \bar{\theta}\| + \lambda_W(\xi)\|\theta - \bar{\theta}\| \right) \tag{68}$$
$$= \zeta_{\mathsf{LN}} \left( \zeta_{\mathsf{LN}}(1 + \eta_X)(\lambda_W(\xi) + \lambda_V) + L(\eta_P + \eta_R) \right) \|\theta - \bar{\theta}\|, \tag{69}$$

where we used the definition that, for matrix tuples $\theta, \bar{\theta}$, $\|\theta - \bar{\theta}\| = \max\{\|\mathbf{W} - \bar{\mathbf{W}}\|, \|\mathbf{V} - \bar{\mathbf{V}}\|, \|\mathbf{P} - \bar{\mathbf{P}}\|, \|\mathbf{R} - \bar{\mathbf{R}}\|\}$. This gives us the desired result in equation (35).

For inputs $\mathbf{X}, \bar{\mathbf{X}}$, let $\widetilde{\mathbf{X}} = \mathsf{LN}(\mathbf{X} + \mathsf{A}_{\mathbf{W},\mathbf{V}}(\mathbf{X}))$ and $\widetilde{\mathbf{X}}' = \mathsf{LN}(\bar{\mathbf{X}} + \mathsf{A}_{\mathbf{W},\mathbf{V}}(\bar{\mathbf{X}}))$. Then we have the following:

$$\|\mathsf{TF}_\theta(\mathbf{X}) - \mathsf{TF}_\theta(\bar{\mathbf{X}})\|_{2,1} = \|\mathsf{LN}(\widetilde{\mathbf{X}} + \mathsf{MLP}_{\mathbf{P},\mathbf{R}}(\widetilde{\mathbf{X}})) - \mathsf{LN}(\widetilde{\mathbf{X}}' + \mathsf{MLP}_{\mathbf{P},\mathbf{R}}(\widetilde{\mathbf{X}}'))\|_{2,1} \tag{70}$$
$$= \sum_{i \in [L]} \|\mathsf{LN}(\widetilde{\mathbf{X}}_{:i} + \mathsf{MLP}_{\mathbf{P},\mathbf{R}}(\widetilde{\mathbf{X}}_{:i})) - \mathsf{LN}(\widetilde{\mathbf{X}}'_{:i} + \mathsf{MLP}_{\mathbf{P},\mathbf{R}}(\widetilde{\mathbf{X}}'_{:i}))\| \tag{71}$$
$$\leq \sum_{i \in [L]} \zeta_{\mathsf{LN}} \|(\widetilde{\mathbf{X}}_{:i} - \widetilde{\mathbf{X}}'_{:i}) + \left( \mathsf{MLP}_{\mathbf{P},\mathbf{R}}(\widetilde{\mathbf{X}}_{:i}) - \mathsf{MLP}_{\mathbf{P},\mathbf{R}}(\widetilde{\mathbf{X}}'_{:i}) \right)\| \tag{72}$$
$$\text{(using equation (19))}$$
$$\leq \sum_{i \in [L]} \zeta_{\mathsf{LN}}(1 + \eta_X) \|\widetilde{\mathbf{X}}_{:i} - \widetilde{\mathbf{X}}'_{:i}\| \quad \text{(using equation (16))} \tag{73}$$
$$= \sum_{i \in [L]} \zeta_{\mathsf{LN}}(1 + \eta_X) \|\mathsf{LN}(\mathbf{X}_{:i} + \mathsf{A}_{\mathbf{W},\mathbf{V}}(\mathbf{X})_{:i}) - \mathsf{LN}(\bar{\mathbf{X}}_{:i} + \mathsf{A}_{\mathbf{W},\mathbf{V}}(\bar{\mathbf{X}})_{:i})\|$$
$$\tag{74}$$
$$\leq \sum_{i \in [L]} \zeta_{\mathsf{LN}}^2 (1 + \eta_X) \|(\mathbf{X}_{:i} - \bar{\mathbf{X}}_{:i}) + (\mathsf{A}_{\mathbf{W},\mathbf{V}}(\mathbf{X})_{:i} - \mathsf{A}_{\mathbf{W},\mathbf{V}}(\bar{\mathbf{X}})_{:i})\| \tag{75}$$
$$\text{(using equation (19))}$$
$$\leq \zeta_{\mathsf{LN}}^2 (1 + \eta_X) \|(\mathbf{X} - \bar{\mathbf{X}}) + (\mathsf{A}_{\mathbf{W},\mathbf{V}}(\mathbf{X}) - \mathsf{A}_{\mathbf{W},\mathbf{V}}(\bar{\mathbf{X}}))\|_{2,1} \tag{76}$$
$$= \zeta_{\mathsf{LN}}^2 (1 + \eta_X) \left( \|\mathbf{X} - \bar{\mathbf{X}}\|_{2,1} + \|\mathsf{A}_{\mathbf{W},\mathbf{V}}(\mathbf{X}) - \mathsf{A}_{\mathbf{W},\mathbf{V}}(\bar{\mathbf{X}})\|_{2,1} \right) \tag{77}$$
$$\leq \zeta_{\mathsf{LN}}^2 (1 + \eta_X)(1 + \lambda_X(\xi)) \|\mathbf{X} - \bar{\mathbf{X}}\|_{2,1} \tag{78}$$
$$\text{(using equation (5) in definition 1),}$$

which gives us the desired result in equation (36). $\square$

### F.3 Proof of Theorem 2

**Corollary 1.** *Consider the following assumptions:*

- *(L1) The sample wise loss $\ell$ in equation (4) is $\alpha$-Lipschitz.*
- *(L2) The final readout layer weights are norm-bounded as $\|\mathbf{\Phi}\| \leq 1$ and the per-token output of each transformer block is norm bounded as $\|\mathbf{X}_{:i}^{(t)}\| \leq \Xi$ for all $i \in [L]$ and $t \in [\tau]$.*
- *(L3) The sequence aggregator is $\boldsymbol{\omega} = (1/L)\mathbf{1}_L$.*

*Under the above assumptions and the conditions of definition 1 and theorem 1, the learning objective $\mathcal{L}$ in equation (4) is $\lambda_{\mathcal{L}}(\xi)$-Lipschitz with respect to the learnable parameters $\Theta = (\mathbf{T}, \theta^{(1)}, \ldots, \theta^{(\tau)}, \mathbf{\Phi})$, where*

$$\lambda_{\mathcal{L}}(\xi) = \alpha \left( \Xi + \lambda_X(\xi)^\tau \left( 1 + \frac{\lambda_\theta(\xi)}{L(\lambda_X(\xi) - 1)} \right) \right). \tag{79}$$

*Proof.* Let us first denote the model parameter tuples as $\Theta = (\mathbf{T}, \theta^{(1)}, \ldots, \theta^{(\tau)}, \mathbf{\Phi})$ and $\bar{\Theta} = (\bar{\mathbf{T}}, \bar{\theta}^{(1)}, \ldots, \bar{\theta}^{(\tau)}, \bar{\mathbf{\Phi}})$. Let $\mathbf{X}^{(0)} = [\mathbf{T}_{v_1} + \mathbf{E}_1, \ldots, \mathbf{T}_{v_L} + \mathbf{E}_L]$ and $\bar{\mathbf{X}}^{(0)} = [\bar{\mathbf{T}}_{v_1} + \mathbf{E}_1, \ldots, \bar{\mathbf{T}}_{v_L} + \mathbf{E}_L]$ denote the initial token embeddings for the same input $X = [v_1, \ldots, v_L], v_i \in [D]$, with model parameters $\Theta$ and $\bar{\Theta}$ respectively. Note that we are not learning the position encoding $\mathbf{E}$ in our setup. For any $t = 1, \ldots, \tau$, let $\mathbf{X}^{(t)} = \mathsf{TF}_{\theta^{(t)}}(\mathbf{X}^{(t-1)})$ and $\bar{\mathbf{X}}^{(t)} = \mathsf{TF}_{\bar{\theta}^{(t)}}(\bar{\mathbf{X}}^{(t-1)})$, both defined recursively.

Then, using the loss function $\mathcal{L}$ in equation (4), we have the following:

$$|\mathcal{L}(\Theta) - \mathcal{L}(\bar{\Theta})| = \left| \frac{1}{n} \sum_{(X,y)\in S} (\ell(y, f_\Theta(X)) - \ell(y, f_{\bar{\Theta}}(X))) \right| \tag{80}$$

$$\leq \frac{1}{n} \sum_{(X,y)\in S} \alpha \, |f_\Theta(X) - f_{\bar{\Theta}}(X)|, \tag{81}$$

where we utilized the assumption that $\ell$ is $\alpha$-Lipschitz. Focusing on the $|f_\Theta(X) - f_{\bar{\Theta}}(X)|$ term in equation (81), we see the following:

$$|f_\Theta(X) - f_{\bar{\Theta}}(X)| = \left| \mathbf{\Phi}(\mathbf{X}^{(\tau)}\boldsymbol{\omega}) - \bar{\mathbf{\Phi}}(\bar{\mathbf{X}}^{(\tau)}\boldsymbol{\omega}) \right| \tag{82}$$

$$= \left| \mathbf{\Phi}\left( \frac{1}{L} \sum_{i=1}^L (\mathbf{X}^{(\tau)}_{:i} - \bar{\mathbf{X}}^{(\tau)}_{:i}) \right) + (\mathbf{\Phi} - \bar{\mathbf{\Phi}})\left( \frac{1}{L} \sum_{i=1}^L \bar{\mathbf{X}}^{(\tau)}_{:i} \right) \right| \tag{83}$$

$$\leq \|\mathbf{\Phi}\| \left( \frac{1}{L} \sum_{i=1}^L \|\mathbf{X}^{(\tau)}_{:i} - \bar{\mathbf{X}}^{(\tau)}_{:i}\| \right) + \|\mathbf{\Phi} - \bar{\mathbf{\Phi}}\| \left( \frac{1}{L} \sum_{i=1}^L \|\bar{\mathbf{X}}^{(\tau)}_{:i}\| \right) \tag{84}$$

$$\leq \frac{1}{L} \|\mathbf{X}^{(\tau)} - \bar{\mathbf{X}}^{(\tau)}\|_{2,1} + \Xi\|\mathbf{\Phi} - \bar{\mathbf{\Phi}}\|, \tag{85}$$

where we utilized the assumption that $\|\mathbf{\Phi}\| \leq 1$ and $\|\bar{\mathbf{X}}_{:i}\| \leq \Xi \forall i \in [L]$. Considering the $\|\mathbf{X}^{(\tau)} - \bar{\mathbf{X}}^{(\tau)}\|_{2,1}$ in the right-hand-side of equation (85), and noting the recursive definition of $\bar{\mathbf{X}}^{(t)} = \mathsf{TF}_{\theta^{(t)}}(\bar{\mathbf{X}}^{(t-1)})$, we have the following:

$$\|\mathbf{X}^{(\tau)} - \bar{\mathbf{X}}^{(\tau)}\|_{2,1} \tag{86}$$

$$= \left\| \mathsf{TF}_{\theta^{(\tau)}}(\mathsf{TF}_{\theta^{(\tau-1)}}(\cdots(\mathsf{TF}_{\theta^{(1)}}(\mathbf{X}^{(0)})))) - \mathsf{TF}_{\bar{\theta}^{(\tau)}}(\mathsf{TF}_{\bar{\theta}^{(\tau-1)}}(\cdots(\mathsf{TF}_{\bar{\theta}^{(1)}}(\bar{\mathbf{X}}^{(0)})))) \right\|_{2,1} \tag{87}$$

$$\leq \left\| \mathsf{TF}_{\theta^{(\tau)}}(\cdots(\mathsf{TF}_{\theta^{(1)}}(\mathbf{X}^{(0)}))) - \mathsf{TF}_{\theta^{(\tau)}}(\cdots(\mathsf{TF}_{\theta^{(1)}}(\bar{\mathbf{X}}^{(0)}))) \right\|_{2,1} \tag{$P_1$}$$

$$+ \sum_{t=1}^{\tau-1} \left\| \mathsf{TF}_{\theta^{(\tau)}}(\cdots(\mathsf{TF}_{\theta^{(t)}}(\bar{\mathbf{X}}^{(t-1)}))) - \mathsf{TF}_{\theta^{(\tau)}}(\cdots(\mathsf{TF}_{\bar{\theta}^{(t)}}(\bar{\mathbf{X}}^{(t-1)}))) \right\|_{2,1} \tag{$P_2$}$$

$$+ \left\| \mathsf{TF}_{\theta^{(\tau)}}(\bar{\mathbf{X}}^{(t-1)}) - \mathsf{TF}_{\bar{\theta}^{(\tau)}}(\bar{\mathbf{X}}^{(t-1)}) \right\|_{2,1} \tag{$P_3$}$$

Utilizing the $\lambda_\mathbf{X}(\xi)$-Lipschitzness of each transformer block with respect to the input (as per theorem 6, equation (36)), and applying it recursively through the $\tau$ transformer blocks, we can bound equation ($P_1$) as:

$$(P_1) \leq \lambda_\mathbf{X}(\xi)^\tau \|\mathbf{X}^{(0)} - \bar{\mathbf{X}}^{(0)}\|_{2,1} = \lambda_\mathbf{X}(\xi)^\tau \sum_{i=1}^L \|\mathbf{T}_{v_i} - \bar{\mathbf{T}}_{v_i}\| \tag{88}$$

$$\leq \lambda_\mathbf{X}(\xi)^\tau \sum_{i=1}^L \|\mathbf{T} - \bar{\mathbf{T}}\| = \lambda_\mathbf{X}(\xi)^\tau L \|\mathbf{T} - \bar{\mathbf{T}}\|. \tag{89}$$

For equation ($P_2$), we will again utilize the $\lambda_\mathbf{X}(\xi)$-Lipschitzness of each transformer block with respect to the input recursively to get the following:

$$(P_2) = \sum_{t=1}^{\tau-1} \left\| \mathsf{TF}_{\theta^{(\tau)}}(\cdots(\mathsf{TF}_{\theta^{(t)}}(\bar{\mathbf{X}}^{(t-1)}))) - \mathsf{TF}_{\theta^{(\tau)}}(\cdots(\mathsf{TF}_{\bar{\theta}^{(t)}}(\bar{\mathbf{X}}^{(t-1)}))) \right\|_{2,1} \tag{90}$$

$$\leq \sum_{t=1}^{\tau-1} \lambda_{\mathbf{X}}(\xi)^{\tau-t} \left\| \mathsf{TF}_{\theta^{(t)}}(\bar{\mathbf{X}}^{(t-1)}) - \mathsf{TF}_{\bar{\theta}^{(t)}}(\bar{\mathbf{X}}^{(t-1)}) \right\|_{2,1} \tag{91}$$

$$\leq \sum_{t=1}^{\tau-1} \lambda_{\mathbf{X}}(\xi)^{\tau-t} \lambda_{\theta}(\xi) \|\theta^{(t)} - \bar{\theta}^{(t)}\|, \tag{92}$$

where we utilize the $\lambda_{\theta}(\xi)$-Lipschitzness of the transformer block with respect to the parameters in the last inequality.

We can use $\lambda_{\theta}(\xi)$-Lipschitzness of each transformer block with respect to the parameters (as per theorem 1, equation (9)) to bound equation ($P_3$) with $\lambda_{\theta}(\xi)\|\theta^{(\tau)} - \bar{\theta}^{(\tau)}\|$.

Substituting this, equation (89) and equation (92) in equation (85), we have

$$|f_{\Theta}(X) - f_{\bar{\Theta}}(X)| \tag{93}$$

$$\leq \frac{1}{L} \left( \lambda_{\mathbf{X}}(\xi)^{\tau} L \|\mathbf{T} - \bar{\mathbf{T}}\| + \left( \lambda_{\theta}(\xi) \sum_{t=1}^{\tau-1} \lambda_{\mathbf{X}}(\xi)^{\tau-t} \|\theta^{(t)} - \bar{\theta}^{(t)}\| \right) + \lambda_{\theta}(\xi) \|\theta^{(\tau)} - \bar{\theta}^{(\tau)}\| \right)$$

$$+ \Xi \|\mathbf{\Phi} - \bar{\mathbf{\Phi}}\| \tag{94}$$

$$\leq \frac{1}{L} \left( \lambda_{\mathbf{X}}(\xi)^{\tau} L \|\Theta - \bar{\Theta}\| + \left( \lambda_{\theta}(\xi) \sum_{t=1}^{\tau-1} \lambda_{\mathbf{X}}(\xi)^{\tau-t} \|\Theta - \bar{\Theta}\| \right) + \lambda_{\theta}(\xi) \|\Theta - \bar{\Theta}\| \right)$$

$$+ \Xi \|\Theta - \bar{\Theta}\| \tag{95}$$

$$= \left( \Xi + \lambda_{\mathbf{X}}(\xi)^{\tau} \left( 1 + \frac{\lambda_{\theta}(\xi)}{L(\lambda_{\mathbf{X}}(\xi) - 1)} \right) \right) \|\Theta - \bar{\Theta}\|. \tag{96}$$

Finally, substituting the above in equation (81) gives us:

$$|\mathcal{L}(\Theta) - \mathcal{L}(\bar{\Theta})| \leq \frac{1}{n} \sum_{(X,y) \in S} \alpha \left( \Xi + \lambda_{\mathbf{X}}(\xi)^{\tau} \left( 1 + \frac{\lambda_{\theta}(\xi)}{L(\lambda_{\mathbf{X}}(\xi) - 1)} \right) \right) \|\Theta - \bar{\Theta}\| \tag{97}$$

$$\leq \alpha \left( \Xi + \lambda_{\mathbf{X}}(\xi)^{\tau} \left( 1 + \frac{\lambda_{\theta}(\xi)}{L(\lambda_{\mathbf{X}}(\xi) - 1)} \right) \right) \|\Theta - \bar{\Theta}\|. \tag{98}$$

This gives us equation (79) in the statement of the corollary. $\square$

## F.4 Multi-headed Attention

As per Yun et al. [2, Section 2, equation 1], we can write multi-headed (self) attention $h$ heads in our notation as:

$$\mathsf{MHA}_{\{\mathbf{W}^{(i)}, \mathbf{V}^{(i)}, \mathbf{H}^{(i)}\}_{i \in [h]}}(\mathbf{X}) = \sum_{i \in [h]} \mathbf{H}^{(i)} \mathsf{A}_{\mathbf{W}^{(i)}, \mathbf{V}^{(i)}}(\mathbf{X}), \tag{99}$$

where $\mathbf{H}^{(i)} \in \mathbb{R}^{d \times d}$ are the head-aggregator matrices. Here we are assuming that each head is of size $d$, same as the $d_{\text{model}}$. This is for ease of exposition, as we can introduce a new variable for head size and get the same guarantees.

Now, for each of the heads $i \in [h]$, let us assume the following (as in definition 1):

$$\|\mathsf{A}_{\mathbf{W}^{(i)}, \mathbf{V}^{(i)}}(\mathbf{X}) - \mathsf{A}_{\mathbf{W}^{(i)}, \mathbf{V}^{(i)}}(\bar{\mathbf{X}})\|_{2,1} \leq \lambda_X(\xi) \|\mathbf{X} - \bar{\mathbf{X}}\|_{2,1}, \tag{100}$$

$$\|\mathsf{A}_{\mathbf{W}^{(i)}, \mathbf{V}^{(i)}}(\mathbf{X}) - \mathsf{A}_{\bar{\mathbf{W}}^{(i)}, \mathbf{V}^{(i)}}(\mathbf{X})\|_{2,1} \leq \lambda_W(\xi) \|\mathbf{W} - \bar{\mathbf{W}}\|. \tag{101}$$

Then, we can show the following for multi-headed attention, assuming $\|\mathbf{H}^i\| \leq \Lambda$ for all $i \in [h]$:

$$\|\mathsf{MHA}_{\{\mathbf{W}^{(i)}, \mathbf{V}^{(i)}, \mathbf{H}^{(i)}\}_{i \in [h]}}(\mathbf{X}) - \mathsf{MHA}_{\{\mathbf{W}^{(i)}, \mathbf{V}^{(i)}, \mathbf{H}^{(i)}\}_{i \in [h]}}(\bar{\mathbf{X}})\|_{2,1} \tag{102}$$

$$= \left\| \sum_{i=1}^{h} \mathbf{H}^{(i)} \left( \mathsf{A}_{\mathbf{W}^{(i)}, \mathbf{V}^{(i)}}(\mathbf{X}) - \mathsf{A}_{\mathbf{W}^{(i)}, \mathbf{V}^{(i)}}(\bar{\mathbf{X}}) \right) \right\|_{2,1} \tag{103}$$

$$\leq \sum_{i=1}^{h} \|\mathbf{H}^{(i)}\| \|\mathsf{A}_{\mathbf{W}^{(i)}, \mathbf{V}^{(i)}}(\mathbf{X}) - \mathsf{A}_{\mathbf{W}^{(i)}, \mathbf{V}^{(i)}}(\bar{\mathbf{X}})\|_{2,1} \leq \Lambda h \lambda_X(\xi) \|\mathbf{X} - \bar{\mathbf{X}}\|_{2,1}. \tag{104}$$

Thus, the stability of multi-headed attention with respect to its input is preserved as with a single head, but with additional constant factors.

Furthermore, in terms of Lipschitz-ness with respect to its parameters, such as $\mathbf{W}$, we can see that

$$\|\mathsf{MHA}_{\{\mathbf{W}^{(i)},\mathbf{V}^{(i)},\mathbf{H}^{(i)}\}_{i\in[h]}}(\mathbf{X}) - \mathsf{MHA}_{\{\bar{\mathbf{W}}^{(i)},\mathbf{V}^{(i)},\mathbf{H}^{(i)}\}_{i\in[h]}}(\mathbf{X})\|_{2,1} \tag{105}$$

$$= \left\|\sum_{i=1}^{h}\mathbf{H}^{(i)}\left(\mathsf{A}_{\mathbf{W}^{(i)},\mathbf{V}^{(i)}}(\mathbf{X}) - \mathsf{A}_{\bar{\mathbf{W}}^{(i)},\mathbf{V}^{(i)}}(\mathbf{X})\right)\right\|_{2,1} \tag{106}$$

$$\leq \sum_{i=1}^{h}\|\mathbf{H}^{(i)}\|\|\mathsf{A}_{\mathbf{W}^{(i)},\mathbf{V}^{(i)}}(\mathbf{X}) - \mathsf{A}_{\bar{\mathbf{W}}^{(i)},\mathbf{V}^{(i)}}(\mathbf{X})\|_{2,1} \tag{107}$$

$$\leq \Lambda\lambda_W(\xi)\sum_{i\in[h]}\|\mathbf{W}^{(i)} - \bar{\mathbf{W}}^{(i)}\|, \tag{108}$$

where we utilize equation (101) for the $\mathbf{W}^{(i)}$ parameters for each of the heads. This shows us that we can establish results for multi-headed attention analogous to those we study for single head attention. The driving factors continue to be $\lambda_X(\xi)$ and $\lambda_W(\xi)$ which are tied to the properties of the masked softmax functions. However, both the terms get multiplicatively magnified with increasing number of heads, and thus any improvement in the stability of the masked softmax function will get more pronounced as the number of heads increase. This intuition is supported by our results in figure 3b.

## G   Role of Sparse Softmax: Technical Details

### G.1   Standard Softmax based Attention

**Lemma 4** (adapted from Li et al. [36] Lemma B.1). *For any* $\mathbf{z}, \bar{\mathbf{z}} \in \mathbb{R}^L$ *with*

$$\max_{i,j\in[L]} z_i - z_j \leq \delta, \quad \text{and} \quad \max_{i,j\in[L]} \bar{z}_i - \bar{z}_j \leq \delta, \tag{109}$$

*for a positive constant* $\delta > 0$*, we have the following:*

$$\|\mathsf{softmax}(\mathbf{z})\|_\infty \leq \frac{e^\delta}{L}, \quad \|\mathsf{softmax}(\mathbf{z}) - \mathsf{softmax}(\bar{\mathbf{z}})\|_1 \leq \frac{e^\delta}{L}\|\mathbf{z} - \bar{\mathbf{z}}\|_1. \tag{110}$$

*Proof.* For any $\mathbf{z} \in \mathbb{R}^L$, without loss of generality, let the first entry $z_1$ be the largest, and the second entry $z_2$ be the smallest. By equation (109), $z_1 - z_2 \leq \delta$. With $\mathbf{s} = \mathsf{softmax}(\mathbf{z})$, the first entry $s_1$ will be the largest. Thus:

$$\|\mathsf{softmax}(\mathbf{z})\|_\infty = s_1 = \frac{\exp(z_1)}{\exp(z_1) + \sum_{i=2}^{L}\exp(z_i)} \tag{111}$$

$$\leq \frac{\exp(z_1)}{\exp(z_1) + \sum_{i=2}^{L}\exp(z_2)} = \frac{\exp(z_1 - z_2)}{\exp(z_1 - z_2) + (L-1)} \leq \frac{\exp(\delta)}{L}. \tag{112}$$

Now we can write $\mathsf{softmax}(\mathbf{z}) - \mathsf{softmax}(\bar{\mathbf{z}})$ as an aggregation of infinitesimal steps along the gradient of the softmax in the direction $\bar{\mathbf{z}} - \mathbf{z}$:

$$\mathsf{softmax}(\mathbf{z}) - \mathsf{softmax}(\bar{\mathbf{z}}) = \int_0^1 \nabla_\varepsilon \mathsf{softmax}(\mathbf{z} + \varepsilon(\bar{\mathbf{z}} - \mathbf{z}))d\varepsilon \tag{113}$$

$$\|\mathsf{softmax}(\mathbf{z}) - \mathsf{softmax}(\bar{\mathbf{z}})\|_1 \leq \|\int_0^1 \nabla_\varepsilon \mathsf{softmax}(\mathbf{z} + \varepsilon(\bar{\mathbf{z}} - \mathbf{z}))d\varepsilon\|_1 \tag{114}$$

$$\leq \int_0^1 \|\nabla_\varepsilon \mathsf{softmax}(\mathbf{z} + \varepsilon(\bar{\mathbf{z}} - \mathbf{z}))\|_1 d\varepsilon. \tag{115}$$

Considering the $\|\nabla_\varepsilon \mathsf{softmax}(\mathbf{z} + \varepsilon(\bar{\mathbf{z}} - \mathbf{z}))\|_1$ term, and denoting $\mathbf{z}(\varepsilon) = \mathbf{z} + \varepsilon(\bar{\mathbf{z}} - \mathbf{z})$ and $\mathbf{s}(\varepsilon) = \mathsf{softmax}(\mathbf{z}(\varepsilon))$, we have

$$\|\nabla_\varepsilon \mathsf{softmax}(\mathbf{z}(\varepsilon))\|_1 = \|[\mathsf{diag}(\mathbf{s}(\varepsilon)) - \mathbf{s}(\varepsilon)\mathbf{s}(\varepsilon)^\top](\bar{\mathbf{z}} - \mathbf{z})\|_1 \tag{116}$$

$$= \sum_{i=1}^{L} \left| (s(\varepsilon)_i - s(\varepsilon)_i^2)(\bar{z}_i - z_i) - \sum_{j \in [L], j \neq i} s(\varepsilon)_i s(\varepsilon)_j (\bar{z}_j - z_j) \right| \tag{117}$$

$$\leq \sum_{i=1}^{L} |s(\varepsilon)_i (\bar{z}_i - z_i)|, \tag{118}$$

since all the $s(\varepsilon)_i, s(\varepsilon)_j \in [0, 1]$. Noting that $s(\varepsilon)_i \leq \|\mathsf{softmax}(\mathbf{s}(\varepsilon)\|_\infty \leq \exp(\delta)/L$, we have

$$\|\nabla_\varepsilon \mathsf{softmax}(\mathbf{z}(\varepsilon))\|_1 \leq \sum_{i=1}^{L} |(\exp(\delta)/L)(\bar{z}_i - z_i)| = \frac{\exp(\delta)}{L} \|\mathbf{z} - \bar{\mathbf{z}}\|_1. \tag{119}$$

Thus

$$\|\mathsf{softmax}(\mathbf{z}) - \mathsf{softmax}(\bar{\mathbf{z}})\|_1 \leq \int_0^1 \|\nabla_\varepsilon \mathsf{softmax}(\mathbf{z} + \varepsilon(\bar{\mathbf{z}} - \mathbf{z}))\|_1 d\varepsilon \tag{120}$$

$$\leq \int_0^1 \frac{\exp(\delta)}{L} \|\mathbf{z} - \bar{\mathbf{z}}\|_1 d\varepsilon = \frac{\exp(\delta)}{L} \|\mathbf{z} - \bar{\mathbf{z}}\|_1, \tag{121}$$

thus giving us equation (110). $\qquad\square$

**Theorem 7.** *Consider the self-attention operation* $\mathsf{A} : \mathbb{R}^{d \times L} \to \mathbb{R}^{d \times L}$ *with input* $\mathbf{X}$ *of* $L$ *token representations and parameters* $\mathbf{W}, \mathbf{V} \in \mathbb{R}^{d \times d}$. *Consider the following assumptions:*

- *(S1) The per-token Euclidean norms are bounded as* $\|\mathbf{X}_{:i}\| \leq \Xi \forall i \in [L]$, *and the parameter norms are bounded at* $\|\mathbf{W}\| \leq \Gamma$ *and* $\|\mathbf{V}\| \leq \Upsilon$.
- *(S2) The per-query semantic dispersion (definition 2) is bounded by* $\delta_s$, *that is:*

$$\forall i \in [L], \max_{j,j' \in [L]} (\mathbf{X}_{:j}^\top \mathbf{W} \mathbf{X}_{:i} - \mathbf{X}_{:j'}^\top \mathbf{W} \mathbf{X}_{:i}) \leq \delta_s. \tag{122}$$

*Then the standard* $\mathsf{softmax}$ *is* $\xi_s$-*Lipschitz with* $\xi_s = e^{\delta_s}/L$, *and the standard attention is Lipschitz with respect to its input and parameters as following for any input pair* $\mathbf{X}, \bar{\mathbf{X}} \in \mathbb{R}^{d \times L}$ *with* $\|\bar{\mathbf{X}}_{:i}\| \leq 1 \forall i \in [L]$, *and parameter pairs* $\mathbf{W}, \bar{\mathbf{W}}, \mathbf{V}, \bar{\mathbf{V}} \in \mathbb{R}^{d \times d}$ *with* $\|\mathbf{W}\| \leq \Gamma, \|\bar{\mathbf{W}}\| \leq \Gamma$ *and* $\|\mathbf{V}\| \leq \Upsilon, \|\bar{\mathbf{V}}\| \leq \Upsilon$:

$$\|\mathsf{A}_{\mathbf{W},\mathbf{V}}(\mathbf{X}) - \mathsf{A}_{\mathbf{W},\mathbf{V}}(\bar{\mathbf{X}})\|_{2,1} \leq \xi_s \Upsilon L (2\Gamma\Xi^2 + 1) \|\mathbf{X} - \bar{\mathbf{X}}\|_{2,1}, \tag{123}$$

$$\|\mathsf{A}_{\mathbf{W},\mathbf{V}}(\mathbf{X}) - \mathsf{A}_{\bar{\mathbf{W}},\mathbf{V}}(\mathbf{X})\|_{2,1} \leq \xi_s \Upsilon L^2 \Xi^3 \|\mathbf{W} - \bar{\mathbf{W}}\|, \tag{124}$$

$$\|\mathsf{A}_{\mathbf{W},\mathbf{V}}(\mathbf{X}) - \mathsf{A}_{\mathbf{W},\bar{\mathbf{V}}}(\mathbf{X})\|_{2,1} \leq L\Xi \|\mathbf{V} - \bar{\mathbf{V}}\|. \tag{125}$$

*Proof.* Now, given the upper bound on the per-query semantic dispersion $\delta_s$ in equation (122), we can apply Lemma 4 with $\delta = \delta_s$, giving us a $\xi_s$-Lipschitz softmax with $\xi_s = \exp(\delta_s)/L$.

Next, we can show equation (123) utilizing lemma 4 and adapting Li et al. [36, Lemma B.2].

$$\|\mathsf{A}_{\mathbf{W},\mathbf{V}}(\mathbf{X}) - \mathsf{A}_{\mathbf{W},\mathbf{V}}(\bar{\mathbf{X}})\|_{2,1} = \|\mathbf{V}\mathbf{X}\mathsf{softmax}(\mathbf{X}^\top \mathbf{W}\mathbf{X}) - \mathbf{V}\bar{\mathbf{X}}\mathsf{softmax}(\bar{\mathbf{X}}^\top \mathbf{W}\bar{\mathbf{X}})\|_{2,1} \tag{126}$$

$$\leq \|\mathbf{V}\mathbf{X}\mathsf{softmax}(\mathbf{X}^\top \mathbf{W}\mathbf{X}) - \mathbf{V}\bar{\mathbf{X}}\mathsf{softmax}(\mathbf{X}^\top \mathbf{W}\mathbf{X})\|_{2,1} \tag{$A_1$}$$

$$+ \|\mathbf{V}\bar{\mathbf{X}}\mathsf{softmax}(\mathbf{X}^\top \mathbf{W}\mathbf{X}) - \mathbf{V}\bar{\mathbf{X}}\mathsf{softmax}(\mathbf{X}^\top \mathbf{W}\bar{\mathbf{X}})\|_{2,1} \tag{$A_2$}$$

$$+ \|\mathbf{V}\bar{\mathbf{X}}\mathsf{softmax}(\mathbf{X}^\top \mathbf{W}\bar{\mathbf{X}}) - \mathbf{V}\bar{\mathbf{X}}\mathsf{softmax}(\bar{\mathbf{X}}^\top \mathbf{W}\bar{\mathbf{X}})\|_{2,1}. \tag{$A_3$}$$

We will handle each of the equation ($A_1$), equation ($A_2$), and equation ($A_3$) individually. We will use $a_{ji}$ to denote the $j$-th entry of $\mathsf{softmax}(\mathbf{X}^\top \mathbf{W}\mathbf{X}_{:i})$, and $\mathsf{a}_{ji}, \bar{\mathsf{a}}_{ji}$ to denote the $j$-th entry

of softmax$(\mathbf{X}^\top \mathbf{W}\bar{\mathbf{X}}_{:i})$ and softmax$(\bar{\mathbf{X}}^\top \mathbf{W}\bar{\mathbf{X}}_{:i})$ respectively. Note that, by lemma 4 and equation (122), all $a_{ji}, \mathsf{a}_{ji}, \bar{\mathsf{a}}_{ji} \leq \xi_s = \exp(\delta_s)/L$.

$$(A_1) = \|\mathbf{V}(\mathbf{X} - \bar{\mathbf{X}})\text{softmax}(\mathbf{X}^\top \mathbf{W}\mathbf{X})\|_{2,1} = \sum_{i=1}^{L} \|\mathbf{V}(\mathbf{X} - \bar{\mathbf{X}})\text{softmax}(\mathbf{X}^\top \mathbf{W}\mathbf{X}_{:i})\| \quad (127)$$

$$= \sum_{i=1}^{L} \|\mathbf{V}\sum_{j=1}^{L}(\mathbf{X}_{:j} - \bar{\mathbf{X}}_{:j})a_{ji}\| \quad (128)$$

$$\leq \|\mathbf{V}\|\sum_{i=1}^{L}\|\sum_{j=1}^{L}(\mathbf{X}_{:j} - \bar{\mathbf{X}}_{:j})a_{ji}\| \leq \|\mathbf{V}\|\sum_{i=1}^{L}\sum_{j=1}^{L}\|\mathbf{X}_{:j} - \bar{\mathbf{X}}_{:j}\||a_{ji}| \quad (129)$$

$$\leq \Upsilon\xi_s\sum_{i=1}^{L}\sum_{j=1}^{L}\|\mathbf{X}_{:j} - \bar{\mathbf{X}}_{:j}\| = \Upsilon\xi_s\sum_{i=1}^{L}\|\mathbf{X} - \bar{\mathbf{X}}\|_{2,1} = \Upsilon\xi_s L\|\mathbf{X} - \bar{\mathbf{X}}\|_{2,1}, \quad (130)$$

where we utilize the fact that $\|\mathbf{V}\| \leq \Upsilon$.

$$(A_2) = \|\mathbf{V}\bar{\mathbf{X}}\left[\text{softmax}(\mathbf{X}^\top \mathbf{W}\mathbf{X}) - \text{softmax}(\mathbf{X}^\top \mathbf{W}\bar{\mathbf{X}})\right]\|_{2,1} \quad (131)$$

$$= \sum_{i=1}^{L}\|\mathbf{V}\bar{\mathbf{X}}[\text{softmax}(\mathbf{X}^\top \mathbf{W}\mathbf{X}_{:i}) - \text{softmax}(\mathbf{X}^\top \mathbf{W}\bar{\mathbf{X}}_{:i})]\| \quad (132)$$

$$= \sum_{i=1}^{L}\|\mathbf{V}\sum_{j=1}^{L}\bar{\mathbf{X}}_{:j}(a_{ji} - \mathsf{a}_{ji})\| \quad (133)$$

$$\leq \|\mathbf{V}\|\sum_{i=1}^{L}\sum_{j=1}^{L}\|\bar{\mathbf{X}}_{:j}\||a_{ji} - \mathsf{a}_{ji}| \leq \Upsilon\Xi\sum_{i=1}^{L}\sum_{j=1}^{L}|a_{ji} - \mathsf{a}_{ji}| \quad (134)$$

$$= \Upsilon\Xi\sum_{i=1}^{L}\|\text{softmax}(\mathbf{X}^\top \mathbf{W}\mathbf{X}_{:i}) - \text{softmax}(\mathbf{X}^\top \mathbf{W}\bar{\mathbf{X}}_{:i})\|_1 \quad (135)$$

$$\leq \Upsilon\Xi\xi_s\sum_{i=1}^{L}\|\mathbf{X}^\top \mathbf{W}(\mathbf{X}_{:i} - \bar{\mathbf{X}}_{:i})\|_1 = \Upsilon\Xi\xi_s\sum_{i=1}^{L}\sum_{j=1}^{L}|\mathbf{X}_{:j}^\top \mathbf{W}(\mathbf{X}_{:i} - \bar{\mathbf{X}}_{:i})| \quad (136)$$

$$\leq \Upsilon\Xi\xi_s\sum_{i=1}^{L}\sum_{j=1}^{L}\|\mathbf{X}_{:j}^\top \mathbf{W}\|\|\mathbf{X}_{:i} - \bar{\mathbf{X}}_{:i}\| = \Upsilon\Xi\xi_s\sum_{i=1}^{L}\|\mathbf{X}_{:i} - \bar{\mathbf{X}}_{:i}\|\left(\sum_{j=1}^{L}\|\mathbf{X}_{:j}^\top \mathbf{W}\|\right) \quad (137)$$

$$\leq \Upsilon\Xi\xi_s\sum_{i=1}^{L}\|\mathbf{X}_{:i} - \bar{\mathbf{X}}_{:i}\|\|\mathbf{W}\|(\sum_{j=1}^{L}\|\mathbf{X}_{:j}\|) \quad (138)$$

$$\leq \Upsilon\Xi^2\xi_s\Gamma L\sum_{i=1}^{L}\|\mathbf{X}_{:i} - \bar{\mathbf{X}}_{:i}\| = \Upsilon\Xi^2\xi_s\Gamma L\|\mathbf{X} - \bar{\mathbf{X}}\|_{2,1}, \quad (139)$$

utilizing equation (110) and the assumption that $\|\mathbf{W}\| \leq \Gamma$ and $\|\mathbf{X}_{:i}\| \leq \Xi$ for all $i \in [L]$.

$$(A_3) = \|\mathbf{V}\bar{\mathbf{X}}\left[\text{softmax}(\mathbf{X}^\top \mathbf{W}\bar{\mathbf{X}}) - \text{softmax}(\bar{\mathbf{X}}^\top \mathbf{W}\bar{\mathbf{X}})\right]\|_{2,1} \quad (140)$$

$$= \sum_{i=1}^{L}\|\mathbf{V}\bar{\mathbf{X}}[\text{softmax}(\mathbf{X}^\top \mathbf{W}\bar{\mathbf{X}}_{:i}) - \text{softmax}(\bar{\mathbf{X}}^\top \mathbf{W}\bar{\mathbf{X}}_{:i})]\| \quad (141)$$

$$= \sum_{i=1}^{L}\|\mathbf{V}\sum_{j=1}^{L}\bar{\mathbf{X}}_{:j}(\mathsf{a}_{ji} - \bar{\mathsf{a}}_{ji})\| \quad (142)$$

$$\leq \|\mathbf{V}\| \sum_{i=1}^{L} \sum_{j=1}^{L} \|\bar{\mathbf{X}}_{:j}\| |\mathsf{a}_{ji} - \bar{\mathsf{a}}_{ji}| \leq \Upsilon \Xi \sum_{i=1}^{L} \sum_{j=1}^{L} |\mathsf{a}_{ji} - \bar{\mathsf{a}}_{ji}| \tag{143}$$

$$= \Upsilon \Xi \sum_{i=1}^{L} \|\mathsf{softmax}(\mathbf{X}^\top \mathbf{W} \bar{\mathbf{X}}_{:i}) - \mathsf{softmax}(\bar{\mathbf{X}}^\top \mathbf{W} \bar{\mathbf{X}}_{:i})\|_1 \tag{144}$$

$$\leq \Upsilon \Xi \xi_s \sum_{i=1}^{L} \|\mathbf{X}^\top \mathbf{W} \bar{\mathbf{X}}_{:i} - \bar{\mathbf{X}}^\top \mathbf{W} \bar{\mathbf{X}}_{:i}\|_1 = \Upsilon \Xi \xi_s \sum_{i=1}^{L} \sum_{j=1}^{L} |(\mathbf{X}_{:j} - \bar{\mathbf{X}}_{:j})^\top \mathbf{W} \bar{\mathbf{X}}_{:i}| \tag{145}$$

$$\leq \Upsilon \Xi \xi_s \sum_{i=1}^{L} \sum_{j=1}^{L} \|\mathbf{X}_{:j} - \bar{\mathbf{X}}_{:j}\| \|\mathbf{W} \bar{\mathbf{X}}_{:i}\| = \Upsilon \Xi \xi_s \sum_{i=1}^{L} \|\mathbf{W} \bar{\mathbf{X}}_{:i}\| \left( \sum_{j=1}^{L} \|\mathbf{X}_{:j} - \bar{\mathbf{X}}_{:j}\| \right) \tag{146}$$

$$= \Upsilon \Xi \xi_s \sum_{i=1}^{L} \|\mathbf{W} \bar{\mathbf{X}}_{:i}\| \|\mathbf{X} - \bar{\mathbf{X}}\|_{2,1} \tag{147}$$

$$\leq \Upsilon \Xi \xi_s \|\mathbf{W}\| \|\mathbf{X} - \bar{\mathbf{X}}\|_{2,1} \sum_{i=1}^{L} \|\bar{\mathbf{X}}_{:i}\| \leq \Upsilon \Xi^2 \xi_s \Gamma L \|\mathbf{X} - \bar{\mathbf{X}}\|_{2,1} \tag{148}$$

Combining the individual bounds on equation ($A_1$), equation ($A_2$), and equation ($A_3$), we have the following bound as per equation (123):

$$\|\mathsf{A}_{\mathbf{W},\mathbf{V}}(\mathbf{X}) - \mathsf{A}_{\mathbf{W},\mathbf{V}}(\bar{\mathbf{X}})\|_{2,1} \leq \xi_s \Upsilon L (2\Gamma \Xi^2 + 1) \|\mathbf{X} - \bar{\mathbf{X}}\|_{2,1}, \tag{149}$$

For equation (124), we note the following:

$$\|\mathsf{A}_{\mathbf{W},\mathbf{V}}(\mathbf{X}) - \mathsf{A}_{\bar{\mathbf{W}},\mathbf{V}}(\mathbf{X})\|_{2,1}$$
$$= \|\mathbf{V}\mathbf{X}\mathsf{softmax}(\mathbf{X}^\top \mathbf{W}\mathbf{X}) - \mathbf{V}\mathbf{X}\mathsf{softmax}(\mathbf{X}^\top \bar{\mathbf{W}}\mathbf{X})\|_{2,1} \tag{150}$$

$$= \sum_{i=1}^{L} \|\mathbf{V}\mathbf{X}(\mathsf{softmax}(\mathbf{X}^\top \mathbf{W}\mathbf{X}_{:i}) - \mathsf{softmax}(\mathbf{X}^\top \bar{\mathbf{W}}\mathbf{X}_{:i}))\| \tag{151}$$

$$\leq \|\mathbf{V}\| \sum_{i=1}^{L} \|\mathbf{X}(\mathsf{softmax}(\mathbf{X}^\top \mathbf{W}\mathbf{X}_{:i}) - \mathsf{softmax}(\mathbf{X}^\top \bar{\mathbf{W}}\mathbf{X}_{:i}))\|. \tag{152}$$

Denoting $a_{ji}$ as the $j$-th entry of $\mathsf{softmax}(\mathbf{X}^\top \mathbf{W}\mathbf{X}_{:i})$ and $\bar{a}_{ji}$ as the $j$-th entry of $\mathsf{softmax}(\mathbf{X}^\top \bar{\mathbf{W}}\mathbf{X}_{:i})$, and using the assumption that $\|\mathbf{V}\| \leq \Upsilon$, we have

$$\|\mathsf{A}_{\mathbf{W},\mathbf{V}}(\mathbf{X}) - \mathsf{A}_{\bar{\mathbf{W}},\mathbf{V}}(\mathbf{X})\|_{2,1} \leq \Upsilon \sum_{i=1}^{L} \| \sum_{j=1}^{L} (a_{ji} - \bar{a}_{ji})\mathbf{X}_{:j}\| \leq \Upsilon \sum_{i=1}^{L} \sum_{j=1}^{L} \|(a_{ji} - \bar{a}_{ji})\mathbf{X}_{:j}\| \tag{153}$$

$$\leq \Upsilon \sum_{i=1}^{L} \sum_{j=1}^{L} |a_{ji} - \bar{a}_{ji}| \|\mathbf{X}_{:j}\| \tag{154}$$

$$\leq \Upsilon \Xi \sum_{i=1}^{L} \|\mathsf{softmax}(\mathbf{X}^\top \mathbf{W}\mathbf{X}_{:i}) - \mathsf{softmax}(\mathbf{X}^\top \bar{\mathbf{W}}\mathbf{X}_{:i})\|_1, \tag{155}$$

where we use the assumption that $\|\mathbf{X}_{:j}\| \leq \Xi$. Now, utilizing lemma 4, we have

$$\|\mathsf{A}_{\mathbf{W},\mathbf{V}}(\mathbf{X}) - \mathsf{A}_{\bar{\mathbf{W}},\mathbf{V}}(\mathbf{X})\|_{2,1} \leq \Upsilon \Xi \sum_{i=1}^{L} \xi_s \|\mathbf{X}^\top \mathbf{W}\mathbf{X}_{:i} - \mathbf{X}^\top \bar{\mathbf{W}}\mathbf{X}_{:i}\|_1 \tag{156}$$

$$= \Upsilon\Xi\xi_s \sum_{i=1}^{L} \sum_{j=1}^{L} |\mathbf{X}_{:j}^\top \mathbf{W}\mathbf{X}_{:i} - \mathbf{X}_{:j}^\top \bar{\mathbf{W}}\mathbf{X}_{:i}| \tag{157}$$

$$\leq \Upsilon\Xi\xi_s \sum_{i=1}^{L} \sum_{j=1}^{L} \|\mathbf{X}_{:j}^\top \mathbf{W} - \mathbf{X}_{:j}^\top \bar{\mathbf{W}}\|\|\mathbf{X}_{:i}\| \tag{158}$$

$$\leq \Upsilon\Xi^2\xi_s \sum_{i=1}^{L} \sum_{j=1}^{L} \|\mathbf{X}_{:j}^\top \mathbf{W} - \mathbf{X}_{:j}^\top \bar{\mathbf{W}}\| \tag{159}$$

$$\leq \Upsilon\Xi^2\xi_s \sum_{i=1}^{L} \sum_{j=1}^{L} \|\mathbf{X}_{:j}\|\|\mathbf{W} - \bar{\mathbf{W}}\| \leq \xi_s L^2\Upsilon\Xi^3\|\mathbf{W} - \bar{\mathbf{W}}\|, \tag{160}$$

where we utilize $\|\mathbf{X}_{:j}\| \leq \Xi$ twice, thus giving us equation (124).

For equation (125), we note that

$$\|\mathsf{A}_{\mathbf{W},\mathbf{V}}(\mathbf{X}) - \mathsf{A}_{\mathbf{W},\bar{\mathbf{V}}}(\mathbf{X})\|_{2,1} = \|\mathbf{V}\mathbf{X}\mathsf{softmax}(\mathbf{X}^\top \mathbf{W}\mathbf{X}) - \bar{\mathbf{V}}\mathbf{X}\mathsf{softmax}(\mathbf{X}^\top \mathbf{W}\mathbf{X})\|_{2,1} \tag{161}$$

$$= \sum_{i=1}^{L} \|(\mathbf{V} - \bar{\mathbf{V}})\mathbf{X}\mathsf{softmax}(\mathbf{X}^\top \mathbf{W}\mathbf{X}_{:i})\| \tag{162}$$

$$\leq \|\mathbf{V} - \bar{\mathbf{V}}\| \sum_{i=1}^{L} \|\mathbf{X}\mathsf{softmax}(\mathbf{X}^\top \mathbf{W}\mathbf{X}_{:i})\|. \tag{163}$$

Noting the fact that $\mathbf{X}\mathsf{softmax}(\mathbf{X}^\top \mathbf{W}\mathbf{X}_{:i})$ is a convex sum of the columns of $\mathbf{X}$, its maximum Euclidean norm is bounded by maximum Euclidean norm of the individual columns, $\max_j \|\mathbf{X}_{:j}\|$, which itself by bounded from above by $\Xi$. This simplifies the right-hand-side above to $L\Xi\|\mathbf{V} - \bar{\mathbf{V}}\|$, giving us equation (125). $\qquad\square$

**Remark 1.** *For the Lipschitz constants in definition 1, $\lambda_X(\xi_s) = \xi_s L\Upsilon(2\Gamma\Xi^2 + 1) = \exp(\delta_s)\Upsilon(2\Gamma\Xi^2 + 1)$, $\lambda_W(\xi_s) = \xi_s\Upsilon L^2\Xi^3 = \exp(\delta_s)\Upsilon L\Xi^3$ and $\lambda_V = L\Xi$ with $\xi_s = \exp(\delta_s)/L$ and $\delta_s$ defined in equation (122). Under the assumption (S2) of theorem 7, $\delta_s \leq 2\Gamma\Xi^2$.*

## G.2 Regular Input-agnostic Sparse Softmax based Attention

**Lemma 5.** *Given a mask $\mathbf{b} \in \{0,1\}^L$ with $k$ nonzeros, define the $i$-th entry of the masked softmax $\mathsf{softmax}_{\mathbf{b}} : \mathbb{R}^L \to S_L$ for an input $\mathbf{z} \in \mathbb{R}^d$ as:*

$$\mathsf{softmax}_{\mathbf{b}}(\mathbf{z})_i = \frac{\exp(z_i)b_i}{\sum_{j=1}^{L} \exp(z_j)b_j}. \tag{164}$$

*Now, for any $\mathbf{z}, \bar{\mathbf{z}} \in \mathbb{R}^L$ with*

$$\max_{i,j\in[L]:b_i=b_j=1} z_i - z_j \leq \delta, \quad \text{and} \quad \max_{i,j\in[L]:b_i=b_j=1} \bar{z}_i - \bar{z}_j \leq \delta, \tag{165}$$

*for a constant $\delta > 0$, we have the following:*

$$\|\mathsf{softmax}_{\mathbf{b}}(\mathbf{z})\|_\infty \leq \frac{e^\delta}{k},$$
$$\|\mathsf{softmax}_{\mathbf{b}}(\mathbf{z}) - \mathsf{softmax}_{\mathbf{b}}(\bar{\mathbf{z}})\|_1 \leq \frac{e^\delta}{k}\|\mathbf{b} \odot (\mathbf{z} - \bar{\mathbf{z}})\|_1 \leq \frac{e^\delta}{k}\|\mathbf{z} - \bar{\mathbf{z}}\|_1, \tag{166}$$

*where $\odot$ denotes the elementwise multiplication of two vectors.*

*Proof.* For any $\mathbf{z}, \bar{\mathbf{z}} \in \mathbb{R}^L$ and a fixed mask $\mathbf{b} \in \{0,1\}^L$ with $k$ nonzeros, let $\mathbf{z}[\mathbf{b}], \bar{\mathbf{z}}[\mathbf{b}] \in \mathbb{R}^k$ denote the $k$-dimensional vectors corresponding to the unmasked entries of $\mathbf{z}, \bar{\mathbf{z}}$. Then, utilizing lemma 4 for a softmax operation over a $k$-length vector with equation (165), we have the following:

$$\|\mathrm{softmax}_{\mathbf{b}}(\mathbf{z})\|_\infty = \|\mathrm{softmax}(\mathbf{z}[\mathbf{b}])\|_\infty \leq \frac{\exp(\delta)}{k}. \tag{167}$$

Furthermore,

$$\|\mathrm{softmax}_{\mathbf{b}}(\mathbf{z}) - \mathrm{softmax}_{\mathbf{b}}(\bar{\mathbf{z}})\|_1 = \|\mathrm{softmax}(\mathbf{z}[\mathbf{b}]) - \mathrm{softmax}(\bar{\mathbf{z}}[\mathbf{b}])\|_1 \tag{168}$$

$$\leq \frac{\exp(\delta)}{k}\|\mathbf{z}[\mathbf{b}] - \bar{\mathbf{z}}[\mathbf{b}]\|_1 \tag{169}$$

$$= \frac{\exp(\delta)}{k}\|\mathbf{b} \odot (\mathbf{z} - \bar{\mathbf{z}})\|_1 \tag{170}$$

$$\leq \frac{\exp(\delta)}{k}\|\mathbf{z} - \bar{\mathbf{z}}\|_1, \tag{171}$$

where the last inequality is from the fact that $\ell_1$ distance between masked vectors is smaller than the $\ell_1$ distance over the full vectors. $\square$

**Theorem 8.** *Consider the self-attention operation* $\mathsf{A} : \mathbb{R}^{d \times L} \to \mathbb{R}^{d \times L}$ *with input* $\mathbf{X}$ *of* $L$ *token representations and parameters* $\mathbf{W}, \mathbf{V} \in \mathbb{R}^{d \times d}$ *utilizing a* $k$*-regular input sparse agnostic masking function* $m : \mathbb{R}^{L \times L} \to \{0,1\}^{L \times L}$ *where* $m(\mathbf{D}) = \mathbf{M} \forall \mathbf{D} \in \mathbb{R}^{L \times L}$. *Consider the following assumptions:*

- *(R1) The per-token Euclidean norms are bounded as* $\|\mathbf{X}_{:i}\| \leq \Xi \forall i \in [L]$, *and the parameter norms are bounded at* $\|\mathbf{W}\| \leq \Gamma$ *and* $\|\mathbf{V}\| \leq \Upsilon$.
- *(R2) The per-query semantic dispersion (definition 2) is bounded by* $\delta_r$, *that is:*

$$\forall i \in [L], \max_{j,j' \in [L], M_{ji} = M_{j'i} = 1} (\mathbf{X}_{:j}^\top \mathbf{W}\mathbf{X}_{:i} - \mathbf{X}_{:j'}^\top \mathbf{W}\mathbf{X}_{:i}) \leq \delta_r. \tag{172}$$

*Then the masked* $\mathrm{softmax}$ *is* $\xi_r$*-Lipschitz with* $\xi_r = e^{\delta_r}/k$, *and the masked attention is Lipschitz with respect to its input and parameters as following for any input pair* $\mathbf{X}, \bar{\mathbf{X}} \in \mathbb{R}^{d \times L}$ *with* $\|\bar{\mathbf{X}}_{:i}\| \leq 1 \forall i \in [L]$, *and parameter pairs* $\mathbf{W}, \bar{\mathbf{W}}, \mathbf{V}, \bar{\mathbf{V}} \in \mathbb{R}^{d \times d}$ *with* $\|\mathbf{W}\| \leq \Gamma, \|\bar{\mathbf{W}}\| \leq \Gamma$ *and* $\|\mathbf{V}\| \leq \Upsilon, \|\bar{\mathbf{V}}\| \leq \Upsilon$:

$$\|\mathsf{A}_{\mathbf{W},\mathbf{V}}(\mathbf{X}) - \mathsf{A}_{\mathbf{W},\mathbf{V}}(\bar{\mathbf{X}})\|_{2,1} \leq \xi_r \Upsilon k(2\Gamma\Xi^2 + 1)\|\mathbf{X} - \bar{\mathbf{X}}\|_{2,1}, \tag{173}$$

$$\|\mathsf{A}_{\mathbf{W},\mathbf{V}}(\mathbf{X}) - \mathsf{A}_{\bar{\mathbf{W}},\mathbf{V}}(\mathbf{X})\|_{2,1} \leq \xi_r \Upsilon Lk\Xi^3\|\mathbf{W} - \bar{\mathbf{W}}\|, \tag{174}$$

$$\|\mathsf{A}_{\mathbf{W},\mathbf{V}}(\mathbf{X}) - \mathsf{A}_{\mathbf{W},\bar{\mathbf{V}}}(\mathbf{X})\|_{2,1} \leq L\Xi\|\mathbf{V} - \bar{\mathbf{V}}\|. \tag{175}$$

*Proof.* Now, given the upper bound on the per-query semantic dispersion $\delta_r$ in equation (172), we can apply Lemma 5 with $\delta = \delta_r$, giving us a $\xi_r$-Lipschitz softmax with $\xi_r = \exp(\delta_r)/k$.

Note that, given a $k$-regular input agnostic masking function $m$ and the corresponding mask matrix $\mathbf{M}$, we know that, for any column $\mathbf{M}_{:i}, i \in [L], \sum_{j=1}^L M_{ji} = k$, and for any row $\mathbf{M}_{i:}, i \in [L], \sum_{j=1}^L M_{ij} = k$ – the mask matrix has $k$ nonzeros in each row and each column. We denote the masked softmax with a mask matrix $\mathbf{M}$ of a dot-product matrix $\mathbf{D} \in \mathbb{R}^{L \times L}$ as $\mathrm{softmax}_{\mathbf{M}}(\mathbf{D})$, defined as the columnwise masked softmax, which itself is denoted as $\mathrm{softmax}_{\mathbf{M}_{:i}}(\mathbf{D}_{:i})$ and defined in equation (164).

For equation (173), we proceed as follows:

$$\|\mathsf{A}_{\mathbf{W},\mathbf{V}}(\mathbf{X}) - \mathsf{A}_{\mathbf{W},\mathbf{V}}(\bar{\mathbf{X}})\|_{2,1}$$

$$= \|\mathbf{V}\mathbf{X}\mathrm{softmax}_{\mathbf{M}}(\mathbf{X}^\top \mathbf{W}\mathbf{X}) - \mathbf{V}\bar{\mathbf{X}}\mathrm{softmax}_{\mathbf{M}}(\bar{\mathbf{X}}^\top \mathbf{W}\bar{\mathbf{X}})\|_{2,1} \tag{176}$$

$$\leq \|\mathbf{V}\mathbf{X}\mathrm{softmax}_{\mathbf{M}}(\mathbf{X}^\top \mathbf{W}\mathbf{X}) - \mathbf{V}\bar{\mathbf{X}}\mathrm{softmax}_{\mathbf{M}}(\mathbf{X}^\top \mathbf{W}\mathbf{X})\|_{2,1} \tag{$B_1$}$$

$$+ \|\mathbf{V}\bar{\mathbf{X}}\mathrm{softmax}_{\mathbf{M}}(\mathbf{X}^\top \mathbf{W}\mathbf{X}) - \mathbf{V}\bar{\mathbf{X}}\mathrm{softmax}_{\mathbf{M}}(\mathbf{X}^\top \mathbf{W}\bar{\mathbf{X}})\|_{2,1} \tag{$B_2$}$$

$$+ \|\mathbf{V}\bar{\mathbf{X}}\mathrm{softmax}_{\mathbf{M}}(\mathbf{X}^\top \mathbf{W}\bar{\mathbf{X}}) - \mathbf{V}\bar{\mathbf{X}}\mathrm{softmax}_{\mathbf{M}}(\bar{\mathbf{X}}^\top \mathbf{W}\bar{\mathbf{X}})\|_{2,1}. \tag{$B_3$}$$

We will handle each of the equation ($B_1$), equation ($B_2$), and equation ($B_3$) individually. We will use $a_{ji}$ to denote the $j$-th entry of masked $\mathsf{softmax}_{\mathbf{M}_{:i}}(\mathbf{X}^\top\mathbf{W}\bar{\mathbf{X}}_{:i})$, and $\mathsf{a}_{ji}$, $\bar{\mathsf{a}}_{ji}$ to denote the $j$-th entry of $\mathsf{softmax}_{\mathbf{M}_{:i}}(\mathbf{X}^\top\mathbf{W}\bar{\mathbf{X}}_{:i})$ and $\mathsf{softmax}_{\mathbf{M}_{:i}}(\bar{\mathbf{X}}^\top\mathbf{W}\bar{\mathbf{X}}_{:i})$ respectively. Note that, by lemma 5 and equation (172), all $a_{ji}, \bar{a}_{ji}, \mathsf{a}_{ji}, \bar{\mathsf{a}}_{ji} \le \xi_r = \exp(\delta_r)/k$.

$$(B_1) = \|\mathbf{V}(\mathbf{X} - \bar{\mathbf{X}})\mathsf{softmax}_{\mathbf{M}}(\mathbf{X}^\top\mathbf{W}\mathbf{X})\|_{2,1} = \sum_{i=1}^{L}\|\mathbf{V}(\mathbf{X} - \bar{\mathbf{X}})\mathsf{softmax}_{\mathbf{M}_{:i}}(\mathbf{X}^\top\mathbf{W}\mathbf{X}_{:i})\| \tag{177}$$

$$= \sum_{i=1}^{L}\|\mathbf{V}\sum_{j=1,M_{ji}=1}^{L}(\mathbf{X}_{:j} - \bar{\mathbf{X}}_{:j})a_{ji}\| \le \|\mathbf{V}\|\sum_{i=1}^{L}\|\sum_{j=1,M_{ji}=1}^{L}(\mathbf{X}_{:j} - \bar{\mathbf{X}}_{:j})a_{ji}\| \tag{178}$$

$$\le \|\mathbf{V}\|\sum_{i=1}^{L}\sum_{j=1,M_{ji}=1}^{L}\|\mathbf{X}_{:j} - \bar{\mathbf{X}}_{:j}\||a_{ji}| \le \Upsilon\xi_r\sum_{i=1}^{L}\sum_{j=1,M_{ji}=1}^{L}\|\mathbf{X}_{:j} - \bar{\mathbf{X}}_{:j}\| \tag{179}$$

$$= \Upsilon\xi_r\sum_{i=1}^{L}\sum_{j=1}^{L}\mathbb{I}(M_{ji} = 1)\|\mathbf{X}_{:j} - \bar{\mathbf{X}}_{:j}\| = \Upsilon\xi_r\sum_{j=1}^{L}\sum_{i=1}^{L}\mathbb{I}(M_{ji} = 1)\|\mathbf{X}_{:j} - \bar{\mathbf{X}}_{:j}\| \tag{180}$$

$$= \Upsilon\xi_r\sum_{j=1}^{L}\|\mathbf{X}_{:j} - \bar{\mathbf{X}}_{:j}\|(\sum_{i=1}^{L}\mathbb{I}(M_{ji} = 1)) \tag{181}$$

$$= \Upsilon\xi_r\sum_{j=1}^{L}\|\mathbf{X}_{:j} - \bar{\mathbf{X}}_{:j}\|k = \Upsilon\xi_r k\|\mathbf{X} - \bar{\mathbf{X}}\|_{2,1} \tag{182}$$

where we utilize the fact that $\|\mathbf{V}\| \le \Upsilon$, and the row sum of the mask matrix is exactly equal to $k$.

$$(B_2) = \|\mathbf{V}\bar{\mathbf{X}}\left[\mathsf{softmax}_{\mathbf{M}}(\mathbf{X}^\top\mathbf{W}\mathbf{X}) - \mathsf{softmax}_{\mathbf{M}}(\mathbf{X}^\top\mathbf{W}\bar{\mathbf{X}})\right]\|_{2,1} \tag{183}$$

$$= \sum_{i=1}^{L}\|\mathbf{V}\bar{\mathbf{X}}[\mathsf{softmax}_{\mathbf{M}_{:i}}(\mathbf{X}^\top\mathbf{W}\mathbf{X}_{:i}) - \mathsf{softmax}_{\mathbf{M}_{:i}}(\mathbf{X}^\top\mathbf{W}\bar{\mathbf{X}}_{:i})]\| \tag{184}$$

$$= \sum_{i=1}^{L}\|\mathbf{V}\sum_{j=1,M_{ji}=1}^{L}\bar{\mathbf{X}}_{:j}(a_{ji} - \mathsf{a}_{ji})\| \le \|\mathbf{V}\|\sum_{i=1}^{L}\sum_{j=1,M_{ji}=1}^{L}\|\bar{\mathbf{X}}_{:j}\||a_{ji} - \mathsf{a}_{ji}| \tag{185}$$

$$\le \Upsilon\Xi\sum_{i=1}^{L}\sum_{j=1,M_{ji}=1}^{L}|a_{ji} - \mathsf{a}_{ji}| \tag{186}$$

$$= \Upsilon\Xi\sum_{i=1}^{L}\|\mathsf{softmax}_{\mathbf{M}_{:i}}(\mathbf{X}^\top\mathbf{W}\mathbf{X}_{:i}) - \mathsf{softmax}_{\mathbf{M}_{:i}}(\mathbf{X}^\top\mathbf{W}\bar{\mathbf{X}}_{:i})\|_1 \tag{187}$$

$$\le \Upsilon\Xi\xi_r\sum_{i=1}^{L}\|\mathbf{M}_{:i} \odot \mathbf{X}^\top\mathbf{W}(\mathbf{X}_{:i} - \bar{\mathbf{X}}_{:i})\|_1 \tag{188}$$

$$= \Upsilon\Xi\xi_r\sum_{i=1}^{L}\sum_{j=1,M_{ji}=1}^{L}|\mathbf{X}_{:j}^\top\mathbf{W}(\mathbf{X}_{:i} - \bar{\mathbf{X}}_{:i})| \tag{189}$$

$$\le \Upsilon\Xi\xi_r\sum_{i=1}^{L}\sum_{j=1,M_{ji}=1}^{L}\|\mathbf{X}_{:j}^\top\mathbf{W}\|\|\mathbf{X}_{:i} - \bar{\mathbf{X}}_{:i}\| \tag{190}$$

$$= \Upsilon\Xi\xi_r\sum_{i=1}^{L}\|\mathbf{X}_{:i} - \bar{\mathbf{X}}_{:i}\|\left(\sum_{j=1,M_{ji}=1}^{L}\|\mathbf{X}_{:j}^\top\mathbf{W}\|\right) \tag{191}$$

$$\leq \Upsilon\Xi\xi_r \sum_{i=1}^{L} \|\mathbf{X}_{:i} - \bar{\mathbf{X}}_{:i}\| \|\mathbf{W}\| \left( \sum_{j=1, M_{ji}=1}^{L} \|\mathbf{X}_{:j}\| \right) \tag{192}$$

$$\leq \Upsilon\Xi^2 k\xi_r \Gamma \sum_{i=1}^{L} \|\mathbf{X}_{:i} - \bar{\mathbf{X}}_{:i}\| = \Upsilon\Xi^2 \xi_r \Gamma k \|\mathbf{X} - \bar{\mathbf{X}}\|_{2,1}, \tag{193}$$

utilizing equation (166), the assumption that $\|\mathbf{W}\| \leq \Gamma$, $\|\mathbf{X}_{:i}\| \leq \Xi$ for all $i \in [L]$, and that the column sum of $\mathbf{M}$ is $k$.

$$(B_3) = \|\mathbf{V}\bar{\mathbf{X}} \left[ \mathsf{softmax}_{\mathbf{M}}(\mathbf{X}^\top \mathbf{W}\bar{\mathbf{X}}) - \mathsf{softmax}_{\mathbf{M}}(\bar{\mathbf{X}}^\top \mathbf{W}\bar{\mathbf{X}}) \right] \|_{2,1} \tag{194}$$

$$= \sum_{i=1}^{L} \|\mathbf{V}\bar{\mathbf{X}}[\mathsf{softmax}_{\mathbf{M}_{:i}}(\mathbf{X}^\top \mathbf{W}\bar{\mathbf{X}}_{:i}) - \mathsf{softmax}_{\mathbf{M}_{:i}}(\bar{\mathbf{X}}^\top \mathbf{W}\bar{\mathbf{X}}_{:i})]\| \tag{195}$$

$$= \sum_{i=1}^{L} \|\mathbf{V} \sum_{j=1, M_{ji}=1}^{L} \bar{\mathbf{X}}_{:j}(\mathsf{a}_{ji} - \bar{\mathsf{a}}_{ji})\| \leq \|\mathbf{V}\| \sum_{i=1}^{L} \sum_{j=1, M_{ji}=1}^{L} \|\bar{\mathbf{X}}_{:j}\| |\mathsf{a}_{ji} - \bar{\mathsf{a}}_{ji}| \tag{196}$$

$$\leq \Upsilon\Xi \sum_{i=1}^{L} \sum_{j=1, M_{ji}=1}^{L} |\mathsf{a}_{ji} - \bar{\mathsf{a}}_{ji}| \tag{197}$$

$$= \Upsilon\Xi \sum_{i=1}^{L} \|\mathsf{softmax}_{\mathbf{M}_{:i}}(\mathbf{X}^\top \mathbf{W}\bar{\mathbf{X}}_{:i}) - \mathsf{softmax}_{\mathbf{M}_{:i}}(\bar{\mathbf{X}}^\top \mathbf{W}\bar{\mathbf{X}}_{:i})\|_1 \tag{198}$$

$$\leq \Upsilon\Xi\xi_r \sum_{i=1}^{L} \|\mathbf{M}_{:i} \odot (\mathbf{X}^\top \mathbf{W}\bar{\mathbf{X}}_{:i} - \bar{\mathbf{X}}^\top \mathbf{W}\bar{\mathbf{X}}_{:i})\|_1 \tag{199}$$

$$= \Upsilon\Xi\xi_r \sum_{i=1}^{L} \sum_{j=1, M_{ji}=1}^{L} |(\mathbf{X}_{:j} - \bar{\mathbf{X}}_{:j})^\top \mathbf{W}\bar{\mathbf{X}}_{:i}| \tag{200}$$

$$\leq \Upsilon\Xi\xi_r \sum_{i=1}^{L} \sum_{j=1, M_{ji}=1}^{L} \|\mathbf{X}_{:j} - \bar{\mathbf{X}}_{:j}\| \|\mathbf{W}\bar{\mathbf{X}}_{:i}\| \tag{201}$$

$$= \Upsilon\Xi\xi_r \sum_{i=1}^{L} \|\mathbf{W}\bar{\mathbf{X}}_{:i}\| \left( \sum_{j=1, M_{ji}=1}^{L} \|\mathbf{X}_{:j} - \bar{\mathbf{X}}_{:j}\| \right) \tag{202}$$

$$\leq \Upsilon\Xi\xi_r \sum_{i=1}^{L} \|\mathbf{W}\| \|\bar{\mathbf{X}}_{:i}\| \left( \sum_{j=1, M_{ji}=1}^{L} \|\mathbf{X}_{:j} - \bar{\mathbf{X}}_{:j}\| \right) \tag{203}$$

$$\leq \Upsilon\Xi^2 \xi_r \|\mathbf{W}\| \sum_{i=1}^{L} \left( \sum_{j=1, M_{ji}=1}^{L} \|\mathbf{X}_{:j} - \bar{\mathbf{X}}_{:j}\| \right) \tag{204}$$

$$\leq \Upsilon\Xi^2 \xi_r \Gamma \sum_{i=1}^{L} \sum_{j=1}^{L} \mathbb{I}(M_{ji} = 1) \|\mathbf{X}_{:j} - \bar{\mathbf{X}}_{:j}\| \tag{205}$$

$$= \Upsilon\Xi^2 \xi_r \Gamma \sum_{j=1}^{L} \|\mathbf{X}_{:j} - \bar{\mathbf{X}}_{:j}\| \left( \sum_{i=1}^{L} \mathbb{I}(M_{ji} = 1) \right) \tag{206}$$

$$= \Upsilon\Xi^2 \xi_r \Gamma \sum_{j=1}^{L} \|\mathbf{X}_{:j} - \bar{\mathbf{X}}_{:j}\| k = \Upsilon\Xi^2 \xi_r \Gamma k \|\mathbf{X} - \bar{\mathbf{X}}\|_{2,1} \tag{207}$$

Combining the individual bounds on equation $(B_1)$, equation $(B_2)$, and equation $(B_3)$, we have the following bound as per equation (173):

$$\|\mathsf{A_{W,V}}(\mathbf{X}) - \mathsf{A_{W,V}}(\bar{\mathbf{X}})\|_{2,1} \leq \xi_r \Upsilon k(2\Gamma\Xi^2 + 1)\|\mathbf{X} - \bar{\mathbf{X}}\|_{2,1}, \tag{208}$$

For equation (174), we note the following:

$$
\begin{aligned}
&\|\mathsf{A_{W,V}}(\mathbf{X}) - \mathsf{A_{\bar{W},V}}(\mathbf{X})\|_{2,1} \\
&= \|\mathbf{VX}\mathsf{softmax_M}(\mathbf{X}^\top\mathbf{WX}) - \mathbf{VX}\mathsf{softmax_M}(\mathbf{X}^\top\bar{\mathbf{W}}\mathbf{X})\|_{2,1} \tag{209} \\
&= \sum_{i=1}^{L}\|\mathbf{VX}(\mathsf{softmax_{M_{:i}}}(\mathbf{X}^\top\mathbf{WX}_{:i}) - \mathsf{softmax_{M_{:i}}}(\mathbf{X}^\top\bar{\mathbf{W}}\mathbf{X}_{:i}))\| \tag{210} \\
&\leq \|\mathbf{V}\|\sum_{i=1}^{L}\|\mathbf{X}(\mathsf{softmax_{M_{:i}}}(\mathbf{X}^\top\mathbf{WX}_{:i}) - \mathsf{softmax_{M_{:i}}}(\mathbf{X}^\top\bar{\mathbf{W}}\mathbf{X}_{:i}))\|. \tag{211}
\end{aligned}
$$

Denoting $a_{ji}$ as the $j$-th entry of masked $\mathsf{softmax_{M_{:i}}}(\mathbf{X}^\top\mathbf{WX}_{:i})$ and $\bar{a}_{ji}$ as the $j$-th entry of the masked $\mathsf{softmax_{M_{:i}}}(\mathbf{X}^\top\bar{\mathbf{W}}\mathbf{X}_{:i})$, and using the assumption that $\|\mathbf{V}\| \leq \Upsilon$, we have

$$
\begin{aligned}
&\|\mathsf{A_{W,V}}(\mathbf{X}) - \mathsf{A_{\bar{W},V}}(\mathbf{X})\|_{2,1} \\
&\leq \Upsilon\sum_{i=1}^{L}\|\sum_{j=1,M_{ji}=1}^{L}(a_{ji} - \bar{a}_{ji})\mathbf{X}_{:j}\| \leq \Upsilon\sum_{i=1}^{L}\sum_{j=1,M_{ji}=1}^{L}\|(a_{ji} - \bar{a}_{ji})\mathbf{X}_{:j}\| \tag{212} \\
&\leq \Upsilon\sum_{i=1}^{L}\sum_{j=1,M_{ji}=1}^{L}|a_{ji} - \bar{a}_{ji}|\|\mathbf{X}_{:j}\| \tag{213} \\
&\leq \Upsilon\Xi\sum_{i=1}^{L}\|\mathsf{softmax_{M_{:i}}}(\mathbf{X}^\top\mathbf{WX}_{:i}) - \mathsf{softmax_{M_{:i}}}(\mathbf{X}^\top\bar{\mathbf{W}}\mathbf{X}_{:i})\|_1, \tag{214}
\end{aligned}
$$

where we use the assumption that $\|\mathbf{X}_{:j}\| \leq 1$. Now, utilizing lemma 5, we have

$$
\begin{aligned}
\|\mathsf{A_{W,V}}(\mathbf{X}) - \mathsf{A_{\bar{W},V}}(\mathbf{X})\|_{2,1} &\leq \Upsilon\Xi\sum_{i=1}^{L}\xi_r\|\mathbf{M}_{:i} \odot (\mathbf{X}^\top\mathbf{WX}_{:i} - \mathbf{X}^\top\bar{\mathbf{W}}\mathbf{X}_{:i})\|_1 \tag{215} \\
&= \Upsilon\Xi\xi_r\sum_{i=1}^{L}\sum_{j=1,M_{ji}=1}^{L}|\mathbf{X}_{:j}^\top\mathbf{WX}_{:i} - \mathbf{X}_{:j}^\top\bar{\mathbf{W}}\mathbf{X}_{:i}| \tag{216} \\
&\leq \Upsilon\Xi\xi_r\sum_{i=1}^{L}\sum_{j=1,M_{ji}=1}^{L}\|\mathbf{X}_{:j}^\top\mathbf{W} - \mathbf{X}_{:j}^\top\bar{\mathbf{W}}\|\|\mathbf{X}_{:i}\| \tag{217} \\
&\leq \Upsilon\Xi^2\xi_r\sum_{i=1}^{L}\sum_{j=1,M_{ji}=1}^{L}\|\mathbf{X}_{:j}^\top\mathbf{W} - \mathbf{X}_{:j}^\top\bar{\mathbf{W}}\| \tag{218} \\
&\leq \Upsilon\Xi^2\xi_r\sum_{i=1}^{L}\sum_{j=1,M_{ji}=1}^{L}\|\mathbf{X}_{:j}\|\|\mathbf{W} - \bar{\mathbf{W}}\| \tag{219} \\
&\leq \xi_r\Xi^3 Lk\Upsilon\|\mathbf{W} - \bar{\mathbf{W}}\|, \tag{220}
\end{aligned}
$$

where we utilize $\|\mathbf{X}_{:j}\| \leq \Xi$ twice, thus giving us equation (174).

For equation (175), we note that

$$\|\mathsf{A_{W,V}}(\mathbf{X}) - \mathsf{A_{W,\bar{V}}}(\mathbf{X})\|_{2,1} = \|\mathbf{VX}\mathsf{softmax_M}(\mathbf{X}^\top\mathbf{WX}) - \bar{\mathbf{V}}\mathbf{X}\mathsf{softmax_M}(\mathbf{X}^\top\mathbf{WX})\|_{2,1}$$

$$\tag{221}$$

$$= \sum_{i=1}^{L} \|(\mathbf{V} - \bar{\mathbf{V}})\mathbf{X}\text{softmax}_{\mathbf{M}_{:i}}(\mathbf{X}^\top \mathbf{W}\mathbf{X}_{:i})\| \tag{222}$$

$$\leq \|\mathbf{V} - \bar{\mathbf{V}}\| \sum_{i=1}^{L} \|\mathbf{X}\text{softmax}_{\mathbf{M}_{:i}}(\mathbf{X}^\top \mathbf{W}\mathbf{X}_{:i})\|. \tag{223}$$

Noting the fact that $\mathbf{X}\text{softmax}_{\mathbf{M}_{:i}}(\mathbf{X}^\top \mathbf{W}\mathbf{X}_{:i})$ is a (sparse) convex sum of the columns of $\mathbf{X}$, its maximum Euclidean norm is bounded by maximum Euclidean norm of the individual columns, $\max_j \|\mathbf{X}_{:j}\|$, which itself by bounded from above by $\Xi$. This simplifies the right-hand-side above to $L\Xi\|\mathbf{V} - \bar{\mathbf{V}}\|$, giving us equation (175). $\qquad\square$

**Remark 2.** *For the Lipschitz constants in definition 1, $\lambda_X(\xi_r) = \xi_r k \Upsilon(2\Gamma\Xi^2 + 1) = \exp(\delta_r)\Upsilon(2\Gamma\Xi^2 + 1)$, $\lambda_W(\xi_r) = \xi_r \Upsilon L k \Xi^3 = \exp(\delta_r)\Upsilon L\Xi^3$ and $\lambda_V = L\Xi$ with $\xi_r = \exp(\delta_r)/k$ and $\delta_r$ defined in equation (172). Under the assumption (R2) of theorem 8, $\delta_r \leq 2\Gamma\Xi^2$.*

**Remark 3.** *Note that, with $k \to L$, which corresponds to standard-softmax based attention, $\delta_r \to \delta_s$, and the results in theorem 8 reduce to the results in theorem 7.*

### G.3 Heavy-hitter Input-dependent Sparse Softmax based Attention

**Lemma 6.** *Given a $k$-heavy-hitter masking function $m : \mathbb{R}^L \to \{0, 1\}$ such that, for any $\mathbf{z} \in \mathbb{R}^L$, with the corresponding mask $\mathbf{b} = m(\mathbf{z})$, the number of nonzeros in $\mathbf{b}$ is exactly $k$, and*

$$\min_{i,j \in [L]: b_i = 1, b_j = 0} z_i - z_j \geq \Delta, \tag{224}$$

*where $\Delta > 0$ denotes the smallest gap between the $k$ heavy-hitter unmasked values in $\mathbf{z}$ and remaining masked values. Furthermore, for any $\mathbf{z}, \bar{\mathbf{z}} \in \mathbb{R}^L$ with corresponding input dependent masks $\mathbf{b} = m(\mathbf{z})$ and $\bar{\mathbf{b}} = m(\bar{\mathbf{z}})$ respectively,*

$$\max_{i,j \in [L]: b_i = b_j = 1} z_i - z_j \leq \delta, \quad \text{and} \quad \max_{i,j \in [L]: \bar{b}_i = \bar{b}_j = 1} \bar{z}_i - \bar{z}_j \leq \delta, \tag{225}$$

*for a constant $\delta > 0$. Denoting the combined masked vector as $\mathbf{c} = \mathbf{b} \vee \bar{\mathbf{b}}$ with $c_i = \mathbb{I}(b_i = 1 \vee \bar{b}_i = 1)$, we have the following:*

$$\|\text{softmax}_{\mathbf{b}}(\mathbf{z})\|_\infty \leq \frac{e^\delta}{k}, \quad \|\text{softmax}_{\mathbf{b}}(\mathbf{z}) - \text{softmax}_{\bar{\mathbf{b}}}(\bar{\mathbf{z}})\|_1 \leq (1 + 1/\Delta)\frac{e^\delta}{k}\|\mathbf{c} \odot (\mathbf{z} - \bar{\mathbf{z}})\|_1 \tag{226}$$

*where $\odot$ denotes the elementwise multiplication of two vectors.*

*Proof.* For any $\mathbf{z}, \bar{\mathbf{z}} \in \mathbb{R}^L$ with input dependent masks $\mathbf{b}, \bar{\mathbf{b}} \in \{0, 1\}^L$ with $k$ nonzeros, let $\mathbf{z}[\mathbf{b}], \bar{\mathbf{z}}[\bar{\mathbf{b}}] \in \mathbb{R}^k$ denote the $k$-dimensional vectors corresponding to the unmasked entries of $\mathbf{z}, \bar{\mathbf{z}}$. Then, utilizing lemma 4 for a softmax operation over a $k$-length vector with equation (165), we have the following:

$$\|\text{softmax}_{\mathbf{b}}(\mathbf{z})\|_\infty = \|\text{softmax}(\mathbf{z}[\mathbf{b}])\|_\infty \leq \frac{\exp(\delta)}{k}. \tag{227}$$

Furthermore,

$$\|\text{softmax}_{\mathbf{b}}(\mathbf{z}) - \text{softmax}_{\bar{\mathbf{b}}}(\bar{\mathbf{z}})\|_1$$
$$\leq \|\text{softmax}_{\mathbf{b}}(\mathbf{z}) - \text{softmax}_{\mathbf{b}}(\bar{\mathbf{z}})\|_1 + \|\text{softmax}_{\mathbf{b}}(\bar{\mathbf{z}}) - \text{softmax}_{\bar{\mathbf{b}}}(\bar{\mathbf{z}})\|_1 \tag{228}$$
$$\leq \frac{\exp(\delta)}{k}\|\mathbf{b} \odot (\mathbf{z} - \bar{\mathbf{z}})\|_1 + \|\text{softmax}_{\mathbf{b}}(\bar{\mathbf{z}}) - \text{softmax}_{\bar{\mathbf{b}}}(\bar{\mathbf{z}})\|_1, \tag{229}$$

where we utilized lemma 5 with the fixed mask $\mathbf{b}$. Now the second term is the masked softmax with two different masks $\mathbf{b}$ and $\bar{\mathbf{b}}$ on the same input $\bar{\mathbf{z}}$. Then the maximum change between the two masked softmax occurs when the entries that go from being masked to being unmasked (or vice versa) – the $i \in [L]$ such that $b_i \otimes \bar{b}_i = 1$ – have the highest values. That is,

$$\|\text{softmax}_{\mathbf{b}}(\bar{\mathbf{z}}) - \text{softmax}_{\bar{\mathbf{b}}}(\bar{\mathbf{z}})\|_1 \leq \sum_{i \in [L]: b_i \otimes \bar{b}_i = 1} |\text{softmax}_{\mathbf{b}}(\bar{\mathbf{z}})_i - \text{softmax}_{\bar{\mathbf{b}}}(\bar{\mathbf{z}})_i| \tag{230}$$

$$\leq \sum_{i \in [L]: b_i \otimes \bar{b}_i = 1} \|\text{softmax}_{\mathbf{b}}(\bar{\mathbf{z}})\|_\infty \tag{231}$$

$$\leq \|\mathbf{b} - \bar{\mathbf{b}}\|_1 \|\text{softmax}_{\mathbf{b}}(\bar{\mathbf{z}})\|_\infty. \tag{232}$$

Let $k' = \|\mathbf{b} - \bar{\mathbf{b}}\|_1$ denote the change in the mask when the input to the mask changes from $\mathbf{z}$ to $\bar{\mathbf{z}}$. Without loss of generality, assume that $\mathbf{z}$ is such that $b_i = 1, \forall i \in [k]$ – that is the first $k$ entries of $\mathbf{z}$ are the heavy-hitters. Similarly, assume that for $\bar{\mathbf{z}}$, the corresponding mask $\bar{\mathbf{b}}$ overlaps with the last $k''$ entries of $\mathbf{b}$ and $\bar{b}_i = 1 \forall i \in [k - k'' + 1, 2k - k'']$. Given that $k' = \|\mathbf{b} - \bar{\mathbf{b}}\|_1 = 2(k - k'')$, we can show that $k'' = (k - k'/2)$.

Note that, by equation (224), we know that there exists thresholds $t, \bar{t} \in \mathbb{R}$ such that

$$\mathbf{z} : \begin{cases} z_i \geq t, i \in [k] \\ z_i \leq t - \Delta, i \in [k + 1, L] \end{cases}, \quad \bar{\mathbf{z}} : \begin{cases} \bar{z}_i \geq \bar{t}, i \in [k - k'' + 1, 2k - k''] \\ z_i \leq \bar{t} - \Delta, i \in [1, k - k''] \cup [2k - k'' + 1, L] \end{cases}.$$

Now we will just consider the first $(2k - k'')$ entries of $\mathbf{z}$ and $\bar{\mathbf{z}}$. We see that

$$(z_i - \bar{z}_i) \begin{cases} \geq (t - \bar{t} + \Delta), i \in [1, k - k''] \\ \leq (-t + \Delta - \bar{t}), i \in [k + 1, 2k - k''] \end{cases} \tag{233}$$

Thus the $\ell_1$ norm between such two $\mathbf{z}$ and $\bar{\mathbf{z}}$ is lower bounded as

$$\|\mathbf{c} \odot (\mathbf{z} - \bar{\mathbf{z}})\|_1 = \sum_{i=1}^{2k-k''} |z_i - \bar{z}_i| \tag{234}$$

$$= \sum_{i=1}^{k-k''} |(z_i - \bar{z}_i)| + \sum_{i=k-k''+1}^{k} |(z_i - \bar{z}_i)| + \sum_{i=k+1}^{2k-k''} |(z_i - \bar{z}_i)| \tag{235}$$

$$\geq \sum_{i=1}^{k-k''} |(z_i - \bar{z}_i)| + \sum_{i=k+1}^{2k-k''} |(z_i - \bar{z}_i)| \tag{236}$$

$$\geq \sum_{i=1}^{k-k''} |(t - \bar{t}) + \Delta| + \sum_{i=k+1}^{2k-k''} |(t - \bar{t}) - \Delta| \tag{237}$$

$$= (k - k'') (|(t - \bar{t}) + \Delta| + |(t - \bar{t}) - \Delta|). \tag{238}$$

Denoting $(t - \bar{t})$ as $\varepsilon$, consider the term $(|\varepsilon + \Delta| + |\varepsilon - \Delta|)$ and note that $\Delta > 0$. We can see that, if $|\varepsilon| \leq \Delta$, the term is equal to $\Delta + |\varepsilon| + \Delta - |\varepsilon| = 2\Delta$. If $|\varepsilon| > \Delta$, then term is equal to $|\varepsilon| + \Delta + |\varepsilon| - \Delta = 2|\varepsilon| > 2\Delta$.

Thus we have

$$\|\mathbf{c} \odot (\mathbf{z} - \bar{\mathbf{z}})\|_1 \geq (k - k'') (|(t - \bar{t}) + \Delta| + |(t - \bar{t}) - \Delta|). \tag{239}$$

$$\geq 2(k - k'')\Delta = k'\Delta = \|\mathbf{b} - \bar{\mathbf{b}}\|_1 \Delta, \tag{240}$$

giving us $\|\mathbf{b} - \bar{\mathbf{b}}\|_1 \leq (1/\Delta)\|\mathbf{c} \odot (\mathbf{z} - \bar{\mathbf{z}})\|_1$. Utilizing this in combination with equation (229) and equation (232), we have

$$\|\text{softmax}_{\mathbf{b}}(\mathbf{z}) - \text{softmax}_{\bar{\mathbf{b}}}(\bar{\mathbf{z}})\|_1 \leq \frac{\exp(\delta)}{k} \|\mathbf{b} \odot (\mathbf{z} - \bar{\mathbf{z}})\|_1 + (1/\Delta)\frac{\exp(\delta)}{k}\|\mathbf{c} \odot (\mathbf{z} - \bar{\mathbf{z}})\|_1$$

$$\tag{241}$$

$$\leq \frac{\exp(\delta)}{k}(1 + 1/\Delta)\|\mathbf{c} \odot (\mathbf{z} - \bar{\mathbf{z}})\|_1, \tag{242}$$

since $\|\mathbf{b} \odot (\mathbf{z} - \bar{\mathbf{z}})\|_1 \leq \|\mathbf{c} \odot (\mathbf{z} - \bar{\mathbf{z}})\|_1$ as $\mathbf{b}$ is contained with $\mathbf{c}$. This gives us the desired result in equation (226). $\qquad\square$

**Theorem 9.** *Consider the self-attention operation* $\mathsf{A} : \mathbb{R}^{d \times L} \to \mathbb{R}^{d \times L}$ *with input* $\mathbf{X}$ *of* $L$ *token representations and parameters* $\mathbf{W}, \mathbf{V} \in \mathbb{R}^{d \times d}$ *utilizing a* $k$*-heavy-hitter input-dependent masking function* $m : \mathbb{R}^L \to \{0, 1\}^L$*, applied columnwise to the dot-product matrix to get a mask matrix* $\mathbf{M} \in \{0, 1\}^{L \times L}$*. Consider the following assumptions:*

- *(H1) For any query-key pairs* $\mathbf{X}, \bar{\mathbf{X}} \in \mathbb{R}^{d \times L}$*, the* $k$*-heavy-hitter mask* $\mathbf{M} = m(\bar{\mathbf{X}}^\top \mathbf{W} \mathbf{X})$ *(applied columnwise) has a minimum per-query semantic separation (definition 3) of* $\Delta_h > 0$*, that is*

$$\forall i \in [L], \min_{j,j' \in [L], M_{ji}=1, M_{j'i}=0} (\bar{\mathbf{X}}_{:j}^\top \mathbf{W} \mathbf{X}_{:i} - \bar{\mathbf{X}}_{:j'}^\top \mathbf{W} \mathbf{X}_{:i}) \geq \Delta_h. \tag{243}$$

- *(H2) A maximum of* $\beta k, \beta > 1$ *query tokens attend to a single key token, that is,* $\|\mathbf{M}_{i:}\|_1 \leq \beta k$ *for any* $i \in [L]$*.*
- *(H3) The per-token Euclidean norms are bounded as* $\|\mathbf{X}_{:i}\| \leq \Xi \forall i \in [L]$*, and the parameter norms are bounded at* $\|\mathbf{W}\| \leq \Gamma$ *and* $\|\mathbf{V}\| \leq \Upsilon$*.*
- *(H4) The per-query semantic dispersion (definition 2) is bounded by* $\delta_h$*, that is:*

$$\forall i \in [L], \max_{j,j' \in [L], M_{ji}=M_{j'i}=1} (\mathbf{X}_{:j}^\top \mathbf{W} \mathbf{X}_{:i} - \mathbf{X}_{:j'}^\top \mathbf{W} \mathbf{X}_{:i}) \leq \delta_h. \tag{244}$$

*Then the masked* softmax *is* $\xi_h$*-Lipschitz with* $\xi_h = (e^{\delta_h}/k)(1 + 1/\Delta_h)$*, and the masked attention is Lipschitz with respect to its input and parameters as following for any input pair* $\mathbf{X}, \bar{\mathbf{X}} \in \mathbb{R}^{d \times L}$ *with* $\|\bar{\mathbf{X}}_{:i}\| \leq 1 \forall i \in [L]$*, and parameter pairs* $\mathbf{W}, \bar{\mathbf{W}}, \mathbf{V}, \bar{\mathbf{V}} \in \mathbb{R}^{d \times d}$ *with* $\|\mathbf{W}\| \leq \Gamma, \|\bar{\mathbf{W}}\| \leq \Gamma$ *and* $\|\mathbf{V}\| \leq \Upsilon, \|\bar{\mathbf{V}}\| \leq \Upsilon$*:*

$$\|\mathsf{A}_{\mathbf{W},\mathbf{V}}(\mathbf{X}) - \mathsf{A}_{\mathbf{W},\mathbf{V}}(\bar{\mathbf{X}})\|_{2,1} \leq \xi_h \Upsilon k \left(2\Gamma\Xi^2(\beta + 1) + \frac{\beta}{1 + 1/\Delta_h}\right) \|\mathbf{X} - \bar{\mathbf{X}}\|_{2,1}, \tag{245}$$

$$\|\mathsf{A}_{\mathbf{W},\mathbf{V}}(\mathbf{X}) - \mathsf{A}_{\bar{\mathbf{W}},\mathbf{V}}(\mathbf{X})\|_{2,1} \leq 2\xi_h \Upsilon L k \Xi^3 \|\mathbf{W} - \bar{\mathbf{W}}\|, \tag{246}$$

$$\|\mathsf{A}_{\mathbf{W},\mathbf{V}}(\mathbf{X}) - \mathsf{A}_{\mathbf{W},\bar{\mathbf{V}}}(\mathbf{X})\|_{2,1} \leq L\Xi \|\mathbf{V} - \bar{\mathbf{V}}\|. \tag{247}$$

*Proof.* Now, given the upper bound on the per-query semantic dispersion $\delta_h$ in equation (244), and the per-query semantic separation $\Delta_h$ in equation (243), we can apply Lemma 6 with $\delta = \delta_h$ and $\Delta = \Delta_h$, giving us a $\xi_h$-Lipschitz softmax with $\xi_h = \exp(\delta_h)(1 + 1/\Delta_h)/k$.

Note that, given a $k$-heavy-hitter input-dependent masking function $m$ and the corresponding mask matrix $\mathbf{M}$, we know that, for any column $\mathbf{M}_{:i}, i \in [L], \sum_{j=1}^{L} M_{ji} = k$. However, unlike the $k$-regular input-agnostic mask, for any row $\mathbf{M}_{i:}, i \in [L], \sum_{j=1}^{L} M_{ij} \neq k$. Here, we will utilize assumption H2 which states that, for any row $\mathbf{M}_{i:}, \sum_{j=1}^{L} M_{ij} \leq \beta k$.

For equation (245), we note that the mask matrix is input-dependent, and thus will denote as mask matrices $\mathbf{M}, \hat{\mathbf{M}}, \bar{\mathbf{M}}$ for the following dot-product matrices $(\mathbf{X}^\top \mathbf{W} \mathbf{X}), (\mathbf{X}^\top \mathbf{W} \bar{\mathbf{X}})$ and $(\bar{\mathbf{X}}^\top \mathbf{W} \bar{\mathbf{X}})$ respectively. That is, $\mathbf{M} = m(\mathbf{X}^\top \mathbf{W} \mathbf{X}), \hat{\mathbf{M}} = m(\mathbf{X}^\top \mathbf{W} \bar{\mathbf{X}}), \bar{\mathbf{M}} = m(\bar{\mathbf{X}}^\top \mathbf{W} \bar{\mathbf{X}})$, where the masking function $m$ is applied columnwise to the dot-product matrices. Given this, we proceed as follows:

$$\begin{aligned}
\|\mathsf{A}_{\mathbf{W},\mathbf{V}}&(\mathbf{X}) - \mathsf{A}_{\mathbf{W},\mathbf{V}}(\bar{\mathbf{X}})\|_{2,1} \\
&= \|\mathbf{V}\mathbf{X}\mathsf{softmax}_{\mathbf{M}}(\mathbf{X}^\top \mathbf{W} \mathbf{X}) - \mathbf{V}\bar{\mathbf{X}}\mathsf{softmax}_{\bar{\mathbf{M}}}(\bar{\mathbf{X}}^\top \mathbf{W} \bar{\mathbf{X}})\|_{2,1} & (248) \\
&\leq \|\mathbf{V}\mathbf{X}\mathsf{softmax}_{\mathbf{M}}(\mathbf{X}^\top \mathbf{W} \mathbf{X}) - \mathbf{V}\bar{\mathbf{X}}\mathsf{softmax}_{\mathbf{M}}(\mathbf{X}^\top \mathbf{W} \mathbf{X})\|_{2,1} & (C_1) \\
&\quad + \|\mathbf{V}\bar{\mathbf{X}}\mathsf{softmax}_{\mathbf{M}}(\mathbf{X}^\top \mathbf{W} \mathbf{X}) - \mathbf{V}\bar{\mathbf{X}}\mathsf{softmax}_{\hat{\mathbf{M}}}(\mathbf{X}^\top \mathbf{W} \bar{\mathbf{X}})\|_{2,1} & (C_2) \\
&\quad + \|\mathbf{V}\bar{\mathbf{X}}\mathsf{softmax}_{\hat{\mathbf{M}}}(\mathbf{X}^\top \mathbf{W} \bar{\mathbf{X}}) - \mathbf{V}\bar{\mathbf{X}}\mathsf{softmax}_{\bar{\mathbf{M}}}(\bar{\mathbf{X}}^\top \mathbf{W} \bar{\mathbf{X}})\|_{2,1}. & (C_3)
\end{aligned}$$

We will handle each of the equation $(C_1)$, equation $(C_2)$, and equation $(C_3)$ individually. We will use $a_{ji}$ to denote the $j$-th entry of masked softmax$_{\mathbf{M}_{:i}}(\mathbf{X}^\top \mathbf{W} \mathbf{X}_{:i})$, and $\mathsf{a}_{ji}, \bar{\mathsf{a}}_{ji}$ to denote the $j$-th entry of softmax$_{\hat{\mathbf{M}}_{:i}}(\mathbf{X}^\top \mathbf{W} \bar{\mathbf{X}}_{:i})$ and softmax$_{\overline{\mathbf{M}}_{:i}}(\bar{\mathbf{X}}^\top \mathbf{W} \mathbf{X}_{:i})$ respectively. Note that, by lemma 6 and equation (244) in assumption H4, all $a_{ji}, \mathsf{a}_{ji}, \bar{\mathsf{a}}_{ji} \leq \exp(\delta_h)/k = \xi_h/(1 + 1/\Delta_h)$.

$$(C_1) = \|\mathbf{V}(\mathbf{X} - \bar{\mathbf{X}})\mathsf{softmax}_{\mathbf{M}}(\mathbf{X}^\top \mathbf{W} \mathbf{X})\|_{2,1} = \sum_{i=1}^{L} \|\mathbf{V}(\mathbf{X} - \bar{\mathbf{X}})\mathsf{softmax}_{\mathbf{M}_{:i}}(\mathbf{X}^\top \mathbf{W} \mathbf{X}_{:i})\| \tag{249}$$

$$= \sum_{i=1}^{L} \|\mathbf{V} \sum_{j=1, M_{ji}=1}^{L} (\mathbf{X}_{:j} - \bar{\mathbf{X}}_{:j}) a_{ji}\| \leq \|\mathbf{V}\| \sum_{i=1}^{L} \| \sum_{j=1, M_{ji}=1}^{L} (\mathbf{X}_{:j} - \bar{\mathbf{X}}_{:j}) a_{ji}\| \tag{250}$$

$$\leq \|\mathbf{V}\| \sum_{i=1}^{L} \sum_{j=1, M_{ji}=1}^{L} \|\mathbf{X}_{:j} - \bar{\mathbf{X}}_{:j}\| |a_{ji}| \leq \Upsilon \frac{\xi_h}{1 + 1/\Delta_h} \sum_{i=1}^{L} \sum_{j=1, M_{ji}=1}^{L} \|\mathbf{X}_{:j} - \bar{\mathbf{X}}_{:j}\| \tag{251}$$

$$= \Upsilon \frac{\xi_h}{1 + 1/\Delta_h} \sum_{i=1}^{L} \sum_{j=1}^{L} \mathbb{I}(M_{ji} = 1) \|\mathbf{X}_{:j} - \bar{\mathbf{X}}_{:j}\| \tag{252}$$

$$= \Upsilon \frac{\xi_h}{1 + 1/\Delta_h} \sum_{j=1}^{L} \sum_{i=1}^{L} \mathbb{I}(M_{ji} = 1) \|\mathbf{X}_{:j} - \bar{\mathbf{X}}_{:j}\| \tag{253}$$

$$= \Upsilon \frac{\xi_h}{1 + 1/\Delta_h} \sum_{j=1}^{L} \|\mathbf{X}_{:j} - \bar{\mathbf{X}}_{:j}\| \left( \sum_{i=1}^{L} \mathbb{I}(M_{ji} = 1) \right) \tag{254}$$

$$\leq \Upsilon \frac{\xi_h}{1 + 1/\Delta_h} \sum_{j=1}^{L} \|\mathbf{X}_{:j} - \bar{\mathbf{X}}_{:j}\| \beta k = \frac{\Upsilon \xi_h \beta k}{1 + 1/\Delta_h} \|\mathbf{X} - \bar{\mathbf{X}}\|_{2,1} \tag{255}$$

where we utilize the fact that $\|\mathbf{V}\| \leq \Upsilon$, and the row sum of the mask matrix is upper bounded by $\beta k$ from assumption H2.

We handle equation $(C_2)$ in the following manner:

$$(C_2) = \|\mathbf{V}\bar{\mathbf{X}} \left[ \mathsf{softmax}_{\mathbf{M}}(\mathbf{X}^\top \mathbf{W} \mathbf{X}) - \mathsf{softmax}_{\hat{\mathbf{M}}}(\mathbf{X}^\top \mathbf{W} \bar{\mathbf{X}}) \right] \|_{2,1} \tag{256}$$

$$= \sum_{i=1}^{L} \|\mathbf{V}\bar{\mathbf{X}}[\mathsf{softmax}_{\mathbf{M}_{:i}}(\mathbf{X}^\top \mathbf{W} \mathbf{X}_{:i}) - \mathsf{softmax}_{\hat{\mathbf{M}}_{:i}}(\mathbf{X}^\top \mathbf{W} \bar{\mathbf{X}}_{:i})]\| \tag{257}$$

$$= \sum_{i=1}^{L} \|\mathbf{V} \sum_{j=1, M_{ji}=1 \vee \hat{M}_{ji}=1}^{L} \bar{\mathbf{X}}_{:j} (a_{ji} - \mathsf{a}_{ji})\| \tag{258}$$

$$\leq \|\mathbf{V}\| \sum_{i=1}^{L} \sum_{j=1, M_{ji}=1 \vee \hat{M}_{ji}=1}^{L} \|\bar{\mathbf{X}}_{:j}\| |a_{ji} - \mathsf{a}_{ji}| \tag{259}$$

$$\leq \Upsilon \Xi \sum_{i=1}^{L} \sum_{j=1, M_{ji}=1 \vee \hat{M}_{ji}=1}^{L} |a_{ji} - \mathsf{a}_{ji}| \tag{260}$$

$$= \Upsilon \Xi \sum_{i=1}^{L} \|\mathsf{softmax}_{\mathbf{M}_{:i}}(\mathbf{X}^\top \mathbf{W} \mathbf{X}_{:i}) - \mathsf{softmax}_{\hat{\mathbf{M}}_{:i}}(\mathbf{X}^\top \mathbf{W} \bar{\mathbf{X}}_{:i})\|_1 \tag{261}$$

$$\leq \Upsilon \Xi \xi_h \sum_{i=1}^{L} \|(\mathbf{M}_{:i} \vee \hat{\mathbf{M}}_{:i}) \odot \mathbf{X}^\top \mathbf{W}(\mathbf{X}_{:i} - \bar{\mathbf{X}}_{:i})\|_1 \tag{262}$$

$$= \Upsilon \Xi \xi_h \sum_{i=1}^{L} \sum_{j=1, M_{ji}=1 \vee \hat{M}_{ji}=1}^{L} |\mathbf{X}_{:j}^\top \mathbf{W}(\mathbf{X}_{:i} - \bar{\mathbf{X}}_{:i})| \tag{263}$$

$$\leq \Upsilon\Xi\xi_h \sum_{i=1}^{L} \sum_{j=1,M_{ji}=1\vee\hat{M}_{ji}=1}^{L} \|\mathbf{X}_{:j}^\top\mathbf{W}\|\|\mathbf{X}_{:i}-\bar{\mathbf{X}}_{:i}\| \tag{264}$$

$$= \Upsilon\Xi\xi_h \sum_{i=1}^{L} \|\mathbf{X}_{:i}-\bar{\mathbf{X}}_{:i}\| \left( \sum_{j=1,M_{ji}=1\vee\hat{M}_{ji}=1}^{L} \|\mathbf{X}_{:j}^\top\mathbf{W}\| \right) \tag{265}$$

$$\leq \Upsilon\Xi\xi_h \sum_{i=1}^{L} \|\mathbf{X}_{:i}-\bar{\mathbf{X}}_{:i}\|\|\mathbf{W}\| \left( \sum_{j=1,M_{ji}=1\vee\hat{M}_{ji}=1}^{L} \|\mathbf{X}_{:j}\| \right) \tag{266}$$

$$\leq \Upsilon\Xi\xi_h\Gamma \sum_{i=1}^{L} \|\mathbf{X}_{:i}-\bar{\mathbf{X}}_{:i}\| \cdot (2k\Xi) = 2\Upsilon\xi_h\Gamma k\Xi^2 \|\mathbf{X}-\bar{\mathbf{X}}\|_{2,1}, \tag{267}$$

utilizing equation (226), the assumption H3 that $\|\mathbf{W}\| \leq \Gamma$, $\|\mathbf{X}_{:i}\| \leq \Xi$ for all $i \in [L]$, and that the column sum of $\mathbf{M}$ is $k$, thus $\sum_{j=1}^{L} \mathbb{I}(M_{ji} = 1 \vee \hat{M}_{ji} = 1) \leq 2k$.

We handle equation $(C_3)$ in the following manner:

$$(C_3) = \|\mathbf{V}\bar{\mathbf{X}}\left[\text{softmax}_{\hat{\mathbf{M}}}(\mathbf{X}^\top\mathbf{W}\bar{\mathbf{X}}) - \text{softmax}_{\bar{\mathbf{M}}}(\bar{\mathbf{X}}^\top\mathbf{W}\bar{\mathbf{X}})\right]\|_{2,1} \tag{268}$$

$$= \sum_{i=1}^{L} \|\mathbf{V}\bar{\mathbf{X}}[\text{softmax}_{\hat{\mathbf{M}}_{:i}}(\mathbf{X}^\top\mathbf{W}\bar{\mathbf{X}}_{:i}) - \text{softmax}_{\bar{\mathbf{M}}_{:i}}(\bar{\mathbf{X}}^\top\mathbf{W}\bar{\mathbf{X}}_{:i})]\| \tag{269}$$

$$= \sum_{i=1}^{L} \|\mathbf{V} \sum_{j=1,\hat{M}_{ji}=1\vee\bar{M}_{ji}=1}^{L} \bar{\mathbf{X}}_{:j}(\mathsf{a}_{ji} - \bar{\mathsf{a}}_{ji})\| \tag{270}$$

$$\leq \|\mathbf{V}\| \sum_{i=1}^{L} \sum_{j=1,\hat{M}_{ji}=1\vee\bar{M}_{ji}=1}^{L} \|\bar{\mathbf{X}}_{:j}\||\mathsf{a}_{ji} - \bar{\mathsf{a}}_{ji}| \tag{271}$$

$$\leq \Upsilon\Xi \sum_{i=1}^{L} \sum_{j=1,\hat{M}_{ji}=1\vee\bar{M}_{ji}=1}^{L} |\mathsf{a}_{ji} - \bar{\mathsf{a}}_{ji}| \tag{272}$$

$$= \Upsilon\Xi \sum_{i=1}^{L} \|\text{softmax}_{\hat{\mathbf{M}}_{:i}}(\mathbf{X}^\top\mathbf{W}\bar{\mathbf{X}}_{:i}) - \text{softmax}_{\bar{\mathbf{M}}_{:i}}(\bar{\mathbf{X}}^\top\mathbf{W}\bar{\mathbf{X}}_{:i})\|_1 \tag{273}$$

$$\leq \Upsilon\Xi\xi_h \sum_{i=1}^{L} \|(\hat{\mathbf{M}}_{:i} \vee \bar{\mathbf{M}}_{:i}) \odot (\mathbf{X}^\top\mathbf{W}\bar{\mathbf{X}}_{:i} - \bar{\mathbf{X}}^\top\mathbf{W}\bar{\mathbf{X}}_{:i})\|_1 \tag{274}$$

$$= \Upsilon\Xi\xi_h \sum_{i=1}^{L} \sum_{j=1,\hat{M}_{ji}=1\vee\bar{M}_{ji}=1}^{L} |(\mathbf{X}_{:j} - \bar{\mathbf{X}}_{:j})^\top\mathbf{W}\bar{\mathbf{X}}_{:i}| \tag{275}$$

$$\leq \Upsilon\Xi\xi_h \sum_{i=1}^{L} \sum_{j=1,\hat{M}_{ji}=1\vee\bar{M}_{ji}=1}^{L} \|\mathbf{X}_{:j} - \bar{\mathbf{X}}_{:j}\|\|\mathbf{W}\bar{\mathbf{X}}_{:i}\| \tag{276}$$

$$= \Upsilon\Xi\xi_h \sum_{i=1}^{L} \|\mathbf{W}\bar{\mathbf{X}}_{:i}\| \left( \sum_{j=1,\hat{M}_{ji}=1\vee\bar{M}_{ji}=1}^{L} \|\mathbf{X}_{:j} - \bar{\mathbf{X}}_{:j}\| \right) \tag{277}$$

$$\leq \Upsilon\Xi\xi_h \sum_{i=1}^{L} \|\mathbf{W}\|\|\bar{\mathbf{X}}_{:i}\| \left( \sum_{j=1,\hat{M}_{ji}=1\vee\bar{M}_{ji}=1}^{L} \|\mathbf{X}_{:j} - \bar{\mathbf{X}}_{:j}\| \right) \tag{278}$$

$$\leq \Upsilon \Xi^2 \xi_h \|\mathbf{W}\| \sum_{i=1}^{L} \left( \sum_{j=1, \hat{M}_{ji}=1 \vee \bar{M}_{ji}=1}^{L} \|\mathbf{X}_{:j} - \bar{\mathbf{X}}_{:j}\| \right) \tag{279}$$

$$\leq \Upsilon \Xi^2 \xi_h \Gamma \sum_{i=1}^{L} \sum_{j=1}^{L} \mathbb{I}(\hat{M}_{ji}=1 \vee \bar{M}_{ji}=1) \|\mathbf{X}_{:j} - \bar{\mathbf{X}}_{:j}\| \tag{280}$$

$$= \Upsilon \Xi^2 \xi_h \Gamma \sum_{j=1}^{L} \|\mathbf{X}_{:j} - \bar{\mathbf{X}}_{:j}\| \left( \sum_{i=1}^{L} \mathbb{I}(\hat{M}_{ji}=1 \vee \bar{M}_{ji}=1) \right) \tag{281}$$

$$\leq \Upsilon \Xi^2 \xi_h \Gamma \sum_{j=1}^{L} \|\mathbf{X}_{:j} - \bar{\mathbf{X}}_{:j}\| \left( \sum_{i=1}^{L} \mathbb{I}(\hat{M}_{ji}=1) + \sum_{i=1}^{L} \mathbb{I}(\bar{M}_{ji}=1) \right) \tag{282}$$

$$\leq \Upsilon \Xi^2 \xi_h \Gamma \sum_{j=1}^{L} \|\mathbf{X}_{:j} - \bar{\mathbf{X}}_{:j}\| \left( \beta k + \beta k \right) \tag{283}$$

$$= 2 \Upsilon \Xi^2 \xi_h \Gamma \beta k \sum_{j=1}^{L} \|\mathbf{X}_{:j} - \bar{\mathbf{X}}_{:j}\| k = 2 \Upsilon \Xi^2 \xi_h \Gamma \beta k \|\mathbf{X} - \bar{\mathbf{X}}\|_{2,1} \tag{284}$$

Combining the individual bounds on equation ($C_1$), equation ($C_2$), and equation ($C_3$), we have the following bound as per equation (245):

$$\|\mathsf{A}_{\mathbf{W},\mathbf{V}}(\mathbf{X}) - \mathsf{A}_{\mathbf{W},\mathbf{V}}(\bar{\mathbf{X}})\|_{2,1} \leq \Upsilon \xi_h k \left( 2\Gamma \Xi^2 (\beta + 1) + \frac{\beta}{1 + 1/\Delta_h} \right) \|\mathbf{X} - \bar{\mathbf{X}}\|_{2,1}. \tag{285}$$

First, let us denote the input-dependent mask matrices as $\mathbf{M}$ and $\bar{\mathbf{M}}$ for the dot-product matrices $(\mathbf{X}^\top \mathbf{W} \mathbf{X})$ and $(\mathbf{X}^\top \bar{\mathbf{W}} \mathbf{X})$ results. Thus $\mathbf{M} = m(\mathbf{X}^\top \mathbf{W} \mathbf{X})$ and $\bar{\mathbf{M}} = m(\mathbf{X}^\top \bar{\mathbf{W}} \mathbf{X})$. Utilizing this for the left-hand-size of equation (246), we note the following:

$$\|\mathsf{A}_{\mathbf{W},\mathbf{V}}(\mathbf{X}) - \mathsf{A}_{\bar{\mathbf{W}},\mathbf{V}}(\mathbf{X})\|_{2,1}$$
$$= \|\mathbf{V}\mathbf{X}\mathsf{softmax}_{\mathbf{M}}(\mathbf{X}^\top \mathbf{W} \mathbf{X}) - \mathbf{V}\mathbf{X}\mathsf{softmax}_{\bar{\mathbf{M}}}(\mathbf{X}^\top \bar{\mathbf{W}} \mathbf{X})\|_{2,1} \tag{286}$$

$$= \sum_{i=1}^{L} \|\mathbf{V}\mathbf{X}(\mathsf{softmax}_{\mathbf{M}_{:i}}(\mathbf{X}^\top \mathbf{W} \mathbf{X}_{:i}) - \mathsf{softmax}_{\bar{\mathbf{M}}_{:i}}(\mathbf{X}^\top \bar{\mathbf{W}} \mathbf{X}_{:i}))\| \tag{287}$$

$$\leq \|\mathbf{V}\| \sum_{i=1}^{L} \|\mathbf{X}(\mathsf{softmax}_{\mathbf{M}_{:i}}(\mathbf{X}^\top \mathbf{W} \mathbf{X}_{:i}) - \mathsf{softmax}_{\bar{\mathbf{M}}_{:i}}(\mathbf{X}^\top \bar{\mathbf{W}} \mathbf{X}_{:i}))\|. \tag{288}$$

Denoting $a_{ji}$ as the $j$-th entry of masked $\mathsf{softmax}_{\mathbf{M}_{:i}}(\mathbf{X}^\top \mathbf{W} \mathbf{X}_{:i})$ and $\bar{a}_{ji}$ as the $j$-th entry of the masked $\mathsf{softmax}_{\bar{\mathbf{M}}_{:i}}(\mathbf{X}^\top \bar{\mathbf{W}} \mathbf{X}_{:i})$, and using the assumption that $\|\mathbf{V}\| \leq \Upsilon$, we have

$$\|\mathsf{A}_{\mathbf{W},\mathbf{V}}(\mathbf{X}) - \mathsf{A}_{\bar{\mathbf{W}},\mathbf{V}}(\mathbf{X})\|_{2,1}$$
$$\leq \Upsilon \sum_{i=1}^{L} \| \sum_{j=1, M_{ji}=1 \vee \bar{M}_{ji}=1}^{L} (a_{ji} - \bar{a}_{ji})\mathbf{X}_{:j}\| \tag{289}$$

$$\leq \Upsilon \sum_{i=1}^{L} \sum_{j=1, M_{ji}=1 \vee \bar{M}_{ji}=1}^{L} \|(a_{ji} - \bar{a}_{ji})\mathbf{X}_{:j}\| \tag{290}$$

$$\leq \Upsilon \sum_{i=1}^{L} \sum_{j=1, M_{ji}=1 \vee \bar{M}_{ji}=1}^{L} |a_{ji} - \bar{a}_{ji}| \|\mathbf{X}_{:j}\| \tag{291}$$

$$\leq \Upsilon\Xi\sum_{i=1}^{L}\|\text{softmax}_{\mathbf{M}_{:i}}(\mathbf{X}^{\top}\mathbf{W}\mathbf{X}_{:i}) - \text{softmax}_{\bar{\mathbf{M}}_{:i}}(\mathbf{X}^{\top}\bar{\mathbf{W}}\mathbf{X}_{:i})\|_1, \tag{292}$$

where we use the assumption that $\|\mathbf{X}_{:j}\| \leq \Xi$. Now, utilizing lemma 6, we have

$$\|\mathsf{A}_{\mathbf{W},\mathbf{V}}(\mathbf{X}) - \mathsf{A}_{\bar{\mathbf{W}},\mathbf{V}}(\mathbf{X})\|_{2,1} \leq \Upsilon\Xi\sum_{i=1}^{L}\xi_h\|(\mathbf{M}_{:i}\vee\bar{\mathbf{M}}_{:i})\odot(\mathbf{X}^{\top}\mathbf{W}\mathbf{X}_{:i} - \mathbf{X}^{\top}\bar{\mathbf{W}}\mathbf{X}_{:i})\|_1 \tag{293}$$

$$= \Upsilon\Xi\xi_h\sum_{i=1}^{L}\sum_{j=1,M_{ji}=1\vee\bar{M}_{ji}=1}^{L}|\mathbf{X}_{:j}^{\top}\mathbf{W}\mathbf{X}_{:i} - \mathbf{X}_{:j}^{\top}\bar{\mathbf{W}}\mathbf{X}_{:i}| \tag{294}$$

$$\leq \Upsilon\Xi\xi_h\sum_{i=1}^{L}\sum_{j=1,M_{ji}=1\vee\bar{M}_{ji}=1}^{L}\|\mathbf{X}_{:j}^{\top}\mathbf{W} - \mathbf{X}_{:j}^{\top}\bar{\mathbf{W}}\|\|\mathbf{X}_{:i}\| \tag{295}$$

$$\leq \Upsilon\Xi^2\xi_h\sum_{i=1}^{L}\sum_{j=1,M_{ji}=1\vee\bar{M}_{ji}=1}^{L}\|\mathbf{X}_{:j}^{\top}\mathbf{W} - \mathbf{X}_{:j}^{\top}\bar{\mathbf{W}}\| \tag{296}$$

$$\leq \Upsilon\Xi^2\xi_h\sum_{i=1}^{L}\sum_{j=1,M_{ji}=1\vee\bar{M}_{ji}=1}^{L}\|\mathbf{X}_{:j}\|\|\mathbf{W} - \bar{\mathbf{W}}\| \tag{297}$$

$$\leq \xi_h\Upsilon\Xi^2\|\mathbf{W} - \bar{\mathbf{W}}\|\sum_{i=1}^{L}\left(\sum_{j=1,M_{ji}=1\vee\bar{M}_{ji}=1}^{L}\|\mathbf{X}_{:j}\|\right), \tag{298}$$

$$\leq \xi_h\Upsilon\Xi^2\|\mathbf{W} - \bar{\mathbf{W}}\|\sum_{i=1}^{L}(2k\Xi), = 2k\xi_h\Upsilon L\Xi^3\|\mathbf{W} - \bar{\mathbf{W}}\|, \tag{299}$$

where we utilize $\|\mathbf{X}_{:j}\| \leq \Xi$ twice, thus giving us equation (246).

Denote the input-dependent mask matrix as $\mathbf{M} = m(\mathbf{X}^{\top}\mathbf{W}\mathbf{X})$ for the dot-product matrix $(\mathbf{X}^{\top}\mathbf{W}\mathbf{X})$, we can express equation (247) as following:

$$\|\mathsf{A}_{\mathbf{W},\mathbf{V}}(\mathbf{X}) - \mathsf{A}_{\mathbf{W},\bar{\mathbf{V}}}(\mathbf{X})\|_{2,1} = \|\mathbf{V}\mathbf{X}\text{softmax}_{\mathbf{M}}(\mathbf{X}^{\top}\mathbf{W}\mathbf{X}) - \bar{\mathbf{V}}\mathbf{X}\text{softmax}_{\mathbf{M}}(\mathbf{X}^{\top}\mathbf{W}\mathbf{X})\|_{2,1}$$

$$\tag{300}$$

$$= \sum_{i=1}^{L}\|(\mathbf{V} - \bar{\mathbf{V}})\mathbf{X}\text{softmax}_{\mathbf{M}_{:i}}(\mathbf{X}^{\top}\mathbf{W}\mathbf{X}_{:i})\| \tag{301}$$

$$\leq \|\mathbf{V} - \bar{\mathbf{V}}\|\sum_{i=1}^{L}\|\mathbf{X}\text{softmax}_{\mathbf{M}_{:i}}(\mathbf{X}^{\top}\mathbf{W}\mathbf{X}_{:i})\|. \tag{302}$$

Noting the fact that $\mathbf{X}\text{softmax}_{\mathbf{M}_{:i}}(\mathbf{X}^{\top}\mathbf{W}\mathbf{X}_{:i})$ is a (sparse) convex sum of the columns of $\mathbf{X}$, its maximum Euclidean norm is bounded by maximum Euclidean norm of the individual columns, $\max_j\|\mathbf{X}_{:j}\|$, which itself by bounded from above by $\Xi$. This simplifies the right-hand-side above to $L\Xi\|\mathbf{V} - \bar{\mathbf{V}}\|$, giving us equation (247). □

**Remark 4.** *For the Lipschitz constants in definition 1,*

$$\lambda_X(\xi_h) = \xi_h\Upsilon k\left(2\Gamma\Xi^2(1+\beta) + \frac{\beta}{(1+1/\Delta_h)}\right)$$

$$= \exp(\delta_h)\Upsilon(1+1/\Delta_h)\left(2\Gamma\Xi^2(1+\beta) + \frac{\beta}{(1+1/\Delta_h)}\right), \tag{303}$$

Table 5: Bounds for $\xi, \lambda_X(\xi), \lambda_W(\xi), \lambda_V$ from definition 1 for different forms of attention. Note that $\lambda_V = L\Xi$ for all forms of attention, and thus elided from this table.

| Attention | $\xi$ | $\lambda_X(\xi)$ | $\lambda_W(\xi)$ |
|---|---|---|---|
| Full (theorem 3) | $\frac{e^{\delta_s}}{L}$ | $e^{\delta_s}\Upsilon(2\Gamma\Xi^2 + 1)$ | $e^{\delta_s}\Upsilon L\Xi^3$ |
| $k$-regular (theorem 4) | $\frac{e^{\delta_r}}{k}$ | $e^{\delta_r}\Upsilon(2\Gamma\Xi^2 + 1)$ | $e^{\delta_r}\Upsilon L\Xi^3$ |
| $k$-heavy-hitter (theorem 5) | $\frac{e^{\delta_h}}{k}(1 + 1/\Delta_h)$ | $e^{\delta_h}\Upsilon\left(\beta + 2\Gamma\Xi^2(\beta+1)(1 + 1/\Delta_h)\right)$ | $2e^{\delta_h}\Upsilon L\Xi^3(1 + 1/\Delta_h)$ |

and $\lambda_W(\xi_h) = 2\xi_h\Upsilon Lk\Xi^3 = 2\exp(\delta_h)\Upsilon L\Xi^3(1 + 1/\Delta_h)$ and $\lambda_V = L\Xi$ with $\xi_h = \exp(\delta_h)(1 + 1/\Delta_h)/k$ and $\delta_h$ defined in equation (244). Under the assumptions (H1) and (H3) of theorem 9, $\delta_h \leq 2\Gamma\Xi^2 - \Delta_h$.

### G.4 Comparison of Bounds between Full and Heavy-hitter Attention

To compare the stability constants for all different forms of attention, we have put them together in table 5. To characterize the conditions when the stability constants for the heavy-hitter sparse attention provides improved guarantees over full attention, we have the following result:

**Corollary 2.** *Consider the definitions and conditions of theorem 3 and theorem 5. Further assume that (i) the maximum per-query semantic dispersion for standard attention is $\delta_s \leq 2\Gamma\Xi^2$, while that of heavy-hitter attention is $\delta_h = c_1\delta_s$, and (ii) the heavy-hitter minimum per-query dot-product separation is $\Delta_h = c_2\delta_s$ for some positive constants $c_1, c_2$. Then $\lambda_W(\xi_h) < \lambda_W(\xi_s)$ when*

$$c_1 + \frac{1}{\delta_s}\log 2\left(1 + \frac{1}{c_2\delta_s}\right) < 1, \tag{304}$$

*and $\lambda_X(\xi_h) < \lambda_X(\xi_s)$ when*

$$c_1 + \frac{1}{\delta_s}\log\left(2\Gamma\Xi^2(1+\beta)\left(1 + \frac{1}{c_2\delta_s}\right) + \beta\right) - \frac{1}{\delta_s}\log(2\Gamma\Xi^2 + 1) < 1. \tag{305}$$

*Proof.* We arrive at equation (304) by comparing $\lambda_W(\xi_s) = \exp(\delta_s)\Upsilon L\Xi^3$ in remark 1 with $\lambda_W(\xi_h) = 2\exp(\delta_h)\Upsilon L\Xi^3(1 + 1/\Delta_h)$. We arrive at equation (305) by comparing $\lambda_X(\xi_s) = \exp(\delta_s)(2\Gamma\Xi^2 + 1)$ in remark 1 with $\lambda_X(\xi_h)$ defined in equation (303) in remark 4. $\square$

Based on this result, we want the constant $c_1$ (corresponding to the semantic dispersion) to be small and the constant $c_2$ (corresponding to the semantic separation) to be large. However, this condition also depends on the full attention dispersion $\delta_s$. We present this relationship for $\lambda_W(\xi_s)$ vs $\lambda_W(\xi_h)$ in figure 21. For small values of $\delta_s$, $c_2$ needs to be quite large and $c_1$ needs to be quite small for $\lambda_W(\xi_h) < \lambda_W(\xi_s)$. However, once $\delta_s$ is large enough, the condition in equation (304) holds for almost all values of $c_1, c_2$. This indicates that it is relatively easy to satisfy the condition for heavy-hitter sparse attention to have better stability constant $\lambda_W(\xi_h)$ with respect to the learnable attention parameter $\mathbf{W}$ than the $\lambda_W(\xi_s)$ for full attention. However, the conditions for $\lambda_X(\xi_h) < \lambda_X(\xi_s)$ in equation (305) are bit more restrictive as it depends on $\beta$ which corresponds to the number of query tokens that might attend to the same key – the attention sink ratio. This relationship is visualized in figure 22. While for small values of $\beta$ (column 1-3) and large enough $\delta_s$, almost all values of $c_1, c_2$ satisfy equation (305). However, as the value of $\beta$

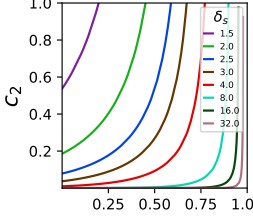

Figure 21: Relationship of $c_1, c_2, \delta_s$ in equation (304). For any value of $\delta_s$, the region above the line denotes values of $c_1, c_2$ for which $\lambda_W(\xi_h) < \lambda_W(\xi_s)$.

increases, the conditions are only satisfied for large values of $\delta_s$ and small enough $c_1$. We present the distribution of the semantic dispersions, semantic separations and sink ratios different datasets computed over the whole training set with the trained model in table 6. Overall, it shows that the full attention dispersion $\delta_s$ is usually significantly larger than the heavy-hitter attention dispersion $\delta_h$. We present different percentiles of the values seen over all queries in all training points across all transformer blocks in the model. Based on these values, we also plug them into the conditions equation (304) and equation (305) in corollary 2 and report the left-hand-side values in the table. We

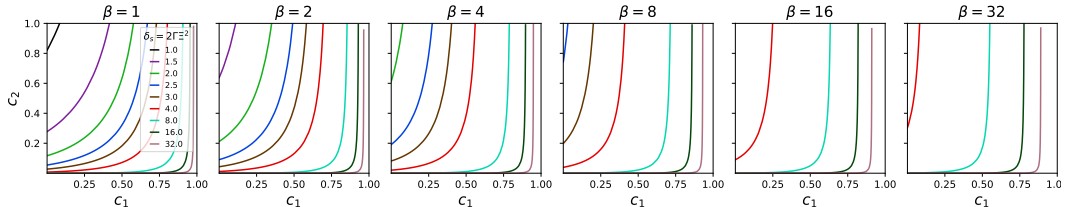

Figure 22: Relationship of $c_1, c_2, \delta_s, \beta$ in equation (305). For any value of $\delta_s$ and $\beta$, the region above the line denotes values of $c_1, c_2$ for which $\lambda_X(\xi_h) < \lambda_X(\xi_s)$. In these figures, we assume $\delta_s = 2\Gamma\Xi^2$ so that we just need to vary $\delta_s$ without considering different values of $\Gamma$ and $\Xi$.

see that, in almost all cases, the left-hand-side values are lower than 1, implying that the heavy-hitter attention has better guarantees, which aligns with the empirical results we see in figure 7. This is especially true when we only consider the values using the 95-th percentile values for semantic dispersions, sink ratios, and the 5-th percentile values for the semantic separations, which is the most relevant quantity as we have been studying the worst-case stability constants. There is one case where the values are not less than 1, counter to what we see in the empirical evaluations of figure 7. Note that we are evaluating these conditions at the optimum (the final trained model) instead of over the whole parameter space. For that reason, it is important to look at the whole loss surface which we do in the sequel.

Table 6: Empirical distribution of the semantic dispersions $\delta_s, \delta_h$, semantic separations $\Delta_h$ and $\beta$ for different datasets. For each metric, we report the 75-th, 90-th, 95-th and 100-th (maximum) percentile (except for $\Delta_h$ for which we report the 25-th, 10-th, 5-th, 0-th (minimum) percentile as it is a lower bound). The left-hand-side (LHS) of equation (304) and equation (305) are computed using the corresponding percentile values. Note that for this set of results, $k = 5$ in heavy-hitter sparse attention.

| Dataset | Full attn dispersion $\delta_s$ | HH attn dispersion $\delta_h$ | HH separation $\Delta_h$ | Sink ratio $\beta$ |
|---|---|---|---|---|
| ListOps | [8.61, 18.5, 29.4, 87.8] | [3.51, 6.74, 9.67, 28.2] | [0.016, 0.005, 0.002, 1e-9] | [0.2, 0.6, 3.0, 119.6] |
| Parity | [8.30, 10.1, 11.2, 19.4] | [2.31, 3.13 3.78, 9.16] | [0.062, 0.022, 0.011, 1e-6] | [1.6, 2.6, 3.2, 6.6] |
| EvenPairs | [2.03, 4.73, 9.44, 14.6] | [1.03, 2.84, 5.50, 8.25] | [0.009, 0.003, 0.002, 3e-8] | [1.2, 3.4, 5.2, 8.0] |
| MissDup | [4.63, 9.25, 17.1, 23.9] | [2.36, 4.25, 4.88, 10.5] | [0.018, 0.006, 0.003, 1e-7] | [1.4, 3.0, 4.2, 8.0] |

| Dataset | LHS (304) | LHS (305) |
|---|---|---|
| ListOps | [0.97, 0.69, 0.56, 0.57] | [0.90, 0.67, 0.59, 0.61] |
| Parity | [0.70, 0.76, 0.80, 1.22] | [0.72, 0.81, 0.86, 1.29] |
| EvenPairs | [3.17, 1.98, 1.31, 1.80] | [3.02, 2.10, 1.42, 1.90] |
| MissDup | [1.53, 1.09, 0.67, 1.41] | [1.53, 1.15, 0.71, 1.20] |

### G.5 Loss Surfaces and Estimated Lipschitz Constants

Beyond understanding the relative behavior of the aforementioned stability (and thus Lipschitz) bounds, we also empirically visualize the training loss landscapes for 4 of the tasks in figure 23. We use the version of transformer block with the ReLU activated MLP (for loss surfaces of transformer blocks with GELU activated MLPs see appendix G.5 in figure 25). We utilize the techniques proposed in Li et al. [37]. Given the training model parameters $\Theta$, we pick two random directions $\vartheta_1$ and $\vartheta_2$, and then plot the training loss $\mathcal{L}(\Theta + x\vartheta_1 + y\vartheta_2)$ at the grid point $(x, y)$, $x, y \in [-1, 1]$. [4] The grid points are computed as a granularity of $\varepsilon = 0.005$ in both axis, that is, $x, y \in \{-1, -1+\varepsilon, -1+2\varepsilon, \ldots, 1-\varepsilon, 1\}$. We utilize the computed loss at each grid point to generate contour plots (a heatmap visualization of the loss surface is provided in appendix G.5 in figure 24). Note that the grid point $(0, 0)$ corresponds to the loss $\mathcal{L}(\Theta)$ of the trained model. The contours on the loss surfaces of full attention model are somewhat asymmetric – see for example, around the center in figure 23c, figure 23d, and moderately in figure 23a. In contrast, the loss surfaces of the heavy-hitter top-$k$ attention model are quite symmetric, especially around the center.

---

[4]Note that the random directions are "filterwise normalized", which means that each matrix of parameters is normalized independently.

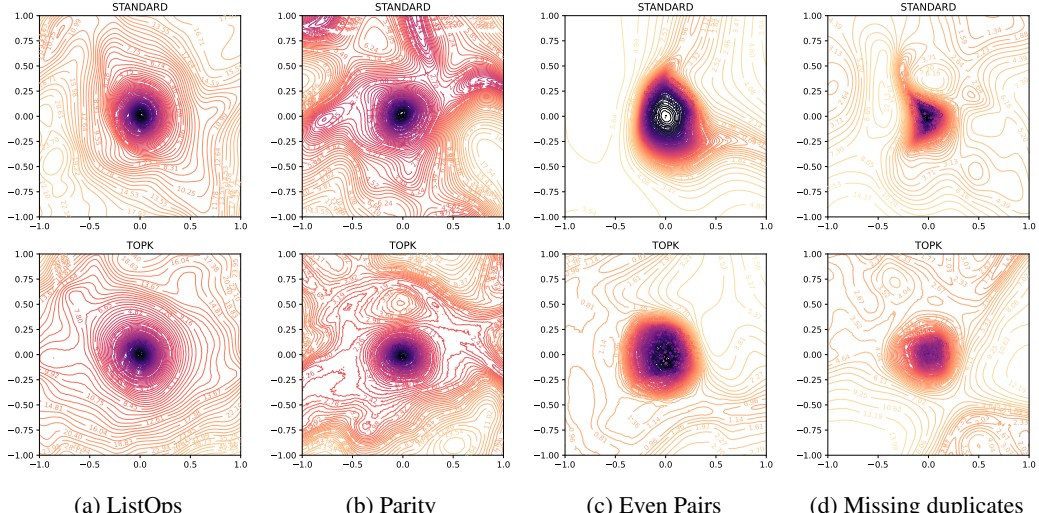

(a) ListOps      (b) Parity      (c) Even Pairs      (d) Missing duplicates

Figure 23: Loss surfaces of the models with full attention (top row) and top-$k$ attention (bottow row) for each of the 4 tasks considered in figure 7 with the corresponding hyperparameters utilizing the filter-normalized version of the loss landscape visualization techniques proposed in Li et al. [37]. Note that the (0,0) grid point corresponds to the final trained model.

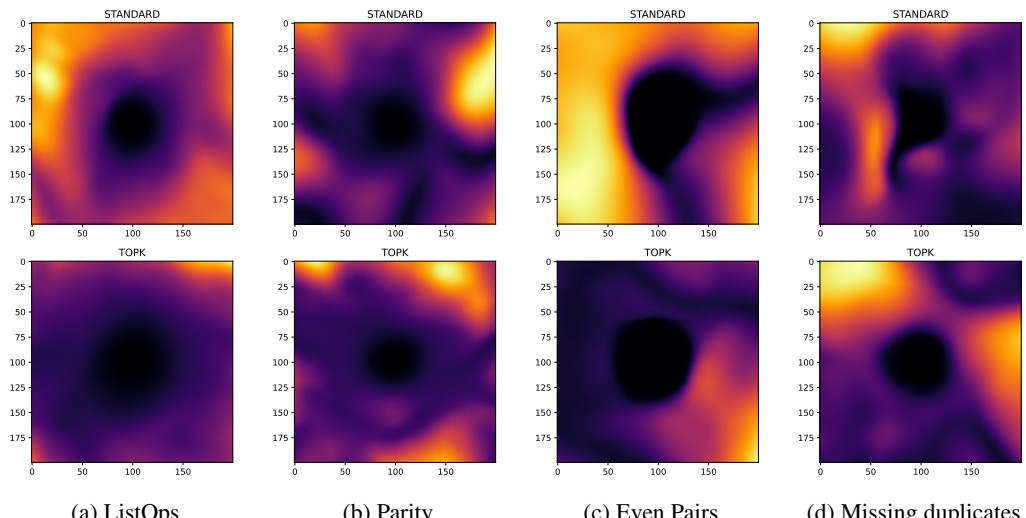

(a) ListOps      (b) Parity      (c) Even Pairs      (d) Missing duplicates

Figure 24: Loss surfaces as in figure 23 but in the form of heatmaps instead of contour plots.

Beyond visualizing the loss surface in 2-dimensions, we also utilize the loss surface to approximately estimate the Lipschitz constant of the model in the selected random directions $\varepsilon_1, \vartheta_2$. Given the loss values $\mathcal{L}(\theta + x\vartheta_1 + y\vartheta_2)$ at grid points $(x, y)$, we compute the following ratios at neighboring horizontal and vertical grid points as an estimate of the Lipschitz constant $\lambda_{\mathcal{L}}$ in theorem 2:

$$\frac{|\mathcal{L}(\Theta + x\vartheta_1 + y\vartheta_2) - \mathcal{L}(\Theta + (x + \varepsilon)\vartheta_1 + y\vartheta_2)|}{\varepsilon\|\vartheta_1\|} \quad \text{and}$$

$$\frac{|\mathcal{L}(\Theta + x\vartheta_1 + y\vartheta_2) - \mathcal{L}(\Theta + x\vartheta_1 + (y + \varepsilon)\vartheta_2)|}{\varepsilon\|\vartheta_2\|}. \tag{306}$$

We plot the distribution of these estimates in figure 27 for the loss surfaces in figure 23 of 4 of the tasks for varying grid ranges $r \in (0, 1]$ with $x, y \in [-r, r]$. We plot the 50-th, 75-th, 95-th and 99-th percentile values of these estimates of the full attention model and the heavy-hitter top-$k$ attention model. We see that near the trained model (small values of the grid range $r$), the distributions of these

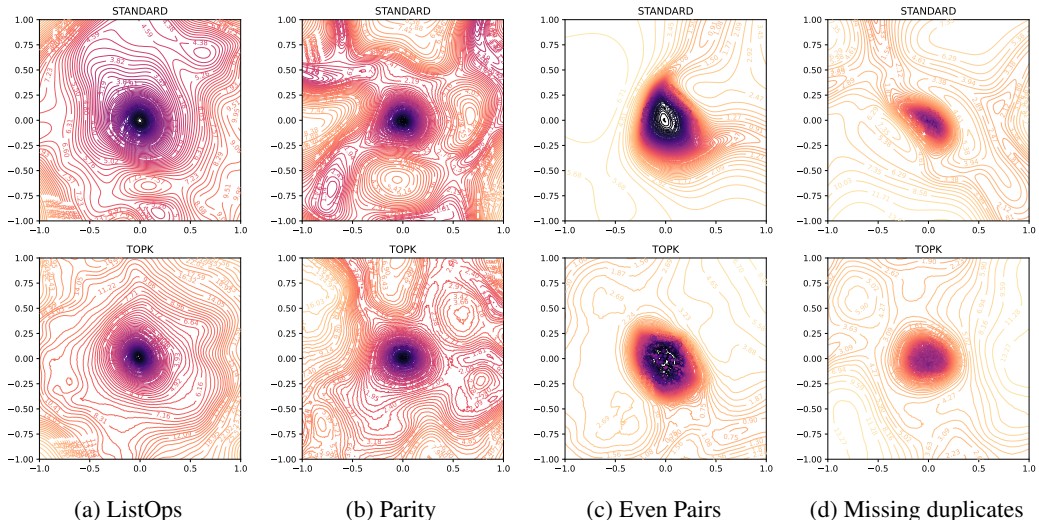

(a) ListOps        (b) Parity        (c) Even Pairs        (d) Missing duplicates

Figure 25: Loss surfaces as in figure 23 of the models with full attention (top row) and top-$k$ attention (bottow row) for each of the 4 tasks considered in figure 7 and table 4. Note that both forms of attention now utilize the MLP with GELU activation for all tasks (as opposed to ReLU activation in figure 23).

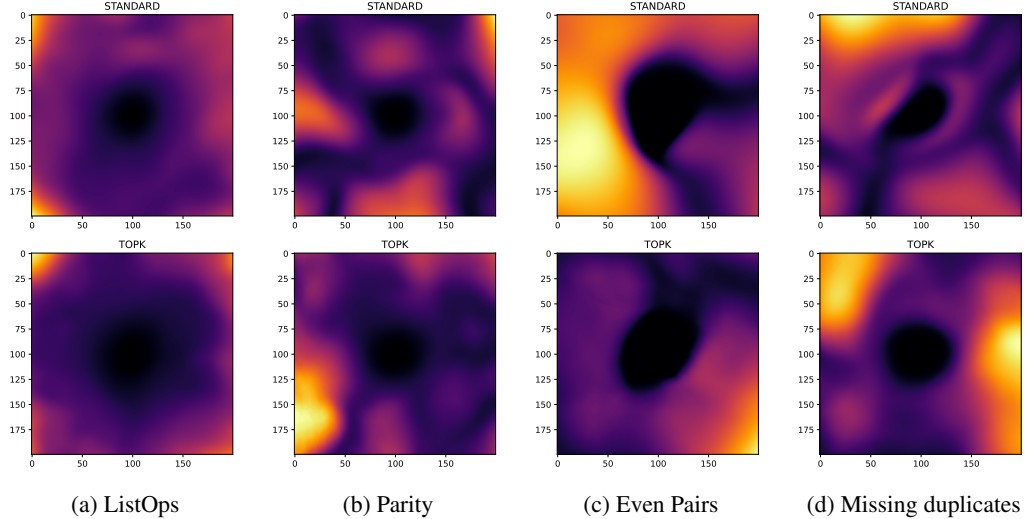

(a) ListOps        (b) Parity        (c) Even Pairs        (d) Missing duplicates

Figure 26: Loss surfaces as in figure 25 but in the form of heatmaps instead of contour plots.

estimates are close for both the models. However, as we move farther away from the trained model (large values of $r$ in the horizontal axis), the distributions change significantly, and the top-$k$ attention model provides a smaller Lipschitz constant estimate compared to the full attention model across all percentiles. This indicates that, at least empirically, the loss for the top-$k$ attention model has a much more favorable Lipschitz constant compared to that of the full attention model, which in turn implies both faster convergence and better generalization guarantees. Thus, our stability-based theoretical investigation in this section appears to align with our empirical observations in section 3.

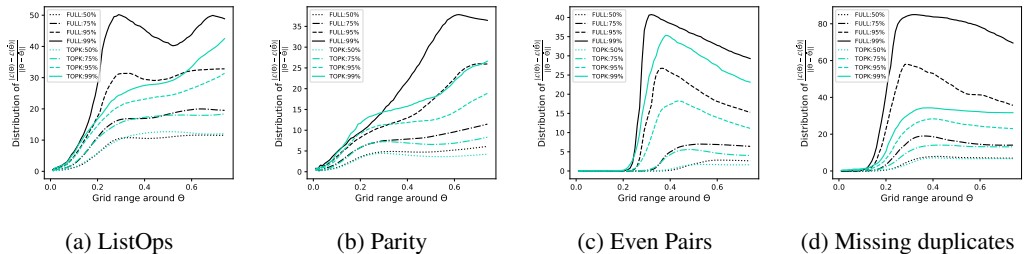

(a) ListOps       (b) Parity       (c) Even Pairs       (d) Missing duplicates

Figure 27: Distribution of the estimated Lipschitz constants computed in the random directions utilized to visualize the loss landscape in figure 23 for full attention and top-$k$ attention each of the 4 tasks considered in figure 7 with the corresponding hyperparameters. We report the distributions of the estimated Lipschitz constants in the vertical axis in terms of the 50-th (dotted), 75-th (dash-dotted), 95-th (dashed) and 99-th (solid) percentiles (lower is better). On the horizontal axis, we denote the radius of the ball around the parameters of the final learned model, and visualize how the distributions vary as the ball radius is increased.

