# OpenReview forum: "Transformers Learn Faster with Semantic Focus"
_NeurIPS.cc/2025/Conference — NeurIPS 2025 poster_

### Official Review · Reviewer_yPUz · 2025-06-25

**Clarity:** 2
**Significance:** 2
**Originality:** 3
**Rating:** 4
**Confidence:** 3

**Summary:**

This paper investigates whether sparse attention in transformers improves learnability and generalization, beyond just computational efficiency. Empirically, input-dependent sparse attention  accelerates convergence and enhances generalization compared to full attention, while input-agnostic sparse attention offers no benefits. Theoretically, the authors link this to "semantic focus": input-dependent sparsity improves the stability of the softmax operation by reducing semantic dispersion.

**Questions:**

1. From my experience, when training with sparse attention, the loss and some benchmark can show similar or better results than full attention. However, the sparse attention will fail in the challenging **downstream tasks** such as Needle in a Haystack. Could the authors provide any explanation using the proposed theory on this issue?

2. Since the paper discusse top-k or k-heavy-hitter attention. Could you provide any theorical or empirical guidence on how to set the k in practice? Or I can ask in a different way, how can we deterimine the sparsity before seeing the input?

**Ethical Concerns:**

["NO or VERY MINOR ethics concerns only"]

**Final Justification:**

I get better understand standing of the experiments and insights from the paper. However, I agree with other reviwers' comment that the paper is too heavy to read. It would be better if the author can improve the writing and make it well organized.

**Limitations:**

The authors already mention their limitation in large scale experiments. I think another limitation is the sparse attention methods used being too old.

**Quality:**

2

**Strengths And Weaknesses:**

Strengths:
- The authors aim to provide a theoretical explaination of why sparse attention improves learnability and generalization, which is different from most of the research that focusing on efficiency.

Weakness:
- I think full attention is already **input-dependent sparse** as most of the softmax score is very low. It's only a matter of being complete zero or a small value. As a result, I guess what interest me more is whether sparse attention can be both efficient while being better than full attention? To be hardware efficient, there are many new input dependent sparse attention methods such as NSA[1], MoBA[2],  SeerAttention[3], MInference[4], Quest[5], and etc that provide block-wise sparsity. The current sparse attention methods used in the paper are not state-of-the-art.
- The paper does not provide results on large models. It's questionable whether their conclusion still holds at scale.




[1] Yuan, Jingyang, et al. "Native sparse attention: Hardware-aligned and natively trainable sparse attention." arXiv preprint arXiv:2502.11089 (2025).

[2] Lu, Enzhe, et al. "Moba: Mixture of block attention for long-context llms." arXiv preprint arXiv:2502.13189 (2025).

[3] Gao, Yizhao, et al. "Seerattention: Learning intrinsic sparse attention in your llms." arXiv preprint arXiv:2410.13276 (2024).

[4] Jiang, Huiqiang, et al. "Minference 1.0: Accelerating pre-filling for long-context llms via dynamic sparse attention." arXiv preprint arXiv:2407.02490 (2024).

[5] Tang, Jiaming, et al. "Quest: Query-aware sparsity for efficient long-context llm inference." arXiv preprint arXiv:2406.10774 (2024).

---

> ### Author Rebuttal · Authors · 2025-07-30
>
> We truly appreciate the reviewer's time and effort in thoroughly reviewing our manuscript. We have tried to address all their comments in the following. We are happy to answer any further questions that can motivate the reviewer to raise their ratings.
>
> ---
>
> > _[..] full attention is already input-dependent sparse [..] only a matter of being complete zero or a small value._
>
> As softmax is intended to be a regularized argmax (with the Shannon entropy regularization), softmax attention is already meant to be **almost** input-dependent sparse, with the difference being that the low values are not exactly zero. However, as our empirical evaluations indicate, this can lead to significant differences in behaviour between full softmax-attention and heavy-hitter top-$k$ sparse attention, and the theoretical analysis explains why this might be occurring, highlighting the effect of the semantic dispersion in full softmax-attention.
>
> Softmax-attention will always put a strictly non-zero probability mass on each of the keys. This property of softmax has already been shown in literature as the source of **attention dispersion**, where the softmax output tends to approach a uniform distribution instead of looking like the one-hot distribution of an argmax (Velickovic et al., 2025; Nakanishi, 2025). Such behaviour of softmax can lead to **representational collapse** in transformers where all tokens end up with the same representation (Barbero et al., 2024). Thus, it seems that full softmax-attention can behave quite differently (and undesirably) in practice from its originally intended smoothed-argmax form.
>
> > _To be hardware efficient, there are many new input dependent sparse attention methods such as NSA[1], MoBA[2], SeerAttention[3], MInference[4], Quest[5], and etc that provide block-wise sparsity. The current sparse attention methods used in the paper are not state-of-the-art._
> >
> > - _[1] Yuan, Jingyang, et al. "Native sparse attention: Hardware-aligned and natively trainable sparse attention."_
> > - _[2] Lu, Enzhe, et al. "Moba: Mixture of block attention for long-context llms."_
> > - _[3] Gao, Yizhao, et al. "Seerattention: Learning intrinsic sparse attention in your llms."_
> > - _[4] Jiang, Huiqiang, et al. "Minference 1.0: Accelerating pre-filling for long-context llms via dynamic sparse attention."_
> > - _[5] Tang, Jiaming, et al. "Quest: Query-aware sparsity for efficient long-context llm inference."_
>
> We thank the reviewer for these state-of-the-art input-dependent hardware-efficient blockwise-sparse attention schemes. We completely agree that top-$k$ attention is not necessarily the state-of-the-art input-dependent sparse attention scheme. We would like to clarify that our theoretical analysis is focused on "heavy-hitter" sparse attention, where the keys with the highest attention scores are the only ones that are not masked. There are many other ways to make the sparse attention input-dependent without necessarily being a heavy-hitter style sparse attention.
>
> We consider top-$k$ attention in our experiments as it is one of the most basic forms of input-dependent sparse attention mechanisms. We believe that it is important to understand some phenomena at the simplest level first to see how consistent it is, and if there is some theory that can explain it. The above schemes [2-5] are designed for inference, and thus their effect on training convergence is unexplored; they are applied to pretrained models trained with full attention. Note that all schemes [1-5] utilize a "top-k" over block scores (each with their own scoring rule). Given that we are selecting blocks (in an input-dependent manner), there is no guarantee that the semantic dispersion within a block will be low. We hope to extend our theory to better understand input-dependent block-sparse schemes in future work.
>
> Gupta et al. (2021) originally consider top-$k$ attention, and focus on the memory efficiency of top-$k$ attention without improving the runtime of the attention. However, various advancements have sped up the computational cost of top-$k$ attention both in terms of runtime and memory overhead such as Zeng et al. (2025). This has shown significant speedup over pytorch native attention implementation, as well as the optimized FlashAttention, with the speedup increasing with context length. So top-$k$ attention can now be significantly faster than standard full attention in terms of actual runtime. Furthermore, other heavy-hitter attention such as Reformer (Kitaev et al., 2020) can also speed up the actual attention runtime with locality sensitive hashing.
>
> Our focus has been on a complimentary angle where we study how sparsity affects optimization convergence, and try to provide theoretical intuition as to when and how can sparse attention converge faster than standard full attention. In situations where top-$k$ (or any other $k$-heavy-hitter attention) would converge faster than standard full attention, the overall runtime would improve cumulatively with faster sparse attention and fewer number of steps to convergence. If the sparse attention is also hardware-efficient, then that benefit will be in addition to the faster convergence --- each training step will be faster, and we will need fewer training steps.
>
> > _The paper does not provide results on large models_
>
> The reviewer is accurate in their assessment here, and this is something we have already listed in our limitations. Our small scale experiments and theoretical analyses provide a strong motivation to attempt these at scale. However, we believe that it is important to understand some phenomena at a small scale to see how consistent it is, and if there is some theory that can explain it. In our thorough small-scale experiments, we see that the behaviours are quite consistent --- heavy-hitter sparse attention does consistently converge faster than full-attention. Furthermore, while we do not consider large models, we do study the effect of increasing the size of the models in terms of the number of transformer blocks, or the number of heads per block; see Figure 3 and Observation IV.
>
> > _[..] when training with sparse attention, [..] show similar or better results than full attention. However, the sparse attention will fail in the challenging downstream tasks such as Needle in a Haystack. Could the authors provide any explanation using the proposed theory on this issue?_
>
> It is true that the input-agnostic sparse attention can fail on associative recall tasks such as Needle in a Haystack. This is expected as the input-agnostic sparsity pattern can easily miss the relevant associations. This is also related to the expressivity results of Yun et al. (2020) for sparse attention transformers. On the other hand, heavy-hitter input-dependent sparse attention can do quite well with associative recall tasks --- as long as the dot-product attention scores can appropriately model the associations, a heavy-hitter sparse attention can appropriately perform associative recall. As one example, Zeng et al. (2020) show that top-$k$ attention can match the associative recall performance of standard attention (see Figure 2(a) in Zeng et al. (2020)), and small values of $k$ is sufficient (see Figure 2(d) in Zeng et al. (2020)). However, we cannot make any general claims here since "downstream performance" does not only depend on the use of full or sparse attention, but also on what data/tasks the model is trained on.
>
> > _Since the paper discusse top-k or k-heavy-hitter attention. Could you provide any theorical or empirical guidence on how to set the k in practice? Or I can ask in a different way, how can we deterimine the sparsity before seeing the input?_
>
> As with many architectural hyperparameters, we think that $k$ is a problem dependent hyperparameter. We are usually not able to select the number of transformer blocks, the number of attention heads, the MLP hidden-layer dimensions, and other architectural hyperparameters without "seeing the input", and we believe this to also be the case with $k$. Ideally, for a given architecture, we would like to make $k$ as small as possible while still retaining the necessary level of expressivity (that is, be able to achieve perfect training accuracy).
>
> At a more high level, softmax is a smoothed version of argmax for the purposes of differentiability, and thus attention is actually crafted to look for the argmax, implying that $k \sim O(1)$ should suffice. This is the motivation behind studying the theoretical properties of the argmax or hardmax attention transformers, where $k=1$ (Strobl et al., 2024).
>
> As an alternate mechanism, there are heavy-hitter sparse attention transformers that can adapt the sparsity level to the input and the problem (Correia et al., 2019), but that scheme also requires the setting of various other problem-dependent hyperparameters. These are an interesting class of sparse attention transformers, and we hope to extend our theoretical framework to such adaptive sparse attention mechanisms in future work.
>
> -----------
> ### References
> - Velickovic, P., et al. "softmax is not enough (for sharp out-of-distribution)". arXiv preprint arXiv:2410.01104 (2024).
> - Nakanishi, K. M. "Scalable-softmax is superior for attention". arXiv preprint arXiv:2501.19399 (2025).
> - Barbero, F., et al. "Transformers need glasses! Information over-squashing in language tasks." NeurIPS (2024).
> - Gupta, A., et al. "Memory-efficient Transformers via Top-$ k $ Attention." SustainNLP workshop (2021).
> - Zeng, Q., et al. "Zeta: Leveraging z-order curves for efficient top-k attention." ICLR (2025).
> - Kitaev, N., et al. "Reformer: The efficient transformer." ICLR (2020).
> - Yun C.,  et al. "$O(n)$ connections are expressive enough: Universal approximability of sparse transformers." NeurIPS (2020).
> - Strobl L., et al. "What formal languages can transformers express? A survey." TACL (2024).
> - Correia, G. M., et al. "Adaptively sparse transformers." EMNLP-IJCNLP (2019).

---

> > ### Comment · Reviewer_yPUz · 2025-08-02
> > **Futher convicing results**
> >
> > Thanks for the explanation. I understand the paper is trying to explain why sparse attention is good and converging fast. But at this stage, I can not fully believe that the preassumption that sparse attention is good. Even with the latest NSA, we have seen a certain amount of gap in challeging downstream NIAH test (e.g. muilti-key, or multi-value). That's something can not show in the training loss. I have been fooled once using sparse attention. The loss and ppl is so good but downstream task is dispointing. So converging fast and lower PPL can be miss-leading and is not a convincing evidence from my side.
> >
> > Thus, I would raise my rating if the author can provide more results and evaluation on tasks like muilti-key retrievel or any other reasonable real tasks.

---

> ### Author Response · Authors · 2025-08-03
>
> We thank the reviewer for their acknowledgement and their time. We respond to them in the following:
>
> > _Thanks for the explanation. I understand the paper is trying to explain why sparse attention is good and converging fast. But at this stage, I can not fully believe that the preassumption that sparse attention is good._
>
> We would like to clarify that our paper is not "_trying to explain why sparse attention is good and converging fast_" but rather trying to understand conditions under which sparse attention can train and generalize faster, **and** conditions under which they will not. Our results clearly show that **not all sparse attention are the same**.
> - Input-agnostic sparse attention mechanisms do not necessarily train/generalize faster than full attention. Our empirical results demonstrate this, and our theoretical results explain why input-agnostic sparse might not show any improved performance.
> - Input-dependent heavy-hitter attention schemes can train/generalize faster than full attention, as clearly demonstrated by our empirical evaluations across many variations of the learning setup. Our theory explains why this might be happening:
>   - The transformer model performance (training convergence or generalization) can depend on the semantic dispersion (Definition 2) -- larger dispersions lead to worse performance.
>   - Heavy-hitter input-dependent sparse attention schemes can significantly reduce the semantic dispersion, thereby improving performance.
>   - Input-agnostic sparse attention schemes do not reduce the semantic dispersion, and thus would not generally give us faster convergence or generalization.
>
> > _Even with the latest NSA, we have seen a certain amount of gap in challeging downstream NIAH test (e.g. muilti-key, or multi-value). That's something can not show in the training loss. I have been fooled once using sparse attention. The loss and ppl is so good but downstream task is dispointing._
>
> We thank the reviewer for sharing their experience with "latest NSA" (we are assuming that NSA refers to "native sparse attention" that focuses on input-dependent block-sparse attention).
>
> It would be great if the reviewer can share a published reference regarding this comment: "_Even with the latest NSA, we have seen a certain amount of gap in challeging downstream NIAH test (e.g. muilti-key, or multi-value). That's something can not show in the training loss. I have been fooled once using sparse attention. The loss and ppl is so good but downstream task is dispointing._"
>
> It is important to understand the actual sparse attention mechanism to make any claims about whether we will really see any improvements. As we have already mentioned in our rebuttal, for native sparse attention mechanism focusing on block-sparse attention, **there is again no guarantee that the semantic dispersion within a block will be low**, and thus, there might not be any performance improvement over full attention. So our empirical findings and theoretical analysis actually aligns with the reviewer's experience.
>
> > _So converging fast and lower PPL can be miss-leading and is not a convincing evidence from my side._
>
> We have presented both training loss and held-out accuracy for all the problems and methods (see for example in Figures 1 and 2). We are not just claiming faster training convergence, but also highlight how the training convergence also translates to faster generalization (on unseen data).

---

> ### Author Response · Authors · 2025-08-03
>
> > _Thus, I would raise my rating if the author can provide more results and evaluation on tasks like muilti-key retrievel or any other reasonable real tasks._
>
> Low semantic dispersion, and thus improved performance, is usually guaranteed for heavy-hitter attention such as top-$k$ attention. As we discussed in our rebuttal, Zeng et al. (2025) have already demonstrated clearly that top-$k$ sparse attention performs well for needle in a haystack associative recall problems. Furthermore, the tasks we have considered such as ListOps (list operations) is a problem requiring (hierarchical) multi-key/value attention as it requires the retrieval of all the keys/values corresponding to operands within the enclosing brackets for the query corresponding to the operator to appropriately solve the problem. This multi-key/value retrieval in the long sequence needs to be done hierarchically through the whole input. And given the nature of the problem, the attention span -- the distance between the query token index and the required key token indices -- can be quite large (as already discussed in Tay et al., 2022). The input-dependent nature of top-$k$ allows it to model this desired behaviour.
>
> Thus, we have already shown the necessary empirical evidence, and provided theoretical support for this empirical findings. Furthermore, existing work has already demonstrated the multi-key retrieval capabilities of top-$k$ sparse attention. Our contribution is not the top-$k$ attention, but understanding when and how heavy-hitter sparse attention outperforms standard full attention (and when sparse attention does not outperform full attention).
>
> Finally, to reiterate, the goal of our study is to understand conditions under which sparse attention can be beneficial in terms of learning convergence and generalization. Our results, supported by our theory, show that not all sparse attention schemes would show benefits all the time, and there are precise conditions --- sparse attention mechanisms that ensure low semantic dispersion --- that lead to improved performance over standard full attention. Input-agnostic sparse attention mechanisms do not ensure such a condition (and this is also possibly why "native sparse attention" with input-dependent block-sparsity does not ensure improved performance). In contrast, input-dependent heavy-hitter attention can ensure low semantic dispersion, and our empirical evaluations demonstrate improved performance over standard full attention.
>
>
> -------
>
> ### References
> - Zeng, Q. et al. "Zeta: Leveraging z-order curves for efficient top-k attention." ICLR (2025).
> - Tay Y, et al. "Long range arena: A benchmark for efficient transformers". ICLR (2021).
> - Nangia, N. and Bowman, S. Listops: A diagnostic dataset for latent tree learning. NAACL: Student Research Workshop (2018).

---

> > ### Comment · Reviewer_yPUz · 2025-08-05
> >
> > The authors have solved some of my concerns in evaluation. I will raise my rating to 4

---

> > > ### Author Response · Authors · 2025-08-06
> > >
> > > Thank you for taking your time to engage with us. We appreciate your updating your evaluation. Please let us know if we can provide further clarification to address any outstanding concerns.

---

### Official Review · Reviewer_XrxX · 2025-06-29

**Clarity:** 3
**Significance:** 3
**Originality:** 3
**Rating:** 4
**Confidence:** 3

**Summary:**

This paper investigates the role of sparse attention in Transformers from both theoretical and empirical perspective. The authors empirically show that input-dependent sparse attention, such as Top-k masking, leads to faster training convergence and better generalization compared to both full attention and input-agnostic sparse variants (e.g., block, banded). Theoretical analysis links this behavior to improved softmax stability and a tighter Lipschitz constant for the training objective. The work is rigorous, well-motivated, and presents both theoretical insight and empirical validation for an underexplored benefit of sparse attention mechanisms.

**Questions:**

See above

**Ethical Concerns:**

["NO or VERY MINOR ethics concerns only"]

**Final Justification:**

I will keep my scores.

**Limitations:**

See above

**Paper Formatting Concerns:**

-

**Quality:**

3

**Strengths And Weaknesses:**

This paper presents strong and consistent empirical evidence that input-dependent sparse attention improves convergence and generalization. Theoretical analysis is solid, linking attention sparsity to improved softmax stability and tighter Lipschitz bounds. Results are robust across multiple tasks, architectures, and hyperparameters.

However, I have several concerns before this paper can be accepted.

1. Lack of literature search. The papers should discuss more about the current advanced sparse attention techniques (e.g., Performer, Longformer, Reformer) as discussed in [1].  Is it possible for the authors to compare these techniques in synthetic data experiments? I am curious about the performance of these techniques.

2. Small-scale synthetic tasks. I appreciate the detailed experiments. A minor suggestion is that the authors could consider some real data experiments.

3. How the choice of $k$ affects performance, generalization, or the theoretical bounds?

[1] Tay Y, Dehghani M, Bahri D, et al. Efficient transformers: A survey[J]. ACM Computing Surveys, 2022, 55(6): 1-28.

---

> ### Author Rebuttal · Authors · 2025-07-30
>
> We truly appreciate the reviewer's time and effort in thoroughly reviewing our manuscript and raising insightful questions. We have tried to address all their comments in the following. We are thankful for their support, and we are happy to answer any further questions that can motivate the reviewer to raise their ratings.
>
> ---------
>
> > _Lack of literature search. The papers should discuss more about the current advanced sparse attention techniques (e.g., Performer, Longformer, Reformer) as discussed in [1]._
> >
> > _[1] Tay Y, Dehghani M, Bahri D, et al. Efficient transformers: A survey[J]. ACM Computing Surveys, 2022, 55(6): 1-28._
>
> We thank the reviewer for the useful pointers, and we apologize for the succinct literature survey in our main paper. We do provide a detailed literature survey in Appendix B where we discuss Reformer (Kitaev et al., 2020) and Longformer (Beltagy et al., 2020) alongside other sparse attention transformers. We thank the reviewer for bringing up Performer. We chose to not include Performer in our discussion as we focus on sparse attention transformers. Instead of sparsifying the attention matrix, techniques such as Performers (Choromanski et al., 2020) consider a low-rank approximation of the post-softmax attention matrix using (orthogonal positive) random features (Rahimi and Recht, 2007). Our current discussion and analysis will not be applicable to low-rank approximating schemes such as Performer; we leave such a study for future work.
>
> > _Is it possible for the authors to compare these techniques in synthetic data experiments? I am curious about the performance of these techniques._
>
> While we did not explicitly consider the input-agnostic strided sparsity pattern from Longformer in our experiments, we do make use of the global tokens from Longformer alongside the input-agnostic banded and block local sparse attention models. These global tokens are critical for the expressivity of the input-agnostic sparse attention transformer models as discussed by the theoretical results of Yun et al. (2020), and also validated in our empirical experiments. However, the inclusion of these global tokens in input-agnostic sparse attention does not speed up the training convergence of input-agnostic sparse attention when compared to full-attention --- see BAND:5(1) and BLOC:5(1) in Figures 1 and 2 in the main paper, and further empirical analyses in Figures 9-18 in Appendix D for varying values of $k$ and varying number of global tokens for input-agnostic sparse attention transformers. We also briefly discuss the implications of Theorem 4 (for regular input-agnostic sparse attention) for strided sparse attention as in Longformer in Lines 332-333.
>
> In contrast, input-dependent heavy-hitter sparse attention such as top-$k$ attention retains necessary expressivity, and does not need the global tokens. Furthermore, top-$k$ attention transformers converge significantly faster than full-attention transformers, highlighting the need for input-dependence for faster convergence. This is corroborated in our theoretical analyses, where the theoretical bounds for input-agnostic sparse attention in Theorem 4 do not necessarily show any improvement over full-attention results in Theorem 3.
>
> While Reformer would have been a good candidate for empirical evaluation of input-dependent heavy-hitter sparse attention transformers, we limited our experiments to top-$k$ attention for reasons discussed in Appendix C, lines 1129-1133:
>
> "The main motivation for selecting top-$k$ over LSH based or clustering based input-dependent sparse attention is that we can then easily ensure that the input-dependent sparse attention attends to exactly the same number of tokens as in the input-agnostic ones -- that is, the number of nonzeros in each column of the attention score matrix is exactly the same across all sparse attention patterns we consider."
>
> We believe that is necessary to ablate the effect of input-dependent vs input-agnostic sparse attention. Otherwise, with methods such as Reformer, a query can often attend to more that $k$ keys depending on the number of key tokens in the same hash bucket as the query token. In such a scenario, it is not clear whether the improved expressivity and convergence of input-dependent sparse attention over input-agnostic ones is due to the input-dependence or due to the ability to attend to more that $k$ keys.
>
> > _A minor suggestion is that the authors could consider some real data experiments._
>
> We consider a small experiment with the PennTreeBank natural language dataset where we use the context of tokens to predict the next token. The text is tokenized using the SentencePiece tokenizer with a vocabulary size of 4096. We consider a transformer with 8 single-headed blocks, embedding size of 32 and MLP hidden dimensionality of 128. We train the model for 50 epochs with SGD (we plan to run it for longer but are reporting intermediate results here). We consider full attention and top-$k$ attention with $k=5$ and report the token misclassification cross-entropy on the training set at increasing number of epochs:
>
> | Method / Epoch | 5 | 10 | 15 | 20 | 25 | 30 | 35 | 40 | 45 | 50 |
> | -- | -- | -- | -- | -- | -- | -- | -- | -- | -- | -- |
> | Full | 3.29 |3.08 |2.97 |2.89 |2.83 |2.79 |2.74 |2.72 |2.69 |2.67 |
> | top-$k$ (% gain) | 3.21 (2.4%) |2.98 (3.2%) |2.87 (3.4%) |2.79 (3.5%) |2.74 (3.2%) |2.70 (3.2%) |2.65 (3.3%) |2.62 (3.7%) |2.59 (3.7%) |2.56 (4.1%) |
>
> We also considered a larger model 16 single-headed transformer blocks for the same experiment for 50 epochs with SGD:
>
> | Method / Epoch | 5 | 10 | 15 | 20 | 25 | 30 | 35 | 40 | 45 | 50 |
> | -- | -- | -- | -- | -- | -- | -- | -- | -- | -- | -- |
> | Full | 3.35 |3.13 |3.00 |2.93 |2.86 |2.81 |2.77 |2.73 |2.69 |2.67 |
> | top-$k$ (% gain) | 3.21 (4.2%) |2.98 (4.8%) |2.87 (4.3%) |2.79 (4.8%) |2.73 (4.5%) |2.67 (5.0%) |2.63 (5.1%) |2.59 (5.1%) |2.56 (4.8%) |2.52 (5.6%) |
>
>
>
> We see that even with natural language problems, heavy-hitter sparse attention is able to optimize the loss faster (in terms of the number of optimization steps) than full attention. We do note that these are preliminary results as we have not been able to verify these results across multiple different settings due to the time constraints during this rebuttal process.
>
> > _How the choice of $k$ affects performance, generalization, or the theoretical bounds?_
>
> The choice of $k$ in sparse attention affects the overall behaviour in multiple ways:
>
> - First, unless the number of transformer blocks are significantly high, larger values of $k$ imply better expressivity for sparse attention transformers (Yun et al., 2020). This is something we see with input-agnostic sparse attention transformers in our additional experiments in Appendix D. For example, in Figure 9, we see how the improved expressivity of larger $k$ with input-agnostic banded sparse attention and block-local sparse attention allows the learning to progress instead of stalling prematurely. In terms of learning convergence with regular input-agnostic sparse attention, we actually discuss how our bounds in Theorem 4 approaches the results for full-attention in Theorem 3 as $k$ grows to $L$, the complete sequence length (which would be full-attention); see Remark 3 on page 52 of our manuscript.
>
> - However, when expressivity is not an issue, as we see in our empirical experiments with heavy-hitter top-$k$ sparse attention, increasing $k$ *can* have an implicit effect of increasing the heavy-hitter semantic dispersion $\delta_h$ (see for example, Definition 2 and Figure 4). Larger semantic dispersion implies larger bounds for the Lipschitz constant of the learning objective, implying slower SGD convergence. This behaviour is again manifested in our additional experiments in Appendix D. For example, in Figure 9, we can see that top-$k$ attention with $k=5$ sometimes converges slightly faster than top-$k$ attention with $k=9$. Larger Lipschitz constants also imply larger algorithmic stability bounds, and thus worse generalization bounds.
>
> -------------
>
> ### References
> - Kitaev, Nikita, et al. "Reformer: The efficient transformer." ICLR (2020).
> - Beltagy, Iz, et al. "Longformer: The long-document transformer." arXiv preprint arXiv:2004.05150 (2020).
> - Choromanski, Krzysztof Marcin, et al. "Rethinking Attention with Performers." ICLR (2020).
> - Rahimi, Ali, and Benjamin Recht. "Random features for large-scale kernel machines." NeurIPS (2007).
> - Yun, Chulhee, et al. "$O(n)$ connections are expressive enough: Universal approximability of sparse transformers." NeurIPS (2020).

---

### Official Review · Reviewer_PEBb · 2025-07-03

**Clarity:** 3
**Significance:** 3
**Originality:** 2
**Rating:** 3
**Confidence:** 3

**Summary:**

This paper shows an interesting case that transformers with sparse attention sometimes learn faster than full attention. The authors try to explain this by showing that sparse attention has smaller semantic dispersion, resulting in smaller Lipschitz constants, thus learns faster.

**Questions:**

lease refer to the Weakness part in the previous section.

**Ethical Concerns:**

["NO or VERY MINOR ethics concerns only"]

**Final Justification:**

I maintain my position that sparse attention learns faster primarily because the loss landscape for top-k attention is sharper, allowing it to learn more quickly at initialization. The main experiments in this paper, Figures 1 and 2 clearly demonstrate that the primary loss difference emerges at initialization. The authors provide no experimental evidence that contradicts this intuition that top-k attention with random initialization, compared to full attention, is less stable. I still question aspects of the paper's main claim.

**Limitations:**

lease refer to the Weakness part in the previous section.

**Quality:**

2

**Strengths And Weaknesses:**

*Strengths*
1. This paper provides extensive experiments to show that sparse attention sometimes learns faster than full attention.
2. This paper provides detailed theoretical analysis for their experimental findings through the lens of Lipschitz constants and smoothness.

*Weaknesses*

I have some concerns for this paper.

1. The paper aims to show that sparse attention is more stable than full attention, i.e., the constant in Definition 1 is smaller for sparse attention. Their analysis shows this is true when semantic separation is large and semantic dispersion is small. This only happens when the model is well trained and the loss is small, i.e., the model is close to the optimal point. For an attention model with random initialization, the semantic separation should be 0 as the attention score is equal for each position. In this case, at initialization, the loss landscape for sparse attention (top-k attention) is less smooth than full attention.

2. Although in section 4 lines 243-259, the authors try to connet the convergence rate with Lipschitz constants, I still feel it is somehow far-fatched, a stable optimization algorithm doesn't mean this is a fast algorithm, especially when the following analysis consider the case to loss is close to the optimal point.

3. Note that in Figures 1, 2, 3, there is a clear loss plateau at initialization, and sparse attention can break this plateau faster compared with full attention. However, if we ignore this plateau, we can see that the convergence rate for sparse attention and full attention is close. In other words, as I analyzed in (1), I think the reason why sparse attention learns faster than full attention is that, at initialization, the loss landscape for sparse attention is sharper, thus the model can soon escape the plateau at initialization, so we can observe sparse attention learns faster.

4. In Figure 4 in [1], there is no clear difference that sparse attention learns faster. Could the authors provide some explanation?

[1] Gupta, Ankit, et al. "Memory-efficient Transformers via Top-$ k $ Attention." arXiv preprint arXiv:2106.06899 (2021).

---

> ### Author Rebuttal · Authors · 2025-07-30
>
> We truly appreciate the reviewer's time and effort in reviewing our manuscript and sharing their insightful intuitions. We have tried to address all their comments in the following. We are thankful for their time, and we are happy to answer any further questions that can motivate the reviewer to raise their ratings.
>
> ---------
>
> > _**Weakness (1).** This only happens when the model is well trained and the loss is small, i.e., the model is close to the optimal point. For an attention model with random initialization, the semantic separation should be 0 as the attention score is equal for each position. In this case, at initialization, the loss landscape for sparse attention (top-k attention) is less smooth than full attention._
>
> We thank the reviewer for sharing their thoughtful intuition. The main theoretical advantage of heavy-hitter attention over standard full-attention would occur when the full-attention semantic dispersion $\delta_s$ is higher than the heavy-hitter semantic dispersion $\delta_h$, while the heavy-hitter semantic separation $\Delta_h$ is not too small. We discuss the precise conditions when these hold in Appendix F.4. However, the semantic separation $\Delta_h$ can be quite small, not only at initialization but also at the optimum (corresponding to the trained model) as we list in Table 4 in Appendix F.4. Even when the semantic separation $\Delta_h$ is quite small (very close to zero), the stability constants $\lambda_W(\xi_h), \lambda_X(\xi_h)$ of heavy-hitter sparse attention can be better than the constants $\lambda_W(\xi_s), \lambda_X(\xi_s)$ for full attention because of the exponential dependence on the semantic dispersion.
>
> The stability constants $\lambda_W(\xi), \lambda_X(\xi)$ of the attention operation, full or sparse, and thus the Lipschitz constant $\lambda_{\mathcal{L}}(\xi)$ of learning objective $\mathcal{L}(\Theta)$ in equation (4) (page 3) depends on both the dispersion and separation. The reviewer posits that the stability constants (and thus the Lipschitz constant of the learning objective) would be higher for heavy-hitter sparse attention than for standard full-attention far away from the optimal points (the trained model), and thus at initialization.
>
> We present the loss landscapes for full-attention (Figure 5 top row) and top-$k$ attention (Figure 5 middle row) with the center of the landscapes at (0,0) denoting the optimum. The loss landscapes of top-$k$ attention do not appear to be sharper than full-attention far away from the optimum (the center at (0,0)). We also present the estimated Lipschitz constants as we move farther away from the optimum (Figure 5 bottom row). The results indicate that, at the optimum --- the trained model, the estimated Lipschitz constants for both sparse and full attention are quite similar. However, as we move away from the optimum model, the estimated Lipschitz constants increase, with the Lipschitz constants for full-attention growing faster than those of the top-$k$ attention, implying that the loss landscapes are equally sharp near the optimal point, but the full-attention loss landscape gets sharper than top-$k$ attention as we move away from the optimum, counter to the reviewer's intuition.
>
> We do acknowledge a caveat that the Lipschitz constants are estimated. However, both our theory and empirical evaluations (and the actual observed convergence results presented in Section 3) imply that heavy-hitter sparse attention has more favourable loss landscapes for optimization.
>
> > _**Weakness (2).** Although in section 4 lines 243-259, the authors try to connet the convergence rate with Lipschitz constants, I still feel it is somehow far-fatched, a stable optimization algorithm doesn't mean this is a fast algorithm, especially when the following analysis consider the case to loss is close to the optimal point._
>
> We apologize for the confusion here. We do not claim or imply that stable optimization algorithms are faster algorithms. Rather, we claim that both the SGD convergence rate bounds for a learning problem and the algorithmic stability bounds of a learning algorithm depend on the Lipschitz constant of the learning objective --- these are established results. Thus, an improved Lipschitz constant provides improved guarantees for both SGD convergence and algorithmic stability guarantees.
>
> Furthermore, we do not make any assumption that the "loss is close to the optimal point", nor do the established results require this assumption. We also discuss the behaviours of the different attention mechanisms close and far from the optimal points in our response to weakness (1) above.
>
> > _**Weakness (3).** Note that in Figures 1, 2, 3, there is a clear loss plateau at initialization, and sparse attention can break this plateau faster compared with full attention. However, if we ignore this plateau, we can see that the convergence rate for sparse attention and full attention is close. In other words, as I analyzed in (1), I think the reason why sparse attention learns faster than full attention is that, at initialization, the loss landscape for sparse attention is sharper, thus the model can soon escape the plateau at initialization, so we can observe sparse attention learns faster._
>
> We thank the reviewer for their thoughtful insight. We have already discussed the behaviours of the different attention mechanisms close and far from the optimal points in our response to weakness (1) above. It is an interesting hypothesis that the initialization of full attention is on a "loss plateau". First, we would like to note that all models, using full or sparse attention, have the same number of parameters, and start learning from the same initialization. Second, we repeat each training 10 times from different initializations (and seeds for sampling training batches), and we present results (the learning curves) aggregated over these 10 trials.
>
> We note that while the training loss curve for full attention appears to "plateau" initially (for example in Figures 1 and 2), the full-attention training loss is actually changing (both increasing and decreasing because of an unfavourable loss landscape) as the training progresses. The effect of this change can be also seen from the variation (relatively small but present) in the heldout accuracy curves, indicating that the full-attention model is changing. This is usually different from a loss plateau where the loss gradient is very small, and thus the model changes are very small.
>
> Furthermore, for evaluations where the full-attention model does not have an initial apparent plateau, such as in Figure 3(e) with the Adam optimizer, top-$k$ sparse attention continues to converge faster (except in the case where both models diverge because of too large of an initial learning rate).
>
> Finally, we consider a small experiment with the PennTreeBank natural language dataset where we use the context of tokens to predict the next token. In this case, full attention does not have that apparent "plateau at initialization" but heavy-hitter sparse transformer continues to optimizer faster. For a 8-block transformer, trained for 50 epochs we see the following token misclassification training cross-entropy:
> |Method / Epoch|5|10|15|20|25|30|35|40|45|50|
> |-|-|-|-|-|-|-|-|-|-|-|
> |Full|3.29|3.08|2.97|2.89|2.83|2.79|2.74|2.72|2.69|2.67|
> |top-$k$ (% gain)|3.21 (2.4)|2.98 (3.2)|2.87 (3.4)|2.79 (3.5)|2.74 (3.2)|2.70 (3.2)|2.65 (3.3)|2.62 (3.7)|2.59 (3.7)|2.56 (4.1)|
>
> With a 16-block transformer model for 50 SGD epochs:
> |Method / Epoch|5|10|15|20|25|30|35|40|45|50|
> |-|-|-|-|-|-|-|-|-|-|-|
> | Full | 3.35 |3.13 |3.00 |2.93 |2.86 |2.81 |2.77 |2.73 |2.69 |2.67 |
> | top-$k$ (% gain) | 3.21 (4.2) |2.98 (4.8) |2.87 (4.3) |2.79 (4.8) |2.73 (4.5) |2.67 (5.0) |2.63 (5.1) |2.59 (5.1) |2.56 (4.8) |2.52 (5.6) |
>
> >  _**Weakness (4).** In Figure 4 in [1], there is no clear difference that sparse attention learns faster. Could the authors provide some explanation?_
> >
> > _[1] Gupta, Ankit, et al. "Memory-efficient Transformers via Top-$k$ Attention." arXiv preprint arXiv:2106.06899 (2021)._
>
> Regarding Figure 4(a) in Gupta et al. (2021), we really appreciate the reviewer making an appropriate connection with the ListOps dataset. There are of course experimental differences in terms of learning rates, schedulers, optimizers, training data, plotting granularities and such. But beyond those, we believe there are a few important points:
>
> - First, these plots show the eval set and test set accuracies, and not the training loss. So we can compare this Figure 4(a) in Gupta et al. (2021) to Figure 1(a) bottom row in our manuscript. Upon this comparison, we can see that in both cases, the held-out accuracies of top-$k$ attention with ListOps improves initially faster than full-attention. So these results are somewhat aligned (barring the plotting granularity).
>
> - Second, our results consider a much smaller value of $k=5$ in Figure 1(a) of our manuscript compared to Figure 4(a) in Gupta et al. (2021) where $k=128$. In our experiments, we see that a small $k=5$ is sufficient in terms of expressivity, achieving 100\% training accuracies for different problems and hyperparameters. The smallest $k$ considered in Gupta et al. (2021) is actually $k=64$. Smaller $k$  implies higher reduction in the semantic dispersion $\delta_h$ compared to the full-attention semantic dispersion $\delta_s$, and thus, better learning convergence guarantees.
>
> - Finally, Gupta et al. (2021) consider a transformer block which makes use of the top-$k$ operation not only in the attention, but also in the MLP (see **Feed-forward as attention** paragraph in Section 2.1 in Gupta et al. (2021)), thereby giving us different transformer blocks. Thus, it is hard to directly compare these two different transformer models.
>
> ------------
> ### References
> -  Gupta, Ankit, et al. "Memory-efficient Transformers via Top-$k$ Attention." arXiv preprint arXiv:2106.06899 (2021).

---

> > ### Comment · Reviewer_PEBb · 2025-08-04
> >
> > I appreciate the authors' efforts to provide explanations for my review. As stated in my original review, the experiments merely indicate that "the sparse attention learns faster than full attention **at initialization**," as shown in Figures 1,2. The authors claim that "sparse attention learns faster than full attention because top-k attention has better stability constants." However, "better stability constants" require certain conditions which are unlikely to occur at initialization.
> >
> > My main claims (or intuitions) are these two points:
> >
> > 1. The stability constants for full attention are better than those for top-k attention at initialization.
> >
> > 2. In Figures 1,2, there is a loss plateau for full attention, while sparse attention can break this plateau faster. This suggests that sparse attention learns faster mainly because the loss landscape for top-k attention is sharper, allowing it to learn faster at initialization.
> >
> > Regarding my first point, the authors claim that "heavy-hitter sparse attention can be better than the constants for full attention because of the exponential dependence on the semantic dispersion," with evidence in Table 4. However, this only computes semantic dispersions and semantic separation "over the whole training set with the trained model." Moreover, the bottom row of Figure 5 starts from the optimal point. How the constants change from the initialization point is not shown.
> >
> > Regarding my second point, the authors claim that "the full-attention training loss is actually changing, the gradient is not zero." However, given that the gradient norm and learning rate are comparable for both sparse and full attention, Figures 1,2 actually show that sparse attention has larger stability constants, as it is less stable and changes faster, contradictory to their previous claim.
> >
> > The authors also provide experiments on the PennTreeBank natural language dataset. Note that the absolute difference is about 0.09 for 8-block transformer and 0.14 for 16-block transformer. This aligns with my previous observation that the main difference between top-k attention and full attention comes from the starting point (i.e., before the 5-th epoch shown in the table). If we ignore the difference at initialization, we can see that the convergence rate for sparse attention and full attention is close.
> >
> > I thank the authors for their clarification regarding my misunderstanding of stability constants.
> >
> > In conclusion, the authors do not provide sufficient evidence to contradict my original claims 1 and 2, so currently I won't change my score.

---

> > > ### Author Response · Authors · 2025-08-04
> > >
> > > We are truly thankful for the reviewer to spend their time elaborating on their concern. This is very helpful for us as we try to clarify our contributions. Please see our responses to specific comments in the following:
> > >
> > > > _However, "better stability constants" require certain conditions which are unlikely to occur at initialization._
> > >
> > > Can the reviewer please clarify reasons or share references that would imply that the conditions "_are unlikely to occur at initialization._"?
> > >
> > > If the main concern is that the semantic separation $\Delta_h$ is close to zero at initialization (as the reviewer posited in their initial review), we discussed both in Appendix F.4 and in our responses that this semantic separation $\Delta_h$ can be quite close to zero even at the trained model, not just at initialization. However, the stability constants for heavy-hitter sparse attention **can still be much better than full attention even when semantic separation is very close to zero** if the semantic dispersion $\delta_s$ is large enough -- in practice, they are much larger than the values needed for the conditions to be satisfied. See Corollary 2 and Figures 19 and 20.
> > >
> > > > _1. The stability constants for full attention are better than those for top-k attention at initialization._
> > > >
> > > > [...]
> > > >
> > > > _Regarding my first point, the authors claim that "heavy-hitter sparse attention can be better than the constants for full attention because of the exponential dependence on the semantic dispersion," with evidence in Table 4. However, this only computes semantic dispersions and semantic separation "over the whole training set with the trained model." Moreover, the bottom row of Figure 5 starts from the optimal point. How the constants change from the initialization point is not shown._
> > >
> > > The bottom row of Figure 5 does start at the optimal point, showing that at the optimal point, the differences in the estimated Lipschitz constants between full attention and heavy-hitter sparse attention is quite low. However, as we move away from the optimal point on the loss landscape, **the Lipschitz constants of heavy-hitter sparse attention becomes significantly smaller than those of full attention.**
> > >
> > > Note that the "initializations" are part of the complete loss landscape. The loss landscape visualizations are generated by moving the model weights along random Gaussian directions from the optimal point (Li et al, 2018). So as we move farther away from the optimal point, the model weights essentially become Gaussian random samples. As "initializations" are usually Gaussian random samples, the loss landscape farther away from the optimal point depict the loss landscape near the initializations. Thus, as we are moving farther away from the optimal point on the visualized loss landscape, we are estimating the Lipschitz constants at regions that are effectively random Gaussian initializations. Our theoretical analyses indicate that heavy-hitter sparse attention will generally have better stability and thus better Lipschitz constants than full attention, and our empirical evidence supports this, especially at regions of the parameter space where the weights are effectively random Gaussian samples.
> > >
> > >
> > >
> > >
> > > ----------------
> > > ### References
> > > - Hao Li, Zheng Xu, Gavin Taylor, Christoph Studer, and Tom Goldstein. Visualizing the loss landscape of neural nets. Advances in neural information processing systems, 2018.

---

> > > ### Author Response · Authors · 2025-08-04
> > >
> > > > _2. In Figures 1,2, there is a loss plateau for full attention, while sparse attention can break this plateau faster. This suggests that sparse attention learns faster mainly because the loss landscape for top-k attention is sharper, allowing it to learn faster at initialization._
> > > >
> > > > [...]
> > > >
> > > > _Regarding my second point, the authors claim that "the full-attention training loss is actually changing, the gradient is not zero." However, given that the gradient norm and learning rate are comparable for both sparse and full attention, Figures 1,2 actually show that sparse attention has larger stability constants, as it is less stable and changes faster, contradictory to their previous claim._
> > >
> > > We would really appreciate if the reviewer can clarify what they mean by "sharper". If by "sharper" they mean larger Lipschitz constant for the learning objective, then we would really appreciate if the reviewer can share some reference that justifies the claim that "sharper" learning objectives can "learn faster at initialization".
> > >
> > > We request this reference because this is quite contrary to standard optimization literature where larger Lipschitz constants imply harder-to-optimize learning objectives, at initialization or otherwise, leading to worse convergence. If we are considering a strongly convex learning objective, and the initialization is already in the convex bowl of the loss, then larger Lipschitz constants can imply that the loss reduces faster. However, the learning loss with transformer based models are definitely not convex (let alone strongly convex).
> > >
> > > In the standard nonconvex loss of neural network models, larger Lipschitz constants imply that the objective can change unexpectedly, and can decrease or **increase** in the direction of gradient descent if the learning rate is not appropriate. This behaviour induced by the large Lipschitz constant hurts the learning convergence (at initialization or otherwise). In these standard cases, better Lipschitz constants imply better convergence.
> > >
> > > Finally, as we have mentioned in our rebuttal, even in the situations where there is no apparent "loss plateau" at initialization, as in Figure 3(e) with the Adam optimizer, the heavy-hitter sparse attention continues to converge faster than full attention.

---

> > > ### Author Response · Authors · 2025-08-04
> > >
> > > > _The authors also provide experiments on the PennTreeBank natural language dataset. Note that the absolute difference is about 0.09 for 8-block transformer and 0.14 for 16-block transformer. This aligns with my previous observation that the main difference between top-k attention and full attention comes from the starting point (i.e., before the 5-th epoch shown in the table). If we ignore the difference at initialization, we can see that the convergence rate for sparse attention and full attention is close._
> > >
> > > First, we would like to again clarify that all model training for the different attention mechanisms are initialized with the same model weights for each experimental trial (our results in the manuscript are aggregated over 10 trials).
> > >
> > > The reviewer seems to suggest that the main advantage of heavy-hitter sparse attention only occurs at the beginning of the learning. We already see this as a strength of heavy-hitter attention, validating our theoretical analyses.
> > >
> > > However, the reviewer seems to indicate that the initial faster convergence is the sole source of the improvement of heavy-hitter sparse attention over full attention. This does not invalidate our claim of faster convergence. However, we would also like to highlight that this is not always the case.
> > >
> > > In the following we elaborate on the PennTreeBank experiments and present results that show that the gain from the initial learning is actually improved upon throughout the learning, and the gap (improvement) between heavy-hitter sparse attention and full attention actually grows as we proceed (until of course both converge eventually).
> > >
> > > For example, for the 8-block model, we report (i) the improvement at the end of the first epoch, (ii) the largest improvement seen during training, and (iii) the epoch when the largest improvement materialized:
> > >
> > > - Improvement after first epoch: 0.066 (absolute), 1.68% (relative)
> > > - Largest absolute improvement: 0.117 at epoch 11/50
> > > - Largest relative improvement: 4.09% at epoch 47/50
> > >
> > > The same quantities for the 16-block model are as follows:
> > >
> > > - Improvement after first epoch: 0.09 (absolute), 2.22% (relative)
> > > - Largest absolute Improvement: 0.170 at epoch 32/50
> > > - Largest relative improvement: 4.09% at epoch 47/50
> > >
> > > We also considered a more challenging version of PennTreeBank with a larger vocabulary of 10000 tokens and a 10-block 4-headed model. We report the corresponding quantities:
> > >
> > > - Improvement after first epoch: 0.078 (absolute), 1.55% (relative)
> > > - Largest absolute improvement: 0.452 at epoch 41/100
> > > - Largest relative improvement: 49.16% at epoch 67/100
> > >
> > > As we see in these examples, the heavy-hitter sparse attention is able to improve upon full attention already at the end of the first epoch, which highlights its advantage, and supports our theoretical analyses. Moreover, these results also show that **improvement obtained by this faster convergence can increase throughout the learning, not just at initialization**, highlighting the more favourable loss landscape of the heavy-hitter sparse attention, allowing for faster convergence throughout.
> > >
> > > Here we also report the convergence for this problem for completeness:
> > >
> > > | Method / Epoch | 10 | 20 | 30 | 40 | 50 | 60 | 70 | 80 | 90 | 100 |
> > > |-|-|-|-|-|-|-|-|-|-|-|
> > > | Full | 3.22 |2.38 |1.76 |1.35 |0.98 |0.73 |0.51 |0.35 |0.24 |0.18 |
> > > | top-$k$ (% gain) | 3.10 (3.7%) |2.14 (10.1%) |1.41 (19.9%) |0.96 (28.9%) |0.63 (35.7%) |0.42 (42.5%) |0.29 (43.1%) |0.21 (40.0%) |0.15 (37.5%) |0.11 (38.9%) |

---

> > > > ### Comment · Reviewer_PEBb · 2025-08-08
> > > >
> > > > Thank you for the authors' response. Regarding my concern that "The stability constants for full attention are better than those for top-k attention at initialization," the authors have attempted to persuade me with two main arguments:
> > > >
> > > > 1. Although the loss for full attention remains almost unchanged at initialization while sparse attention loss changes rapidly, the authors suggest this actually indicates larger Lipschitz constants for full attention.
> > > >
> > > > 2. The authors argue that when starting far from the optimal point, model weights essentially become Gaussian random samples, thus eliminating the need to provide experiments starting from random initialization points.
> > > >
> > > > However, these arguments fail to convince me. I maintain my position that sparse attention learns faster primarily because the loss landscape for top-k attention is sharper, allowing it to learn more quickly at initialization. Main experimetns in this paper, Figures 1 and 2 clearly demonstrate that the primary loss difference emerges at initialization. Furthermore, top-k attention with random initialization, compared to full attention, is less stable (although the authors ask me to provide references for this claim, I believe this is a reasonably intuitive conclusion), moreover, the authors provide no experimental evidence that contradicts this intuition.
> > > >
> > > > While I still question aspects of the paper's main claim, I appreciate the authors' efforts to address my concerns and will adjust my rating accordingly.

---

> > > > > ### Author Response · Authors · 2025-08-08
> > > > >
> > > > > We thank the reviewer for their time and for sharing their intuitions, and for reevaluating our manuscript. We realize that all this takes time and effort, and we truly appreciate it.
> > > > >
> > > > > > _1. Although the loss for full attention remains almost unchanged at initialization while sparse attention loss changes rapidly, the authors suggest this actually indicates larger Lipschitz constants for full attention._
> > > > > >
> > > > > > _[...]_
> > > > > >
> > > > > > _However, these arguments fail to convince me. I maintain my position that sparse attention learns faster primarily because the loss landscape for top-k attention is sharper, allowing it to learn more quickly at initialization. Main experimetns in this paper, Figures 1 and 2 clearly demonstrate that the primary loss difference emerges at initialization._
> > > > >
> > > > > We would like to emphasize that sharper loss landscapes (implying larger Lipschitz constants) are generally harder to optimize even in the easier convex optimization setup. The largest learning rate allowing convergence is inversely proportional to the sharpness of the objective, implying that sharper objectives require smaller learning rates, and thus have slower convergences. This is related to the fact that, if the objective/loss changes rapidly, then we cannot take large steps in the direction of the gradient as we might overshoot the minima. This effect is only exacerbated in the complex nonconvex loss landscape of neural networks. And thus, sharper loss landscapes and larger Lipschitz constants imply slower convergence, contrary to the reviewer's intuition.
> > > > >
> > > > > Regarding the specific results in Figures 1 and 2 in the manuscript, both full attention and top-$k$ attention (as well as all other attention schemes) start training from the same random initialization (and same hyperparameters). However, it appears that full attention training is stuck in a "loss plateau" while top-$k$ attention training proceeds faster. While this is hard to verify in these high dimensional loss landscapes, we believe that this is because the sharper loss landscape of full attention makes it harder for the optimization to get into the basin of the local minima as it possibly keeps overshooting the basin. It is not a loss plateau but rather the optimization is unable to find and descend down the basin of the local minima. In contrast, the (relatively) smoother loss landscape of the top-$k$ attention allows the optimization to find the basin and stay in it, thus reducing the objective faster.
> > > > >
> > > > > Furthermore, we also provide various other experimental results in the manuscript and in the rebuttal where full attention does not have this "loss plateau" at initialization, and top-$k$ continues to outperform full attention significantly, with the improvement of full attention increasing as the optimization continues. These results imply that the primary loss differences do not emerge at initialization, but continue to grow throughout the learning process.
> > > > >
> > > > >
> > > > >
> > > > > > _2. The authors argue that when starting far from the optimal point, model weights essentially become Gaussian random samples, thus eliminating the need to provide experiments starting from random initialization points._
> > > > > >
> > > > > > _[...]_
> > > > > >
> > > > > > _Furthermore, top-k attention with random initialization, compared to full attention, is less stable (although the authors ask me to provide references for this claim, I believe this is a reasonably intuitive conclusion), moreover, the authors provide no experimental evidence that contradicts this intuition._
> > > > >
> > > > > As we mentioned in our responses, and detailed in Appendix F.5, we use the scheme from Li et al. (2018), which generates the 2D loss landscape by setting the trained model weights $\theta$ at the origin $(0,0)$ and computing the loss at a grid-point $(x,y)$ by considering the model with weights $(\theta + x \vartheta_1 + y \vartheta_2)$ where $\vartheta_1, \vartheta_2$ are random noise weights. As the values of $x,y$ increase, the model weights are essentially random noise --- this is a well established result, and the basis of schemes such as diffusion where adding enough noise to the signal makes its indistinguisable from the noise distribution. The bottom row of Figure 5 in the manuscript show the estimated Lipschitz constants at the origin (the trained model), and also show how the estimated Lipschitz constants vary as we move away from the origin to weights that are essentially random.
> > > > >
> > > > > It is in these regions away from the origin where top-$k$ has better Lipschitz constants than full attention. Thus we believe that we show experimental evidence of top-$k$ having more favourable learning objectives than full attention all across the loss landscape, including when the weights are random noise.
> > > > >
> > > > > -------
> > > > >
> > > > > ### References
> > > > > - Hao Li, Zheng Xu, Gavin Taylor, Christoph Studer, and Tom Goldstein. Visualizing the loss landscape of neural nets. Advances in neural information processing systems, 2018.

---

### Official Review · Reviewer_spof · 2025-07-05

**Clarity:** 2
**Significance:** 3
**Originality:** 3
**Rating:** 5
**Confidence:** 4

**Summary:**

The paper compares input-dependent and input-agnostic sparse attention in terms of learnability, convergence, and generalization, both empirically and theoretically. Firstly, through extensive experiments, the authors show that input-dependent sparsity enhances convergence and generalization, whereas the input-agnostic counterpart does not. Then, the paper theoretically argues that the top-k mask reduces the semantic dispersion, which improves the soft-max’s input-stability and lowers the Lipschitz constant of the training loss, leading to better algorithmic stability and faster convergence.

**Questions:**

The reviewer would love to see the concerns in the above addressed and revise the rating.

**Ethical Concerns:**

["NO or VERY MINOR ethics concerns only"]

**Final Justification:**

The detailed clarification by the authors has addressed my concerns. In the comment, I have encouraged the authors to integrate their explanations in a revised version of the paper. I am leaning towards accepting the paper.

**Limitations:**

yes

**Paper Formatting Concerns:**

-

**Quality:**

3

**Strengths And Weaknesses:**

## Clarity

1. Overall, the presentation is too dense which makes it hard to follow, and I strongly recommend the authors improve readability. This is explained in detail as follows.
   1. The figures 1,2,3,5 are very small, and their labels and legends are impossible to read. Further, each figure has many subplots, which at least to me is overwhelming.
- I am privileged by being able to zoom in with an electronic device, but a reader could have been reading it from a hard copy and not been able to read the figures.
- Consider suppressing some information to make space for the figures. For instance, in the caption of Figure 1, “The legend is the same across all datasets” seems removable without major damage. As another example, each row in figure 3 has four plots. Do we really need all four plots right here to make the point? It seems that two per row would be enough in the main text, and more figures could be left in the appendix.
  2. A large number of notations are defined throughout the paper, and it is sometimes difficult to remember all of them. Further, some notations have conflicting definitions.
- Let us take $\\beta$ for example. In page 6, $\\beta$ is used as the smoothness constant, but **the same symbol $\\beta$ is used on page 9 (line 352\) for a different meaning**. Notably, for the latter usage, the symbol was not given a name **until 50 pages later** in the paper where $\\beta$ is now called sink ratio.
- It would be great if there is a centralized table summarizing the symbol and its high-level meaning, and a link to where it is best contextualized. This would not only help readers keep track of the notations, but it also prevents overloaded definitions.
2. Minor typo: line 165 “absense”

## Significance

1. Theorem 5 relies on the assumption that $\\beta$ (the maximum number of queries attending to a single key) is bounded. While Table 4 gives empirical percentiles of $\\beta$, the theoretical guarantees depend on its worst-case value. Could the authors comment on the maximum observed values of $\\beta$ across the experiments?
2. How does the per-step computational cost of calculating the top-k mask compare to the baselines? The reason for this question is that, while top-k mask has better performance given the same number of steps, it may have a higher time-per-step. Therefore, a fairer comparison is to fix a time budget (instead of number of steps) for all the baselines and see their performance.

---

> ### Author Rebuttal · Authors · 2025-07-30
>
> We truly appreciate the reviewer's time and effort in thoroughly reviewing our manuscript and pointing out notational and typographical errors. We have tried to address all their comments in the following. We are thankful for their support, and we are happy to answer any further questions that can motivate the reviewer to raise their ratings.
>
> ---------
>
> >   _Overall, the presentation is too dense which makes it hard to follow, and I strongly recommend the authors improve readability._
>
> We apologize for the dense presentation, and we thank the reviewer for their helpful suggestion. We were trying to present an in-depth empirical analysis to the extent possible to highlight the robustness of the relative behaviour across experimental settings. We will update our manuscript to incorporate the reviewer's suggestions.
>
> > _A large number of notations are defined throughout the paper, and it is sometimes difficult to remember all of them. Further, some notations have conflicting definitions._
>
> We thank the reviewer for thoroughly reading our manuscript and identifying this notation overload. We really appreciate it! We will correct the discussion in the beginning of section 4 on page 6 to consider $\alpha_1$-Lipschitz and $\alpha_2$-smooth objectives (instead of the current $\alpha$-Lipschitz and $\beta$-smooth objective) to disambiguate the use of $\beta$ later.
>
> Furthermore, we will add a longer version of the following table to clarify notation:
>
>   | Symbol | Meaning |
>   | ------ | ------- |
>   | $X$    | Input string of token indices |
>   | $f$    | Ground truth function |
>   | $y$    | Output $f(X)$ |
>   | $\mathcal{V}$ | Vocabulary |
>   | $D$ | Number of tokens in the vocab |
>   | $L$ | Sequence length |
>   | $\mathbf{X}$ | Sequence embeddings |
>   | $\mathbf{W} = \mathbf{Q}^\top \mathbf{K} $ | Query-key projection matrix |
>   | $\mathbf{V}$ | Value projection matrix |
>   | $\mathbf{P}$ | MLP first layer weights |
>   | $\mathbf{R}$ | MLP second layer weights |
>   | $\textsf{LN}$ | Layer normalization |
>   | $\sigma$ | MLP activation |
>   | $\mathbf{M}$ | Mask matrix |
>   | $\mathbf{T}$ | Initial token embeddings |
>   | $\mathbf{E}$ | Positional embeddings |
>   | $\boldsymbol{\omega}$ | Token projection vector |
>   | $\boldsymbol{\phi}$ | Readout layer weights |
>   | $\theta^{(t)}$ | $t$-th transformer block parameters |
>   | $\Theta$ | Full model parameters |
>   | $f_\Theta$ | Learned model with parameters $\Theta$ |
>   | $k$ | Number of unmasked keys per query in sparse attention |
>   | $\xi$ | (Sparse) softmax stability |
>   | $\lambda_X(\xi)$ | Input stability constant for self-attention  |
>   | $\lambda_W(\xi)$ | Parameter $\mathbf{W}$ stability constant for self-attention |
>   | $\lambda_V$ | Parameter $\mathbf{V}$ stability constant for self-attention |
>   | $\lambda_\theta(\xi)$ | Per-transformer-block parameter stability |
>   | $\lambda_{\mathbf{X}}(\xi)$ | Per-transformer-block input stability |
>   | $\lambda_{\mathcal{L}}(\xi)$ | Learning objective Lipschitz constant |
>   | $\delta_s$ | Per-query maximum semantic dispersion for standard softmax attention |
>   | $\delta_r$ | Per-query maximum semantic dispersion for regular input-agnostic $k$-sparse attention |
>   | $\delta_h$ | Per-query maximum semantic dispersion for heavy-hitter $k$-sparse attention |
>   | $\Delta_h$ | Per-query minimum semantic separation for heavy-hitter $k$-sparse attention |
>   | $\beta k$ | The maximum number of queries that attend to a specific key |
>   | $\beta$ | The "sink ratio" |
>   | $\Xi$ | Per token embedding Euclidean norm upper bound |
>   | $\Gamma$ | Maximum spectral norm of the query-key projection matrix $\mathbf{W}$ |
>   | $\Upsilon$ | Maximum spectral norm of the value projection matrix $\mathbf{V}$ |
>
>
> > _Could the authors comment on the maximum observed values of $\beta$ across the experiments?_
>
> In the worst case, $\beta$ can be as large as $L / k$, where $L$ is the length of the input sequence, and $k$ is the number of unmasked keys per query token. This can happen especially in the later transformer blocks where the model starts aggregating all relevant information in certain tokens for putting together the final output, and all queries attend to these "aggregating keys". In all our experiments, $L \in [40, 600]$ and the smallest $k$ is 5, thus, the worst case $\beta \in [8, 120]$.
>
> However, note that $\beta$ shows up only in the input-stability constant $\lambda_X(\xi_h)$, and not in the parameter-stability constants $\lambda_W(\xi_h)$ and $\lambda_V$ in Theorem 5 for the heavy-hitter attention. From our detailed comparison between full attention and heavy-hitter attention in Appendix F.4, $\beta$ is within a logarithm term; see equation (277) in Corollary 2 where the relative performance of full and heavy-hitter sparse attention depends on $\beta$ approximately as $O(\log \beta)$. Thus $\beta$ has a smaller effect than the full attention dispersion $\delta_s$.
>
> At the risk of introducing further notation, we believe that the dependence on the worst case $\beta$ can be improved to depend on the distribution of the per-key sink-ratios (as well as the other parameters) in Theorem 5. For this reason, we present the different percentiles of the actual dispersions, separations and sink ratios.
>
> We also want to apologize for a mistake in Table 4, where we actually report the distribution of $\beta k$, the number of queries attending to a specific key, and not the sink ratio $\beta$ distribution, thus the actual values of $\beta$ are even lower as they would be divided by $k = 5$. Below, we present an updated version of Table 4 (Top) where we report both $\beta k$ and $\beta$ for different percentiles (including the worst case $\beta$ value):
>
> | Dataset |  $\delta_s$ [75%, 90%, 95%, max]  | $\delta_h$ [75%, 90%, 95%, max]  |  $\Delta_h$ [25%, 10%, 5%, min] |  $\beta k$ [75%, 90%, 95%, max] |   $\beta$ [75%, 90%, 95%, max] |
> | ----  | ----- | ---- | ----- | ---- | ---- |
> | ListOps | [8.61, 18.5, 29.4, 87.8] | [3.51, 6.74, 9.67, 28.2] | [0.016, 0.005, 0.002, 1e-9] | [1, 3, 15, 598] |  [0.2, 0.6, 3.0, 119.6] |
> | Parity | [8.30, 10.1, 11.2, 19.4] | [2.31, 3.13, 3.78, 9.16] | [0.062, 0.022, 0.011, 1e-6] | [8, 13, 16, 33] | [1.6, 2.6, 3.2, 6.6 ] |
> | EvenPairs | [2.03, 4.73, 9.44, 14.6] | [1.03, 2.84, 5.50, 8.25] | [0.009, 0.003, 0.002, 3e-8] | [6, 17, 26, 40] | [1.2, 3.4, 5.2, 8.0] |
> | MissingDup | [4.63, 9.25, 17.1, 23.9] | [2.36, 4.25, 4.88, 10.5] | [0.018, 0.006, 0.003, 1e-7] | [7, 15, 21, 40] | [1.4, 3.0, 4.2, 8.0 ] |
>
> The actual worst-case $\beta$ values are close to $L/k$. However, the worst-case $\delta_s$ values are also much higher than their 95-th percentile values, thus making the overall bound still favourable for heavy-hitter attention. In the following, we report the updated left-hand-sides of equation (276) and (277) as in Table 4 (bottom), including the worst case values for all the involved parameters (note that the LHS (277) values are slightly lower than the ones reported in the paper as we corrected the use of $\beta$ instead of $\beta k$ though the effect is minor as $\beta$ is in the logarithm):
>
> | Dataset |  LHS (276)  |  LHS (277) |
> | ----  | ----- | ---- |
> | ListOps | [0.97, 0.69, 0.56, 0.57] | [0.90, 0.67, 0.59, 0.61] |
> | Parity  | [0.70, 0.76, 0.80, 1.22] | [0.72, 0.81, 0.86, 1.29] |
> | Even Pairs | [3.17, 1.98, 1.31, 1.8] | [3.02, 2.10, 1.42, 1.90] |
> | Missing Dup. | [1.53, 1.09, 0.67, 1.41] | [1.53, 1.15, 0.71, 1.20] |
>
> As can be seen, even with ListOps, where $\beta \approx 120$, the bounds are quite favourable for heavy-hitter attention because of the outsized effect for the larger standard full-attention semantic dispersion $\delta_s \approx 8$ compared to $\delta_h \approx 28$.
>
> > _How does the per-step computational cost of calculating the top-k mask compare to the baselines?_
>
> Thank you for this great question. Naively computing the top-$k$ mask would require $O(L \log k)$ per-query (as opposed to the $O(L)$ cost per query for standard attention). Gupta et al. (2021) focus on the memory efficiency of top-$k$ attention without improving the runtime of the attention. However, various advancements have sped up the computational cost of top-$k$ attention both in terms of runtime and memory overhead such as Zeng et al. (2025). This has shown significant speedup over pytorch native attention implementation, as well as the optimized FlashAttention, with the speedup increasing with context length. Thus, top-$k$ attention can now be significantly faster than standard full attention in terms of actual runtime. Furthermore, other heavy-hitter attention such as Reformer (Kitaev et al., 2020) can also speed up the actual attention runtime with locality sensitive hashing.
>
> Our focus has been on a complimentary angle where we study how sparsity affects optimization convergence, and try to provide theoretical intuition as to when and how sparse attention can converge faster than standard full attention. In situations where top-$k$ (or any other $k$-heavy-hitter attention) would converge faster than standard full attention, the overall runtime would improve cumulatively with faster sparse attention and fewer steps to convergence.
>
> ------
> ### References
> - Gupta, Ankit, et al. "Memory-efficient Transformers via Top-$ k $ Attention." SustainNLP workshop (2021).
> - Zeng, Qiuhao, et al. "Zeta: Leveraging z-order curves for efficient top-k attention." ICLR (2025).
> - Kitaev, Nikita, et al. "Reformer: The efficient transformer." ICLR (2020).

---

> > ### Comment · Reviewer_spof · 2025-08-08
> > **Thanks to the authors for their clarification**
> >
> > Thank you for the detailed clarification! It has addressed my concerns. I encourage the authors to integrate them in a revised version of the paper. I have increased my rating accordingly.

---

### Note · Authors · 2025-08-16

We focus on learning convergence & generalization of sparse attention transformers. We present empirical results that show that heavy-hitter sparse attention (such as top-$k$ attention) significantly and consistently improve over full attention (while other sparse attention do not). Then we perform a novel theoretical analysis leveraging the notion of **_semantic dispersion_** which provide conditions for such behaviour, and we verify that these conditions do hold in practice.

We thank the reviewers for their time and thoughtful reviews. The following were the main technical points of discussion:

Reviewer PEBb shared their intuition that the loss surface of heavy-hitter attention is probably sharper than that of full attention (thus has larger Lipschitz constants) at random initialization, and thus leads to faster decrease in the loss but only at initialization.
- We note that larger Lipschitz constants hurt convergence, not help, even at initialization.
- We also estimated the Lipschitz constants in Figure 5 (bottom row) not only at the optimum but over the whole loss landscape, including various instantiations of random weights as one would have at initialization.
- This aligns with our theory that heavy-hitter attention has better Lipschitz constants than full attention and other sparse attention.
- Finally, we present additional results where the improvement of heavy-hitter attention over full attention increases at the optimization proceeds.

Reviewer XrxX highlighted that we only consider experiments on small-scale synthetic tasks.
- We presented results for a next-token prediction task with the PennTreeBank dataset which show that heavy-hitter attention continues to converge faster than full attention.

Reviewer yPUz raised concerns that sparse attention is of interest if they are more efficient, but the efficient forms of native-sparse attention lack expressivity and fail at tasks such as needle-in-a-haystack.
- We presented evidence from existing literature that heavy-hitter such as top-$k$ attention are extremely efficient in practice, and also solve such tasks.
- Our theoretical results show a separation between different forms of sparse attention, and highlight how the semantic dispersion is significantly reduced only with heavy-hitter style attention, but not with input-agnostic sparse attention (such as various native-sparse attention).

We hope that this helps the Area Chair better assess the strengths and limitations of our submission.

---

### Decision · Program_Chairs · 2025-09-17

**Decision:**

Accept (poster)

**Comment:**

This paper makes a strong case that input-dependent sparse attention (like top-k) helps transformers learn faster and generalize better than both standard full attention and simpler input-agnostic sparse methods. The authors back this up with solid, consistent experiments on smaller-scale tasks and a theoretical argument that this works by focusing the model's "semantic attention," which improves optimization stability. While the theory's bounds are somewhat loose and the experiments aren't large-scale, the core finding is robust and insightful. The reviewers' concerns about the loss landscape and scalability were adequately addressed in the rebuttal. Despite its limitations, the paper offers a valuable new perspective on sparsity, focusing on learning speed rather than just computational efficiency, which justifies acceptance.